

# PALADYN v1.0, a comprehensive land surface-vegetation-carbon cycle model of intermediate complexity

Matteo Willeit and Andrey Ganopolski

Potsdam Institute for Climate Impact Research (PIK), Potsdam, Germany

*Correspondence to:* Matteo Willeit (willeit@pik-potsdam.de)

**Abstract.** PALADYN is presented, a new comprehensive and computationally efficient land surface–vegetation–carbon cycle model designed to be used in Earth system models of intermediate complexity for long-term simulations and paleoclimate studies.

The model treats in a consistent manner the interaction between atmosphere, terrestrial vegetation and soil through the fluxes of energy, water and carbon. Energy, water and carbon are conserved. The model explicitly treats permafrost, both in physical processes and as important carbon pool. The model distinguishes 9 surface types of which 5 are different vegetation types, bare soil, land ice, lake and ocean shelf. Including the ocean shelf allows to treat continuous changes in sea level and shelf area associated with glacial cycles. Over each surface type the model solves the surface energy balance and computes the fluxes of sensible, latent and ground heat and upward shortwave and longwave radiation. It includes a single snow layer.

Vegetation and bare soil share a single soil column. The soil is vertically discretized into 5 layers where prognostic equations for temperature, water and carbon are consistently solved. Phase changes of water in the soil are explicitly considered. A surface hydrology module computes precipitation interception by vegetation, surface runoff and soil infiltration. The soil water equation is based on Darcy's law. Given soil water content, the wetland fraction is computed based on a topographic index. The temperature profile is also computed in the upper part of ice sheets and in the ocean shelf soil.

Photosynthesis is computed using a light use efficiency model. Carbon assimilation by vegetation is coupled to the transpiration of water through stomatal conductance. The model includes a dynamic vegetation module with 5 plant functional types competing for the gridcell share with their respective net primary productivity.





The model distinguishes between mineral soil carbon, peat carbon, buried carbon and shelf carbon. Each soil carbon 'type' has its own soil carbon pools generally represented by a litter, a fast and a

slow carbon pool in each soil layer. Carbon can be redistributed between the layers by vertical diffusion. For the vegetated macro surface type, decomposition is a function of soil temperature and soil moisture. Carbon in permanently frozen layers is assigned a long turnover time which effectively locks carbon in permafrost. Carbon buried below ice sheets and on flooded ocean shelfs is treated differently. The model also includes a dynamic peat module.

PALADYN includes carbon isotopes $^{13}$C and $^{14}$C, which are tracked through all carbon pools. Isotopic discrimination is modelled only during photosynthesis.

A simple methane module is implemented to represent methane emissions from anaerobic carbon decomposition in wetlands (including peatlands) and flooded ocean shelf.

The model description is accompanied by a thorough model evaluation in offline mode for the

present day and the historical period.

## 1   Introduction

Land surface models (LSMs) represent an essential component of Earth system models of different complexity. Currently LSMs simulate the interaction between atmosphere, vegetation, land surface and upper soil through the fluxes of energy, water and carbon. Modern LSMs are the result of a

gradual convergence of initially separate modeling approaches: climate, carbon cycle and vegetation dynamics models (e.g. Pitman, 2003; Sellers et al., 1997).

In the earlier climate models very simple land surface schemes with bucket hydrology and without explicit vegetation representation were used (Manabe, 1969). The 2nd generation LSMs (Sellers et al., 1997) simulated soil temperature and moisture in several layers and the water and energy

exchange between the land surface and the atmosphere were mediated by vegetation represented as a big leaf (e.g. (Deardorff, 1978), BATS (Dickinson et al., 1986) and SiB (Sellers et al., 1986)). This step was required because biological processes play a major role in controlling evapotranspiration. In 2nd generation LSMs the behaviour of leaf stomata, which controls the rate of transpiration of water from plants, was represented based on empirical relations with climate (e.g. Jarvis, 1976).

The 3rd generation of LSMs included additionally a mechanistic representation of photosynthesis (Farquhar et al.; Collatz et al., 1991) which could then directly be related to stomatal conductance used to compute transpiration (Ball et al., 1987; Leuning, 1995).

Terrestrial biogeochemical models followed a separate line of development. These models were designed to simulate the exchanges of carbon between the atmosphere and terrestrial ecosystems for

a given climate and geographic vegetation distribution (e.g. Raich et al., 1991; Melillo et al., 1993; Running and Coughlan, 1988; Running and Gower, 1991; Foley, 1994).



Equilibrium biogeography models were developed alongside terrestrial carbon cycle models to simulate the global vegetation distribution for given climatic conditions (Woodward, 1987; Prentice et al., 1992; Haxeltine and Prentice, 1996b; Neilson, 1995). However, equilibrium models do not simulate the processes of plant growth, competition and mortality that govern the dynamics of vegetation changes. Global dynamic vegetation models have been developed for this purpose (Haxeltine and Prentice, 1996b; Sitch et al., 2003; Cox, 2001; Friend et al., 1997; Foley et al., 1996; Woodward et al., 1998; Brovkin et al., 1997).

Since it was shown that climate-vegetation feedbacks may be important, the first attempts to incorporate interactive vegetation into climate models were made (Henderson-Sellers, 1993; Claussen, 1994). While during the 1990s and 2000s climate models and then Earth system models (ESMs) based on coupled general circulation models (GCMs) remained too expensive to perform long-term simulations, a new class of models - Earth system models of intermediate complexity (EMICS, Claussen et al. (2002)) - emerged. The EMIC CLIMBER-2 (Petoukhov et al., 2000; Ganopolski et al., 2001) was one of the first Earth system models which included both terrestrial carbon cycle and vegetation dynamics based on VECODE (Brovkin et al., 1997, 2002) and has been also used to estimate the strength of the climate–vegetation feedback (Willeit et al., 2014b) and the carbon cycle feedback (Willeit et al., 2014a). Later, similar and more comprehensive vegetation models were incorporated in both complex and intermediate complexity ESMs (e.g. Cox, 2001; Oleson et al., 2004; Krinner et al., 2005; Sato et al., 2007; Reick et al., 2013).

A limitation of previous land surface modeling approaches is that different model components are not necessarily consistent because initially they were developed as stand-alone models. Additionally, initially LSMs have been developed with the intention to capture the processes which are important for climate change projections on the time scales of centuries, thus missing processes which might play an important role on longer time scales. This was fully justified by the fact that complex ESMs were and are still too computationally expensive to be used on much longer time scales, such as for simulations of glacial cycles. Only recently some existing models have been adapted to include slower processes, for example peat carbon dynamics (Wania et al., 2009; Kleinen et al., 2012; Spahni et al., 2013; Stocker et al., 2014). However, the simulation of processes with long time scales, such as peat carbon accumulation or inert permafrost carbon dynamics, require necessarily a transient modelling approach, which is made feasible only by a fast model.

Here we present a new land surface model primarily designed for paleoclimate applications, and therefore named PAleo LAnd DYNamics model (PALADYN), although also applicable to many other types of studies, including multi-ensemble future projections. The model has been designed to represent the land processes which are thought to be important both on short and very long time scales. The physical and biochemical processes are consistently coupled with each other. The model is intended to be used in the next generation of the CLIMBER EMIC and to substitute VECODE. CLIMBER employs a statistical dynamical atmosphere model. This type of model does not explicitly





simulate weather and therefore PALADYN is designed to simulate climatological mean seasonal
cycle. Typical application of such model is simulations of Earth system dynamics on astronomical
and geological time scales. This is why particular attention is given to the selection of the proper
complexity of the different processes which are represented in order to capture the main feedbacks
in the system but at the same time maintain the model computationally efficient. We expect that
PALADYN can also be used in other EMICs since most of them still employ rather simplistic LSMs.

## 2   Model overview

PALADYN is designed to operate on coarse resolution required for long-term simulations. Here we
test the model on a 5x5°horizonal resolution.

In each grid cell the model distinguishes 9 surface types (5 vegetation types, bare soil, ice sheets,
lakes and ocean shelf) (Fig. 1a). All surface type fractions can change over time. The fraction of
vegetation types and bare soil is computed by the dynamic vegetation module. The model is also able
to handle changes in the fraction of ice sheet and ocean shelf, when given as input. This is necessary
to simulate glacial cycles. So far lakes are implemented in the model only as a placeholder.

Over each surface type, except ocean shelf, the model solves the surface energy balance and com-
putes the fluxes of sensible, latent and ground heat and upward shortwave and longwave radiation.

Vegetation and bare soil share a single soil column (Fig. 1b) where temperature, moisture and
carbon are discretized in 5 vertical layers reaching down to a depth of 3.9 m. The top soil layer is
20 cm thick. A single snow layer with pronostic temperature and density is included in the model on
top of the soil column. A 1d-heat diffusion equation is solved to compute snow and soil temperature
with the ground heat flux as top boundary condition. Snowmelt and phase changes of water in the soil
are explicitly considered. A surface hydrology module computes rainfall intercepted by vegetation,
surface runoff and infiltration. Infiltration provides the top boundary condition for the solution of
soil water equation based on Darcy's law. Given soil water content the wetland fraction is computed
following a simplified TOPMODEL approach (Niu et al., 2005).

For the ice sheet fraction, the temperature of the snow layer and of the top 3.9 m of ice below is
computed in the same way as for the soil, but phase changes in the ice are inhibited. The temperature
of the soil below the shelf water is needed to calculate the decomposition rate of shelf carbon. It is
computed from a 1-d diffusion equation with the shelf water temperature prescribed as top boundary
condition and assuming that the soil is saturated with liquid and/or frozen water. Phase changes are
accounted for.

Photosynthesis is computed following Sitch et al. (2003); Haxeltine and Prentice (1996a, b). Car-
bon assimilation by vegetation is coupled to the transpiration of water vapor through stomatal con-
ductance. PALADYN includes a dynamic vegetation module based on TRIFFID (Cox, 2001) with 5
plant functional types competing for the gridcell share with their respective net primary production.




PALADYN includes a representation of soil carbon processes, including slow processes that are
130 thought to be relevant over multimillenial time scales associated with glacial-interglacial transitions
when the appearance and disappearance of continental ice sheets, changes in sea level and land area
can potentially strongly affect the land carbon cycle. PALADYN therefore includes processes with
a long time scale, such as accumulation of carbon in peatlands, inert carbon locked in perenially
frozen ground and carbon buried below ice sheets. It also accounts for changes in land area due to
135 sea level variations and isostatic adjustment of the litosphere to the ice sheet loading. During periods
of low sea level the model allows vegetation to grow on exposed ocean shelfs. When sea level is
rising the exposed shelf becomes flooded and the vegetation dies.

The soil of the vegetated grid cell part, below the ice sheet and below the shelf water has its own
carbon pools (Fig. 1c) represented in general by a litter, a fast and a slow carbon pool in each soil
layer. Carbon can be redistributed between the layers by vertical diffusion. For the vegetated part,
decomposition of organic matter is a function of soil temperature and soil moisture. Carbon in per-
manently frozen layers is assigned a long turnover time which effectively locks carbon in permafrost.
Carbon buried below ice sheets and carbon on the flooded ocean shelf are treated separately.

A representation of peatland dynamics is also included in PALADYN. In inundated areas, peat is
145 formed by accumulating carbon at the surface in the seasonally anoxically decomposing acrotelm.
When the acrotelm carbon exceeds a critical value, carbon is transferred to the catotelm below the
water table.

PALADYN also includes carbon isotopes $^{13}$C and $^{14}$C, which are tracked trough all carbon pools.
Isotopic discrimination is modelled only during photosynthesis.
A simple methane module is implemented to represent methane emissions from anaerobic carbon
decomposition in wetlands (including peatlands) and flooded ocean shelf.

The processes represented in PALADYN are illustrated in Fig. 2 and Fig. 3.

All physical model components and photosynthesis are integrated with an implicit timestepping
scheme with a time step of one day. Dynamic vegetation and soil carbon processes are integrated
with an implicit timestepping scheme with a time step of one month.

The model is written in FORTRAN90 and uses the NCIO package (Robinson and Perrette, 2015)
to handle input and output of data.

This paper describes the model representation of processes over ice free land. Processes related to
changes in land-ice-ocean mask, buried and ocean shelf carbon will be described in a forthcoming
paper.

## 3 Surface energy balance and fluxes

The surface energy balance equation at the land surface is written as:

$$(1-\alpha)SW^{\downarrow} + \epsilon LW^{\downarrow} - LW^{\uparrow} - H - LE - G = 0, \tag{1}$$



where $\alpha$ is surface albedo, $SW^{\downarrow}$ is the incoming shortwave radiation, $\epsilon$ is the surface emissivity

for longwave radiation, $LW^{\downarrow}$ and $LW^{\uparrow}$ are the incoming and outgoing longwave radiation at the surface, $H$ is the sensible heat flux, $LE$ is the latent heat flux and $G$ the ground heat flux. Equation (1) is then solved for the skin temperature, $T_{\star}$, using the formulations for the energy fluxes described next. All symbols are defined in Table 1.

The surface emitted longwave radiation is given by the Stefan-Boltzmann law with a surface type

dependent emissivity $\epsilon$ to account for the fact that the surface is not a perfect black body:

$$LW^{\uparrow} = \epsilon\sigma T_{\star}^4. \tag{2}$$

The sensible heat flux is computed from the temperature gradient between the surface and a reference height above the surface and an aerodynamic resistance, $r_{\mathrm{a}}$ (section 3.3), using the bulk aerodynamic formula:

$$H = \frac{\rho_{\mathrm{a}}C_{\mathrm{p}}}{r_{\mathrm{a}}}(T_{\star} - T_{\mathrm{a}}), \tag{3}$$

where $\rho_{\mathrm{a}}$ is air density, $C_{\mathrm{p}}$ is the specific heat of air, $r_{\mathrm{a}}$ is the aerodynamic resistance and $T_{\mathrm{a}}$ is the temperature of the air at a reference level $z_{\mathrm{ref}}$.

Similarly, the latent heat flux over unvegetated surfaces is expressed in terms of the specific humidity gradient between the surface and a reference atmospheric level with the addition of a factor

$\beta_{\mathrm{s}}$ (section 3.4) representing a possible limitation in the moisture supply:

$$LE = L\frac{\rho_{\mathrm{a}}}{r_{\mathrm{a}}}\beta_{\mathrm{s}}\left(q_{\mathrm{sat}}(T_{\star}) - q_{\mathrm{a}}\right). \tag{4}$$

$L$ is the latent heat of vaporisation, $q_{\mathrm{sat}}$ is the specific humidity at saturation and $q_{\mathrm{a}}$ is the specific humidity of air. Over vegetation the latent heat flux consists of contributions from transpiration of water vapour through leaf stomata during photosynthesis, soil/snow evaporation and sublimation

from below the canopy and evaporation of rainfall intercepted by the canopy:

$$LE = L\frac{\rho_{\mathrm{a}}}{r_{\mathrm{a}} + r_{\mathrm{s}}}\left(q_{\mathrm{sat}}(T_{\star}) - q_{\mathrm{a}}\right) + L\frac{\rho_{\mathrm{a}}}{r_{\mathrm{a}} + r_{\mathrm{a,can}}}\beta_{\mathrm{s}}\left(q_{\mathrm{sat}}(T_{\mathrm{s,1}}) - q_{\mathrm{a}}\right) + LE_{\mathrm{can}}. \tag{5}$$

$r_{\mathrm{a,can}}$ is the aerodynamic resistance between the soil surface and the vegetation canopy (section 3.3) and $r_{\mathrm{s}}$ is the canopy resistance to water vapor flux through the leaf stomata as described in detail in section 3.4). $T_{\mathrm{s,1}}$ is the temperature of the top soil layer, or the snow layer temperature if snow

is present. Canopy evaporation and sublimation, $E_{\mathrm{can}}$, is computed using the skin temperature from the previous time step as described in section 5.1.

The ground heat flux is represented by conduction of heat between the skin and the center of the snow layer or top soil layer:

$$G = 2\lambda_{\mathrm{s,1}}\frac{T_{\star} - T_{\mathrm{s,1}}}{\Delta z_1}. \tag{6}$$

$\lambda_{\mathrm{s,1}}$ is the heat conductivity and $\Delta z_1$ is the thickness of the snow layer or top soil layer.



The prognostic terms in $T_\star$ in the formulation of the surface energy fluxes are then linearized using Taylor series expansion assuming that the temperature at the new time step, $T_{\star,n+1} = T_{\star,n} + \Delta T$ with $\Delta T_\star \ll T_\star$:

$$T_{\star,n+1}^4 = T_{\star,n}^4 + 4T_{\star,n}^3(T_{\star,n+1} - T_{\star,n}), \qquad (7)$$

$$q_{\text{sat}}(T_{\star,n+1}) = q_{\text{sat}}(T_\star) + \left.\frac{dq_{\text{sat}}}{dT_\star}\right|_{T_\star = T_{\star,n}} (T_{\star,n+1} - T_{\star,n}). \qquad (8)$$

Equation (1) can then be solved explicitly for the skin temperature at the new time step, $T_{\star,n+1}$, separately for each surface type.

If snow is present and the skin temperature is above freezing the surface energy fluxes are diag-

205 nosed first with the skin temperature greater then $0\,^\circ\mathrm{C}$ and then with skin temperature set to $0\,^\circ\mathrm{C}$. The difference between the sum of the energy fluxes is then used to melt part of the snow layer.

Given the new skin temperatures, the ground heat flux $G$ and its derivative with respect to top soil or snow temperature ($\partial G/\partial T_{s,1}$) are diagnosed and used as input for the 1d soil heat diffusion equation. After the top-soil/snow temperature has been updated as described in section 4, it is used

to compute the total ground heat flux $G_{\text{new}} = G + \partial G/\partial T_{s,1}\Delta T_{s,1}$. Skin temperature is then updated using $G_{\text{new}}$ and all remaining surface energy and water fluxes are diagnosed.

In the next sections the surface parameters needed for the solution of the surface energy balance equation are described.

### 3.1 Surface albedo

PALADYN distinguishes between direct and diffuse albedo in the visible and infrared spectral bands. For ice sheets and bare soil the surface albedo is computed as a weighted mean of snowfree ($\alpha_{\text{snfree}}$) and snow ($\alpha_{\text{sn}}$) albedoes:

$$\alpha = f_{\text{sn}}\alpha_{\text{sn}} + (1 - f_{\text{sn}})\alpha_{\text{snfree}}, \qquad \text{bare soil, ice sheets.} \qquad (9)$$

The soil albedo in the visible and near infrared band are prescribed from (Dazlich and Los, 2009).

The fraction considered to be snow covered depends on snow height ($h_{\text{sn}}$) and snowfree roughness length ($z_0^{\text{snfree}}$) of the surface (Section 3.3) following Oleson et al. (2004):

$$f_{\text{sn}} = \frac{h_{\text{sn}}}{h_{\text{sn}} + 10z_0^{\text{snfree}}}. \qquad (10)$$

The albedo of grass and shrubs is computed by additionally separating the snowfree albedo into bare soil and canopy albedo through a sky view factor $f_{\text{sv}}$:

$$\alpha = f_{\text{sn}}\alpha_{\text{sn}} + (1 - f_{\text{sn}})(1 - f_{\text{sv}})\alpha_{\text{snfree}}^{\text{can}} + (1 - f_{\text{sn}})f_{\text{sv}}\alpha_{\text{soil}}, \qquad \text{grass, shrubs.} \qquad (11)$$





The sky view factor is a function of the leaf area index ($L_{\mathrm{ai}}$), the stem area index ($S_{\mathrm{ai}}$) and an extinction coefficient $k_{\mathrm{ext}}$ (Table 2) (e.g. Otto et al., 2011):

$$f_{\mathrm{sv}} = \exp\left[-k_{\mathrm{ext}}(L_{\mathrm{ai}} + S_{\mathrm{ai}})\right]. \tag{12}$$

The forest albedo is computed as a weighted mean of canopy albedo ($\alpha_{\mathrm{can}}$) and albedo of the ground below the canopy ($\alpha_{\mathrm{g}}$):

$$\alpha = f_{\mathrm{sv}}\alpha_{\mathrm{g}} + (1 - f_{\mathrm{sv}})\alpha_{\mathrm{can}}, \qquad \text{forest.} \tag{13}$$

The direct beam sky view factor for forest includes a daily radiation weigthed solar zenith angle ($\mu$) dependence following Campbell and Norman (1989):

$$f_{\mathrm{sv}}^{\mathrm{dir}} = \exp\left[-k_{\mathrm{ext}}\frac{(L_{\mathrm{ai}} + S_{\mathrm{ai}})}{\cos\mu}\right]. \tag{14}$$

The sky view factor for diffuse radiation is derived by fitting the relation given by Verseghy et al. (1993) and is taken to be:

$$f_{\mathrm{sv}}^{\mathrm{dif}} = \exp\left[-k_{\mathrm{ext}}\frac{(L_{\mathrm{ai}} + S_{\mathrm{ai}})}{\cos 45°}\right]. \tag{15}$$

The albedo of the ground below the canopy, $\alpha_{\mathrm{g}}$, is computed as in Eq. (9). $\alpha_{\mathrm{can}}$ varies between snowfree canopy albedo and snow-covered canopy albedo depending on the canopy fraction covered by snow:

$$\alpha_{\mathrm{can}} = f_{\mathrm{sn}}^{\mathrm{can}}\alpha_{\mathrm{sn}}^{\mathrm{can}} + (1 - f_{\mathrm{sn}}^{\mathrm{can}})\alpha_{\mathrm{snfree}}^{\mathrm{can}}. \tag{16}$$

The canopy fraction covered by snow, $f_{\mathrm{sn}}^{\mathrm{can}}$, is described in Section 5.1. For $\alpha_{\mathrm{snfree}}^{\mathrm{can}}$ the PFT specific values derived from MODIS data in Houldcroft et al. (2009) for the TRIFFID PFTs are used and $\alpha_{\mathrm{sn}}^{\mathrm{can}}$ values are taken from Moody et al. (2007) based on MODIS data (Table 5).

Snow albedo is parameterized as a function of the solar zenith angle and a snow ageing factor. The diffuse albedo of freshly fallen snow is set to 0.95 in the visible band and to 0.65 in the near infrared band. The actual albedo of snow for diffuse radiation depends on a snow age factor $f_{\mathrm{age}}$:

$$\alpha_{\mathrm{sn}}^{\mathrm{vis,dif}} = \alpha_{\mathrm{sn,fresh}}^{\mathrm{vis,dif}} - 0.05 f_{\mathrm{age}}, \tag{17}$$

$$\alpha_{\mathrm{sn}}^{\mathrm{nir,dif}} = \alpha_{\mathrm{sn,fresh}}^{\mathrm{nir,dif}} - 0.25 f_{\mathrm{age}}. \tag{18}$$

The snow age factor is intended to represent the effect of snow grain size increase on albedo (Warren and Wiscombe, 1980). For simplicity and to account for the fact that a statistical dynamical atmosphere does not resolve single snowfall events but rather returns a smoothly varying daily snowfall rate, $f_{\mathrm{age}}$ is parameterized as a function of skin temperature and snowfall rate as described in Appendix A. If the skin temperature is at melting point, the snow albedo is further reduced by 0.2 to account for the formation of melt ponds.



The direct beam snow albedo is then computed as (Dickinson et al., 1986):

$$\alpha_{\mathrm{sn}}^{\mathrm{dir}} = \alpha_{\mathrm{sn}}^{\mathrm{dif}} + 0.4 f_\mu (1 - \alpha_{\mathrm{sn}}^{\mathrm{dif}}), \tag{19}$$

where the solar zenith angle factor is slightly modified from Dickinson et al. (1986) to correct for the bias highlighted by Gardner and Sharp (2010):

$$f_\mu = 0.5 \left( \frac{3}{1 + 2\cos\mu} - 1 \right). \tag{20}$$

### 3.2 Surface emissivity

The broadband emissivity ($\epsilon$) used to compute the net longwave radiation at the surface is a surface-type dependent parameter. It is taken to be equal to 0.96 for all vegetation types, 0.9 for bare soil, 0.99 for snow-covered ground and 0.99 for ice (Jin and Liang, 2006; Walters et al., 2014).

### 3.3 Aerodynamic resistances

The aerodynamic resistance, $r_{\mathrm{a}}$, is computed for each surface type accounting for atmospheric stability through a bulk Richardson number following BATS (Dickinson et al., 1986). The drag coefficients for neutral stratification are obtained from boundary–layer theory:

$$C_{\mathrm{DN}}^{\mathrm{m}} = \left[ \kappa \ln \left( \frac{z_{\mathrm{ref}} - d}{z_{\mathrm{m}}} \right) \right]^2, \qquad \text{drag coefficient for momentum} \tag{21}$$

$$C_{\mathrm{DN}}^{\mathrm{h}} = \kappa^2 \ln \left( \frac{z_{\mathrm{ref}} - d}{z_{\mathrm{m}}} \right) \ln \left( \frac{z_{\mathrm{ref}} - d}{z_{\mathrm{h}}} \right), \qquad \text{drag coefficient for heat and water.} \tag{22}$$

$z_{\mathrm{ref}}$ is a reference height above the surface and $d$ is the zero-plane displacement, the height above the ground at which zero wind speed is achieved, and depends on the surface type. $z_{\mathrm{m}}$ is the roughness length for momentum and is computed as the weighted mean of the roughness length of snow ($z_0^{\mathrm{sn}}$) and the roughness length of the snow-free surface ($z_0^{\mathrm{snfree}}$):

$$z_{\mathrm{m}} = f_{\mathrm{sn}} z_0^{\mathrm{sn}} + (1 - f_{\mathrm{sn}}) z_0^{\mathrm{snfree}}. \tag{23}$$

A logarithmic averaging would be more appropriate here (Zeng and Wang, 2007), but for computational efficiency a simple linear weighting is prefered. This simplification does not significantly affect the model results. The snow covered fraction is given by Eq. (10) for all surface types. For vegetated surfaces, $z_0^{\mathrm{snfree}}$ is given by a weighted mean of vegetation ($z_0^{\mathrm{v}}$) and bare soil ($z_0^{\mathrm{b}}$) roughness:

$$z_0^{\mathrm{snfree}} = V z_0^{\mathrm{v}} + (1 - V) z_0^{\mathrm{b}}. \tag{24}$$

where the weight $V$ depends on the vegetation state:

$$V = \frac{L_{\mathrm{ai}} + S_{\mathrm{ai}}}{(L_{\mathrm{ai}} + S_{\mathrm{ai}})_{\mathrm{crit}}}, \tag{25}$$





and is limited to be lower than 1. The critical value of $(L_{ai} + S_{ai})_{crit}$ is set to 2. Zeng and Wang

(2007) showed that model results are not very sensitive to the formulation of $V$. $z_0^v$ is taken as 1/10

of the vegetation height and the displacement $d = 0.7 V h_v$. Vegetation height $h_v$ varies over time for

each PFT and differs between PFTs. For bare soil, snow and ice $d = 0$ and the values of $z_0$ are given

in Table 2.

In general the roughness length for heat and water vapor differs from the roughness length for

momentum and is defined by (Garratt, 1994; Milly and Shmakin, 2002):

$$\ln\left(\frac{z_m}{z_h}\right) = 2. \tag{26}$$

$z_h$ is therefore almost an order of magnitude smaller than $z_m$.

Although the surface energy balance equation in PALADYN is solved with a daily time step,

which implies that the diurnal cycle in atmospheric stability can not be resolved by the model,

the inclusion of a simple Richardson number dependence based on daily mean temperatures in the

computation of the drag coefficients significantly improves the simulated surface energy fluxes when

the stratification is unstable. The bulk Richardson number is calculated as (Dickinson et al., 1986):

$$R_i = g z_{ref} \frac{(1 - T_\star/T_a)}{V_a^2 + 1}, \tag{27}$$

where $g$ is the acceleration due to gravity and $V_a$ is the wind speed at the reference level $z_{ref}$. The

drag coefficients for the unstable case ($R_i < 0$) are then adjusted to account for atmospheric stability:

$$C_D^{m/h} = C_{DN}^{m/h}\left(1 + 24.5\sqrt{-C_{DN}^{m/h} R_i}\right) \quad R_i < 0. \tag{28}$$

Finally, the aerodynamic resistance for sensible and latent heat flux is given by:

$$r_a = \frac{1}{C_D^h V_a} \tag{29}$$

The aerodynamic resistance for the transfer of heat and water between the ground and the canopy

is parameterized partly following Zeng et al. (2005):

$$r_{a,can} = \frac{1}{C_{can} u_\star}. \tag{30}$$

$u_\star = V_a C_D^h \sqrt{C_D^h}$ is the friction velocity and represents the wind speed incident on the leaves. The

drag coefficient $C_{can}$ is given by:

$$C_{can} = W C_{bare} + (1 - W) C_{dense}. \tag{31}$$

The weight $W$ depends on the leaf and stem area index, $W = e^{-(L_{ai} + S_{ai})}$. $C_{can}$ varies between

$C_{dense}$ for dense canopy and $C_{bare}$ when $W$ tends to zero. The values of $C_{dense}$ and $C_{bare}$ are given

in Table 2.





### 3.4 Surface resistance to water vapor fluxes

Additionally to the aerodynamic resistances, the flux of water vapor from the ground or canopy is
subject to additional resistances. For evaporation from bare soil this surface resistance is represented
in terms of a $\beta_{\mathrm{s}}$-factor. Different parameterisations of $\beta_{\mathrm{s}}$ have been proposed to be used in global cli-
mate models (e.g. Mahfouf and Noilhan, 1991). The model, in particular the geographic distribution
and extent of modelled bare soil, is strongly dependent on the formulation of the $\beta_{\mathrm{s}}$ factor. Thus var-
ious surface resistance formulations for bare soil evaporation are implemented in PALADYN with
the default $\beta_{\mathrm{s}}$ depending on top-soil moisture ($\theta_1$) and field capacity ($\theta_{\mathrm{fc}}$) following Lee and Pielke
(1992):

$$\beta_{\mathrm{s}} = \begin{cases} \frac{1}{4}\left[1 - \cos\left(\pi\frac{\theta_1}{\theta_{\mathrm{fc}}}\right)\right]^2 & \theta_1 < \theta_{\mathrm{fc}} \\ 1 & \theta_1 \geq \theta_{\mathrm{fc}}. \end{cases} \tag{32}$$

The resistance for transpiration of water through the leaf stomata is coupled to the uptake of
carbon during photosynthesis and is simply the inverse of the canopy conductance calculated by the
photosynthesis module (section 6.1) after conversion to units of $\mathrm{m\,s^{-1}}$:

$$r_{\mathrm{s}} = \frac{1}{g_{\mathrm{can}}}. \tag{33}$$

Evaporation and sublimation from the canopy and sublimation from snow and ice are assumed to
occur without surface resistance ($r_{\mathrm{s}} = 0$ and $\beta_{\mathrm{s}} = 1$).

## 4 Snow and soil temperature

The heat transfer in the snow-soil column is represented by a one-dimensional heat diffusion equa-
tion:

$$c\frac{\partial T_{\mathrm{s}}}{\partial t} = \frac{\partial}{\partial z}\left[\lambda\frac{\partial T_{\mathrm{s}}}{\partial z}\right]. \tag{34}$$

Equation 34 assumes that the lateral heat transport and vertical heat transport other than by conduc-
tion are small and can be neglected. Other models include for example the vertical heat advection by
the water penetrating into the soil (e.g. Cox et al., 1999). Equation 34 also assumes that there are no
heat sources inside the soil column. Heat generated by organic matter decomposition is an example
of internally generated heat (e.g. Khvorostyanov et al., 2008). In Eq. 34 $c$ is the volumetric heat
capacity and $\lambda$ is the thermal conductivity of soil/snow. Equation 34 is solved with the ground heat
flux as top boundary condition and zero heat flux at the bottom of the soil column. Eventually the
deep permafrost model of Willeit and Ganopolski (2015) is going to be coupled to PALADYN with
the geothermal heat flux as bottom boundary condition. The numerical solution of Eq. 34 follows
the fully implicit formulation in Oleson et al. (2013). The snow/soil temperature profile is calculated
first without phase change and then readjusted for phase change following Oleson et al. (2013). If





the new temperature of snow or of a soil layer containing frozen water is greater than 0 °C the excess

energy is used to melt snow or frozen soil water. If all snow is melting during one time step and excess energy is remaining, this energy is added to the top soil layer. If soil temperature drops below 0 °C soil water starts to freeze. Observations show that liquid water exists in the soil at temperatures well below 0 °C because of adsorption forces, capillarity and ground heterogeneity (e.g. Williams and Smith, 1989) and the presence of solutes (e.g. Watanabe and Mizoguchi, 2002). To allow liquid

water to coexist with ice below 0 °C, a freezing point depression is included in the model and the maximum liquid water content for soil temperatures $T_s$ below $T_0 = 273.15\,\mathrm{K}$ is formulated as (e.g. Cox et al., 1999; Niu and Yang, 2006; Ekici et al., 2014):

$$w_w^{\max} = \Delta z \rho_w \theta_{\mathrm{sat}} \left[ \frac{L_f(T_s - T_0)}{g T_s \psi_{\mathrm{sat}}} \right]^{-1/b}, \qquad (35)$$

where $\Delta z$ is the layer thickness, $\rho_w$ the density of water, $\theta_{\mathrm{sat}}$ the porosity of the soil, $L_f$ the la-

355 tent heat of fusion of water, $\psi_{\mathrm{sat}}$ is the matric potential at saturation and $b$ the Clapp-Hornberger parameter (Section 5.4).

### 4.1 Snow and soil thermal properties

Snow plays a crucial role in insulating the ground below from the cold air temperatures. A realistic parameterisation of snow thermal properties is therefore fundamental to simulate frozen soil dynam-

360 ics. In particular, PALADYN is very sensitive to the parameterisation of snow thermal conductivity. Hence, several snow thermal conductivity formulations that are all dependent on snow density are included in the model (Yen, 1981; Jordan, 1991; Riche and Schneebeli, 2013). The default snow thermal conductivity is from (Riche and Schneebeli, 2013):

$$\lambda_{\mathrm{sn}} = \lambda_a - 1.06 \times 10^{-5} \rho_{\mathrm{sn}} + 3 \times 10^{-6} \rho_{\mathrm{sn}}^2. \qquad (36)$$

$\lambda_a$ is the air thermal conductivity and the snow density $\rho_{\mathrm{sn}}$ is described in detail in Section 5.2.

The volumetric heat capacity of snow depends on snow density and specific heat capacity of ice ($C_i$):

$$c_{\mathrm{sn}} = C_i \rho_{\mathrm{sn}}. \qquad (37)$$

Soil heat capacity is a volume weighted mean of dry soil and liquid and frozen water:

$$c = (1 - \theta_{\mathrm{sat}}) c_s + \theta_w \rho_w C_w + \theta_i \rho_i C_i, \qquad (38)$$

where $c_s$ is the volumetric heat capacity of dry soil (Table 3), $\theta_w$ and $\theta_i$ are the volumetric soil liquid and frozen water contents, respectively, $C_w$ ist the specific heat capacity of water and $\rho_i$ is the density of ice.

Soil heat conductivity is a combination of heat conductivity of water, ice and dry soil following

Farouki (1981):

$$\lambda = K \lambda_{\mathrm{sat}} + (1 - K) \lambda_{\mathrm{dry}}, \qquad (39)$$



with

$$\lambda_{\text{sat}} = \lambda_{\text{s}}^{1-\theta_{\text{sat}}} \lambda_{\text{w}}^{\frac{\theta_{\text{w}}}{\theta}\theta_{\text{sat}}} \lambda_{\text{i}}^{\frac{\theta_{\text{i}}}{\theta}\theta_{\text{sat}}}, \tag{40}$$

where $\theta$ is the total (liquid plus frozen) volumetric soil water content. The original logarithmic formulation of the Kersten number ($K$) is approximated by a linear function of relative soil moisture:

$$K = \begin{cases} \frac{1}{1-0.35}\left(\frac{\theta}{\theta_{\text{sat}}}-0.35\right) & T_{\text{s}} \geq 0\,^{\circ}C \\ \frac{\theta}{\theta_{\text{sat}}} & T_{\text{s}} < 0\,^{\circ}C \end{cases}. \tag{41}$$

$K$ is limited to be between 0 and 1. $\lambda_{\text{w}}$ and $\lambda_{\text{i}}$ are the thermal conductivities of water and ice, respectively. $\lambda_{\text{s}}$ and $\lambda_{\text{dry}}$ are globally uniform soil parameters (Table 3). Alternatively the values can be chosen to be dependent on soil texture and soil organic carbon content following as described in Appendix B. The inclusion of variable $\lambda_{\text{s}}$ and $\lambda_{\text{dry}}$ does not fundamentally affect the main model results, hence for computational efficiency the parameters are taken to be uniform in space and constant in time by default.

## 5 Hydrology

### 5.1 Canopy water

Re-evaporation of canopy intercepted water contributes significantly to the total water flux from the surface to the atmosphere (e.g. Dirmeyer et al., 2006). Therefore, PALADYN includes a representation of rain and snow intercepted by vegetation. Rain is assumed to be intercepted only by trees while snow is intercepted by all PFTs. The prognostic equations for canopy liquid water ($w_{\text{can}}^{\text{w}}$) and snow ($w_{\text{can}}^{\text{s}}$) are similar and written in terms of canopy interception, canopy evaporation/sublimation and a canopy water removal term as:

$$\frac{dw_{\text{can}}^{\text{w/s}}}{dt} = I_{\text{can}}^{\text{w/s}} - E_{\text{can}}^{\text{w/s}} - \frac{w_{\text{can}}^{\text{w/s}}}{\tau_{\text{w/s}}}. \tag{42}$$

Canopy interception and evaporation are given by:

$$I_{\text{can}}^{\text{w/s}} = \alpha_{\text{int}}^{\text{w/s}} P_{\text{r/s}} \left(1 - \exp\left[-k_{\text{ext}}(L_{\text{ai}} + S_{\text{ai}})\right]\right), \tag{43}$$

$$E_{\text{can}}^{\text{w/s}} = \frac{\rho_{\text{a}}}{r_{\text{a}}}(q_{\text{sat}}(T_{\star}) - q_{\text{a}})f_{\text{can}}^{\text{w/s}}. \tag{44}$$

$P_{\text{r}}$ is the rain rate and $P_{\text{s}}$ the snowfall rate. $\alpha_{\text{int}}^{\text{w}}$ and $\alpha_{\text{int}}^{\text{s}}$ are interception factors (Table 2). The wet canopy fraction $f_{\text{can}}^{\text{w}}$ and the snow covered canopy fraction $f_{\text{can}}^{\text{s}}$ are assumed to increase linearly with $w_{\text{can}}^{\text{w}}$ and $w_{\text{can}}^{\text{s}}$, respectively, up to a maximum water and snow amount that the canopy can hold, $w_{\text{can}}^{\text{max}} = 0.2(L_{\text{ai}} + S_{\text{ai}})$ (e.g Verseghy et al., 1993). $\tau_{\text{w}}$ and $\tau_{\text{s}}$ are the water and snow canopy removal time scales, respectively (Table 2). Negative canopy evaporation, that is dew deposition, is



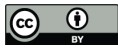

inhibited. If skin temperature is greater than $0\,°C$, all snow is removed from the canopy and added to the snow layer on the ground. Finally $E_{\text{can}} = E_{\text{can}}^{\text{w}} + E_{\text{can}}^{\text{s}}$ is diagnosed and used in the solution of the surface energy balance equation (Eq. (1)). The rate of rain and snow reaching the ground is then derived as:

$$P_{\text{r/s,g}} = P_{\text{r/s}} - E_{\text{can}}^{\text{w/s}} - \frac{dw_{\text{can}}^{\text{w/s}}}{dt}. \tag{45}$$

The area weighted $P_{\text{r/s,g}}$ over the vegetated and bare soil surface tiles are then used as input to the surface hydrology module.

### 5.2 Snow

The snow water equivalent evolution of the single snow layer is determined by the snowfall rate $P_{\text{s,g}}$,
the snowmelt rate $M_{\text{s}}$ and sublimation $E_{\text{s}}$:

$$\frac{dw_{\text{sn}}}{dt} = P_{\text{s,g}} - M_{\text{s}} - E_{\text{s}}. \tag{46}$$

To prevent an indefinite accumulation of snow, $w_{\text{sn}}$ is limited to be below $w_{\text{sn,crit}} = 1000\,\text{kg}\,\text{m}^{-2}$ and the snow excess is added to 'frozen water runoff'.

The density of snow is important because it determines the thickness of snow and hence influences
surface albedo and surface roughness and because it controls the thermal properties of snow (Section 4.1). The parameterisation of snow density is based partly on Anderson (1976); Pitman et al. (1991). The density of freshly fallen snow is temperature dependent following Anderson (1976):

$$\rho_{\text{sn,fresh}} = \rho_{\text{sn,min}} + 1.7\,(T_{\text{a}} - T_0 + 15)^{1.5} \qquad T_0 - 15 < T_{\text{a}} < T_0 + 2. \tag{47}$$

$\rho_{\text{sn,min}}$ is the minimum snow density (Table 2). The effect of self-loading on snow compaction is
425 taken into account using the relation proposed by Kojima (1967) as implemented in Pitman et al. (1991) and the prognostic equation for snow density accounting also for the density of freshly fallen snow is written as (Pitman et al., 1991):

$$\frac{d\rho_{\text{sn}}}{dt} = \frac{0.5g\rho_{\text{sn}}w_{\text{sn}}}{\eta} + P_{\text{s,g}}\frac{\rho_{\text{sn,fresh}} - \rho_{\text{sn}}}{w_{\text{sn}}}, \tag{48}$$

where $\eta$ is the viscosity depending both on the load and temperature:

$$\eta = \eta_0 \exp\left[k_{\text{T}}(T_0 - T_{\text{s}}) + k_\rho \rho_{\text{sn}}\right]. \tag{49}$$

The values of the parameters $\eta_0$, $k_{\text{T}}$ and $k_\rho$ are given in Table 2. The effects of snow metamorphism and snow melting on snow density are neglected. Snow thickness is then computed as:

$$h_{\text{sn}} = \frac{w_{\text{sn}}}{\rho_{\text{sn}}}. \tag{50}$$



### 5.3 Surface runoff and infiltration

Subgrid scale surface hydrology is represented using a TOPMODEL approach (Beven and Kirkby, 1979) as implemented in Niu et al. (2005). The fraction of a grid cell that is assumed to be at saturation, $f_{\text{sat}}$, is determined by the grid cell mean water table position ($z_\nabla$) and the spatially varying maximum saturated fraction $f_{\text{sat}}^{\text{max}}$ computed by Stocker et al. (2014) from the compound topographic index (CTI) derived from the high resolution ETOPO1 topography as (Niu et al., 2005):

$$f_{\text{sat}} = f_{\text{sat}}^{\text{max}} e^{-f_\nabla z_\nabla}. \tag{51}$$

$f_\nabla$ is a parameter whose value is given in Table 2. If the surface is snowfree the wetland fraction is set equal to the saturated fraction ($f_{\text{wet}} = f_{\text{sat}}$), while it is set to zero otherwise. The grid cell mean water table depth is estimated directly from the volumetric water content in the soil column, the peat fraction ($f_{\text{peat}}$) and the water table in peat ($z_\nabla^{\text{peat}}$) as:

$$z_\nabla = (1 - f_{\text{peat}}) \left( h_{\text{soil}} - \sum_l \frac{\theta_l}{\theta_{\text{sat},l}} \Delta z_l \right) + f_{\text{peat}} z_\nabla^{\text{peat}}. \tag{52}$$

$h_{\text{soil}}$ is the soil column depth and the sum is over all soil layers. The peat water table is assumed to follow the grid cell mean seasonal water table variations but with an amplitude limited to the acrotelm thickness (Section 6.3.1). The maximum soil infiltration rate is then computed from the

450 saturated hydraulic conductivity of the top soil layer ($k_{\text{sat},1}$) assuming that infiltration can occur only in the unsaturated part of the grid cell:

$$q_{\text{inf}}^{\text{max}} = k_{\text{sat},1}(1 - f_{\text{sat}}). \tag{53}$$

Surface runoff is then calculated assuming that all liquid water that reaches the surface is rooted directly to runoff over the saturated fraction of the grid cell and considering that the maximum

infiltration rate can not be exceeded:

$$R_{\text{w}} = f_{\text{sat}}(P_{\text{r,g}} + M_{\text{s}}) + (1 - f_{\text{sat}}) \cdot \max\left(0, P_{\text{r,g}} + M_{\text{s}} - q_{\text{inf}}^{\text{max}}\right). \tag{54}$$

The actual soil infiltration rate is then computed as:

$$q_{\text{inf}} = P_{\text{r,g}} + M_{\text{s}} - R_{\text{w}}. \tag{55}$$

### 5.4 Soil hydrology

Water in the soil is assumed to be limited to flow in the vertical direction. Making use of the conservation of mass, the change in volumetric water content over time is then given by the vertical divergence of the water flux and a sink term from soil water extraction by evapotranspiration ($e$):

$$\rho_{\text{w}} \Delta z_l \frac{d\theta_{\text{w},l}}{dt} = q_{l-1} - q_l - e_l, \tag{56}$$





where $l$ is the soil layer index. This equation is solved with infiltration ($q_{\mathrm{inf}}$) as top boundary con-
dition and a free drainage bottom boundary condition, i.e. the water flux at the bottom of the soil
column ($q_{\mathrm{drain}}$) is set equal to the bottom hydraulic conductivity. The soil water flux $q$ is expressed
by Darcy's law:

$$q = k \frac{\partial(\psi - z)}{\partial z}, \tag{57}$$

where $k$ is the hydraulic conductivity and $\psi$ is the matric potential. $z$ is the vertical coordinate and
is positive downwards from the surface. The numerical solution of Eq. (56) follows the formulation
in Oleson et al. (2013).

The hydraulic conductivity and the matric potential are soil hydraulic properties dependent on soil
texture and volumetric soil water following Clapp and Hornberger (1978):

$$\psi = \psi_{\mathrm{sat}} \left( \frac{\theta_{\mathrm{w}}}{\theta_{\mathrm{sat}}} \right)^{-b} \tag{58}$$

$$k = k_{\mathrm{sat}} \left( \frac{\theta_{\mathrm{w}}}{\theta_{\mathrm{sat}}} \right)^{2b+3}. \tag{59}$$

Similarly to the discussion on soil thermal parameters in Section 4.1, hydraulic conductivity and
matric potential at saturation, $k_{\mathrm{sat}}$ and $\psi_{\mathrm{sat}}$, and the Clapp and Hornberger parameter $b$ are set to
global uniform values by default (Table 3). However, a soil texture and soil organic matter content
dependent formulation of $k_{\mathrm{sat}}$, $\psi_{\mathrm{sat}}$ and $b$ is also available (Appendix B).

## 6 Biogeochemistry and vegetation dynamics

### 6.1 Photosynthesis

Daily photosynthesis is modelled following the general ligh use efficiency model described in Hax-
eltine and Prentice (1996a, b) as implemented in the LPJ dynamic vegetation model (Sitch et al.,
2003), with some modifications. Compared to other models it has the advantage that it computes
daily integrated photosynthesis without the need to explicitly resolve the diurnal cycle and therefore
saves computing time. It also makes it convenient to be coupled to the physical PALADYN compo-
nents, which are also integrated with a daily time step. Daily gross photosynthesis is computed from
a light limited ($J_{\mathrm{E}}$) and a Rubisco limited rate ($J_{\mathrm{C}}$) as:

$$A_{\mathrm{g}} = \frac{J_{\mathrm{E}} + J_{\mathrm{C}} - \sqrt{(J_{\mathrm{E}} + J_{\mathrm{C}})^2 - 4\theta_{\mathrm{r}} J_{\mathrm{E}} J_{\mathrm{C}}}}{2\theta_{\mathrm{r}}} \beta_{\theta}. \tag{60}$$

$$J_{\mathrm{E}} = c_1 \cdot APAR, \tag{61}$$

$$J_{\mathrm{C}} = c_2 \cdot V_{\mathrm{m}}. \tag{62}$$

$\theta_{\mathrm{r}}$ is a shape parameter and $APAR$ is the absorbed photosynthetically active radiation computed as:

$$APAR = 0.5 SW^{\downarrow} \alpha_{\mathrm{a}} \left( 1 - e^{-k_{\mathrm{ext}} L_{\mathrm{ai}}} \right) \Delta t (1 - \alpha_{\mathrm{leaf}}) c_{\mathrm{q}}. \tag{63}$$



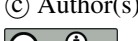

Half of the downwelling shortwave radiation is assumed to be in the photosynthetically active wavelength range, $\alpha_{\mathrm{a}}$ accounts for reductions in PAR utilisation efficiencies in natural ecosystems, the factor $1 - e^{-k_{\mathrm{ext}} L_{\mathrm{ai}}}$ scales to the canopy, $\alpha_{\mathrm{leaf}}$ is the leaf albedo in the PAR range, $\Delta t$ is the length of day in seconds and $c_{\mathrm{q}}$ is a conversion factor from $\mathrm{J\,m^{-2}}$ to $\mathrm{mol\,m^{-2}}$. Parameter values are given in Table 4 and more details on the formulation of $c_1$ and $c_2$ and the maximum daily rate of net photosynthesis $V_{\mathrm{m}}$ are given in Appendix C.

Leaf respiration, $R_{\mathrm{d}}$, is scaled to $V_{\mathrm{m}}$ as:

$$R_{\mathrm{d}} = a_{\mathrm{C3/4}} V_{\mathrm{m}} \beta_{\theta}, \tag{64}$$

and daily net assimilation is then calculated as:

$$A_{\mathrm{n}} = A_{\mathrm{g}} - R_{\mathrm{d}}. \tag{65}$$

Daytime net assimilation can then be computed by adding nighttime respiration:

$$A_{\mathrm{nd}} = A_{\mathrm{n}} + \left(1 - \frac{d_{\mathrm{h}}}{24}\right) R_{\mathrm{d}}. \tag{66}$$

$\beta_{\theta}$ is a soil moisture limiting factor:

$$\beta_{\theta} = \sum_{l} \frac{\theta_{\mathrm{w},l} - \theta_{\mathrm{wp}}}{\theta_{\mathrm{fc}} - \theta_{\mathrm{wp}}} r_l, \tag{67}$$

$\theta_{\mathrm{wp}}$ and $\theta_{\mathrm{fc}}$ are the soil moisture values at wilting point and field capacity, respectively. $r_l$ is the fraction of roots in layer $l$ (section 6.2.2). If the soil temperature of layer $l$ is below -2 °C the corresponding term in Eq. (67) is set to 0.

$c_1$ and $c_2$ depend on the intercellular partial pressure of $CO_2$ ($p_{\mathrm{i}}$), which is proportional to the atmospheric $CO_2$ concentration ($p_{\mathrm{a}}$):

$$p_{\mathrm{i}} = \lambda_{\mathrm{c}} p_{\mathrm{a}}. \tag{68}$$

In LPJ $\lambda_{\mathrm{c}}$ is computed iteratively from potential and actual evapotranspiration (Sitch et al., 2003). To reduce the computation cost and in light of recent developments, in PALADYN $\lambda_{\mathrm{c}}$ is derived from the optimal stomatal conductance model (Medlyn et al., 2011), which predicts that stomatal conductance for water vapour $g_{\mathrm{can}}$ is given by:

$$g_{\mathrm{can}} = g_{\mathrm{min}} + \left(1 + \frac{g_1}{\sqrt{VPD}}\right) \frac{A_{\mathrm{nd}}}{c_{\mathrm{a}}}. \tag{69}$$

$VPD$ is the vapor pressure deficit between leaf surface and ambient air. Since $CO_2$ has to diffuse trough the stomata into the leaf interior before being fixed by photosynthesis and at the same time water vapour diffuses through the stomata from the leaf interior to the canopy air, $g_{\mathrm{can}}$ and $A_{\mathrm{nd}}$ are also related by:

$$g_{\mathrm{can}} = g_{\mathrm{min}} + 1.6 \frac{A_{\mathrm{nd}}}{c_{\mathrm{a}} - c_{\mathrm{i}}}. \tag{70}$$



$g_{\min}$ is a minimum stomatal conductance (Table 5) and $c_a$ and $c_i$ are the atmospheric and intercellular $CO_2$ mole fractions. From Eqs. (69) and (70) $\lambda_c$ can simply be derived (e.g. Medlyn et al., 2011):

$$\lambda_c = 1 - \frac{1.6}{1 + g_1/\sqrt{VPD}}. \tag{71}$$

To a first approximation, the values of $g_1$ are taken to be constant PFT specific parameters (Table 5) based on the data reported in Lin et al. (2015). As will be shown in section 6.5, based on a simple model, the ratio of $c_i$ and $c_a$ is also the main parameter determining the carbon isotopic discrimination during photosynthesis. Therefore the PFT specific discrimination is used as an additional constraint on $g_1$ values.

   Finally, maintenance respiration and growth respiration are computed and net primary production

($NPP$) is derived as in Sitch et al. (2003).

## 6.2 Vegetation dynamics

There are a number of existing dynamic global vegetation models spanning a large range of different approaches of varying complexity. The appropriate model complexity for PALADYN, balancing low computational expenses and a realistic representation of continental-scale vegetation dynam-

ics, is represented by the top-down modelling approach of the TRIFFID dynamic global vegetation model (Cox, 2001; Clark et al., 2011). Another main advantage of this type of model is that it does not require interannual climate variability, which can not be provided by a statistical-dynamical atmosphere model like CLIMBER. The PALADYN dynamic vegetation scheme is therefore based on TRIFFID. The model distinguishes five plant functional types: broadleaved trees, needleleaved

trees, C3 and C4 grass and shrubs. Vegetation carbon $C_v$ and fractional area coverage $\nu$ of each PFT $i$ are described by a coupled system of first order differential equations based on the Lotka-Volterra approach for modelling competition between species:

$$\frac{dC_{v,i}}{dt} = (1 - \lambda_{NPP,i})NPP_i - \Lambda_{loc,i}, \tag{72}$$

$$\frac{d\nu_i}{dt} = \frac{\lambda_{NPP,i}NPP_i}{C_{v,i}}\nu_{i,\star}\left(1 - \sum_j c_{ij}\nu_j\right) - \gamma_{v,i}\nu_{i,\star}. \tag{73}$$

$\nu_{i,\star} = \max(\nu_i, \nu_{seed})$, where $\nu_{seed}$ is a small seeding fraction used to ensure that a PFT is always seeded (Table 6). $\lambda_{NPP}$ is a factor determining the partitioning of $NPP$ between increase of vegetation carbon of the existing vegetated area (Eq. (72)) and spreading of the given PFT (Eq. (73)) and is given by:

$$\lambda_{NPP} = \begin{cases} 0 & L_{ai,b} < L_{ai}^{\min} \\ \frac{L_{ai,b} - L_{ai}^{\min}}{L_{ai}^{\max} - L_{ai}^{\min}} & L_{ai}^{\min} \leq L_{ai,b} \leq L_{ai}^{\max} \\ 1 & L_{ai,b} > L_{ai}^{\max}. \end{cases} \tag{74}$$





$L_{\mathrm{ai,b}}$ is the 'balanced' leaf area index that would be reached if the plant was in full leaf and $L_{\mathrm{ai}}^{\min}$ and $L_{\mathrm{ai}}^{\max}$ are PFT specific parameters (Table 5). $\Lambda_{\mathrm{loc}}$ is the local litterfall rate:

$$\Lambda_{\mathrm{loc}} = \Lambda_{\mathrm{l}} + \gamma_{\mathrm{r}} C_{\mathrm{v,r}} + \gamma_{\mathrm{s}} C_{\mathrm{v,s}}. \tag{75}$$

Litterfall from leaf turnover is given by $\Lambda_{\mathrm{l}} = \gamma_{\mathrm{l}} C_{\mathrm{v,l}}$ for evergreen plants and is computed from the phenological status (section 6.2.1) for deciduous plants. The $\gamma$s are PFT dependent turnover rates of leaf, root and stem carbon (Table 5). Vegetation carbon $C_{\mathrm{v}}$ is directly related to the balanced leaf area index through the relations of leaf ($C_{\mathrm{v,l}}$), root ($C_{\mathrm{v,r}}$) and stem ($C_{\mathrm{v,s}}$) carbon to $L_{\mathrm{ai,b}}$:

$$C_{\mathrm{v,l}} = \frac{L_{\mathrm{ai,b}}}{SLA}, \tag{76}$$

$$C_{\mathrm{v,r}} = \mathcal{L}, \tag{77}$$

$$C_{\mathrm{v,s}} = a_{\mathrm{wl}} L_{\mathrm{ai,b}}^{5/3}. \tag{78}$$

$$C_{\mathrm{v}} = C_{\mathrm{v,l}} + C_{\mathrm{v,r}} + C_{\mathrm{v,s}}. \tag{79}$$

$SLA$ is the specific leaf area and is PFT dependent following Kattge et al. (2011) (Table 5). $a_{\mathrm{wl}}$ is a PFT specific allometric coefficient (Table 5). The competition coefficients, $c_{ij}$, represent the impact of vegetation type $j$ on the vegetation type of interest $i$. TRIFFID is based on a tree-shrub-grass dominance hierarchy with dominant types $i$ limiting the expansion of sub-dominant types $j$ ($c_{ji} = 1$), but not vice-versa ($c_{ij} = 0$). While in TRIFFID the tree types and grass types co-compete with competition coefficients dependent on their relative heights, in PALADYN they compete only based on their $NPP$ ($c_{ij} = 0.5$ and $c_{ji} = 0.5$). Addtitionally, in PALADYN we implemented a dependence of the competition coefficients on bioclimatic limits, i.e. the coldest month temperature ($T_{\mathrm{cmon}}^{\min/\max}$) and growing degree days ($gdd_{\min}$) as given in Table 5. In a given grid cell, PFTs outside of the bioclimatic limits are not competitive and will be dominated by other PFTs, regardless of the tree-shrub-grass dominance.

The last term in Eq. (73) represents vegetation disturbance. In TRIFFID, the disturbance rate $\gamma_{\mathrm{v}}$ is taken to be a constant PFT specific parameter. In reality, on a global scale, disturbance is mainly caused by fire, which shows a strong dependence on climatic conditions and fuel availability (e.g. Thonicke et al., 2010). We therefore introduce a simple parameterisation for fire disturbance based on top-soil moisture and aboveground biomass loosely following Reick et al. (2013) and Arora and Boer (2005):

$$\gamma_{\mathrm{v}} = \gamma_{\mathrm{v,min}} + \frac{1}{\tau_{\mathrm{fire}}} \max\left(0, \frac{\theta_{\mathrm{crit}} - \theta_1}{\theta_{\mathrm{crit}}}\right) \max\left(0, \min\left(1, \frac{C_{\mathrm{v,ag}} - C_{\mathrm{v,low}}}{C_{\mathrm{v,high}} - C_{\mathrm{v,low}}}\right)\right). \tag{80}$$

$\gamma_{\mathrm{v,min}}$ is a minimum constant disturbance rate intended to represent disturbances other than fire (e.g. windthrow (Reick et al., 2013) and insect outbreaks, among others (e.g. Dale et al., 2001)) (Table 6). $\tau_{\mathrm{fire}}$ is a characteristic fire return time scale, $\theta_{\mathrm{crit}}$ is the critical soil moisture below which fires can occur and $C_{\mathrm{v,low}}$ and $C_{\mathrm{v,high}}$ are values of aboveground biomass ($C_{\mathrm{v,ag}}$) that define the fuel availability limitation function. All parameter values are listed in Table 6.





Vegetation height is computed from stem carbon following Cox (2001):

$$590 \quad h_{\mathrm{v}} = \frac{C_{\mathrm{v,s}}}{0.01 \cdot a_{\mathrm{ws}}} \left( \frac{a_{\mathrm{wl}}}{C_{\mathrm{v,s}}} \right)^{3/5}. \tag{81}$$

$a_{\mathrm{ws}}$ is the ratio of total to respiring stem carbon (Table 5). The stem area index, $S_{\mathrm{ai}}$, is taken to be 1/10 of $L_{\mathrm{ai,b}}$.

The dynamic vegetation model has a monthly time step.

### 6.2.1 Phenology

The phenology of the PFTs is controlled by the coldest month temperature following Sitch et al. (2003). If the coldest month temperature falls below a PFT specific value $T_{\mathrm{cmon}}^{\mathrm{phen}}$ (Table 5), then the PFT in the gridcell is assumed to be deciduous and $L_{\mathrm{ai}}$ is computed from:

$$L_{\mathrm{ai}} = \phi L_{\mathrm{ai,b}}. \tag{82}$$

$\phi$ increases linearly with the growing degree days ($gdd$) above a PFT specific base temperature $T_{\mathrm{base}}^{\mathrm{gdd}}$, at a PFT specific rate determined by $gdd_{\mathrm{crit}}$:

$$\phi = \frac{gdd}{gdd_{\mathrm{crit}}}. \tag{83}$$

After $\phi$ reaches its maximum value of 1 it remains constant until the air temperature drops below $T_{\mathrm{base}}^{\mathrm{gdd}}$. Then leaf senesence starts when the temperature drops below $T_{\mathrm{base}}^{\mathrm{gdd}}$ and continues until all leaves are lost to litter at 5 °C below $T_{\mathrm{base}}^{\mathrm{gdd}}$. Raingreen phenology is not represented in the model. Needleleaf trees are assumed to always be evergreen, because given the low PFT specific leaf area (Table 5), they would not be competitive in very cold regions if they were decidous.

### 6.2.2 Root distribution

The vertical distribution of roots in the soil plays an important role in land surface models. It determines the water that is accessible to the plants and hence controls the exchange of water between the surface and the atmosphere. Water availability affects also plant productivity and consequently plays an important role in the competition between plant functional types. It also controls the vertical distribution of root litter input to the soil which is an important factor determining vertical soil carbon distribution. In PALADYN we adopt the root distribution scheme proposed by Zeng (2001). The root fraction in each soil layer ($r_l$) is derived from the cumulative root fraction:

$$615 \quad r(z) = 1 - 0.5 \left( e^{-d_{\mathrm{r},1} z} + e^{-d_{\mathrm{r},2} z} \right). \tag{84}$$

$d_{\mathrm{r},1}$ and $d_{\mathrm{r},2}$ are PFT specific paramters (Table 5).

### 6.3 Soil carbon

Traditionally, in terrestrial biosphere models soil carbon has been represented in terms of vertically-integrated pools. Only recently vertically discretized soil carbon has started to be included in these



models (e.g. Koven et al., 2009; Schaphoff et al., 2013). Vertically integrated models are unable to represent soil carbon dynamics in permafrost areas, where only part of the carbon stored in the soil column is affected by the seasonal thawing of the upper soil. Large quantities of carbon are stored in the permanently frozen soils around the Arctic (Tarnocai et al., 2009; Hugelius et al., 2014; Schuur et al., 2015) and to model the dynamics of this carbon stock it is necessary to include carbon

separately in different soil layers. A proper representation of the permafrost carbon pool is important especially for carbon cycle modelling over long time scales.

Therefore, PALADYN has carbon distributed over the different soil layers where temperature and soil water are also computed. Additionally, each grid cell distinguishes between soil carbon in four different soil columns: mineral soil carbon and peat carbon below the vegetated surface tile, buried

carbon below ice sheets and shelf carbon below the water on the ocean shelf (Fig. 1c). Each layer generally contains three carbon pools with different decomposability (Fig. 3). For unfrozen mineral soil carbon the three pools are organized into litter, fast and slow carbon following (Sitch et al., 2003). This structure is modified for peatlands, perennialy frozen soils and buried carbon.

The generic prognostic equations for litter, fast and slow soil carbon pools are written as:

$$\frac{\partial C_{\mathrm{lit}}(z)}{\partial t} = \Lambda(z) - k_{\mathrm{lit}}(z)C_{\mathrm{lit}}(z) + \frac{\partial}{\partial z}\left(D(z)\frac{\partial C_{\mathrm{lit}}}{\partial z}\right) \tag{85}$$

$$\frac{\partial C_{\mathrm{fast}}(z)}{\partial t} = (1 - f_{\mathrm{lit}}^{\mathrm{resp}})f_{\mathrm{lit}\to\mathrm{fast}}k_{\mathrm{lit}}(z)C_{\mathrm{lit}}(z) - k_{\mathrm{fast}}(z)C_{\mathrm{fast}}(z) + \frac{\partial}{\partial z}\left(D(z)\frac{\partial C_{\mathrm{fast}}}{\partial z}\right) \tag{86}$$

$$\frac{\partial C_{\mathrm{slow}}(z)}{\partial t} = (1 - f_{\mathrm{lit}}^{\mathrm{resp}})f_{\mathrm{lit}\to\mathrm{slow}}k_{\mathrm{lit}}(z)C_{\mathrm{lit}}(z) - k_{\mathrm{slow}}(z)C_{\mathrm{slow}}(z) + \frac{\partial}{\partial z}\left(D(z)\frac{\partial C_{\mathrm{slow}}}{\partial z}\right). \tag{87}$$

Litter carbon is increased by litterfall $\Lambda(z)$. A fraction $f_{\mathrm{lit}}^{\mathrm{resp}}$ of the decomposed litter carbon goes directly into the atmosphere, while the rest goes partly into the fast carbon pool ($f_{\mathrm{lit}\to\mathrm{fast}}$) and partly

into the slow carbon pool ($f_{\mathrm{lit}\to\mathrm{slow}}$) (Table 7). Each carbon pool decomposes at a specific rate $k$, which depends on soil temperature and soil moisture. The vertical redistribution of soil carbon between soil layers is represented as a diffusive process with diffusivity $D(z)$.

Over the vegetated grid cell part, the local litter, the litter originating from competition between the PFTs and the litter from large scale disturbances are aggregated to give an average litterfall

($\Lambda_{\mathrm{veg}}(z)$) as in (Cox, 2001). Litter from the roots is added to the different soil layers depending on the root fraction in each layer, while litter from leaves and stem is added to the top soil layer.

When ice sheets are expanding into vegetated areas, a fraction $f_{\mathrm{veg}\to\mathrm{bur}}$ of the vegetation carbon is assumed to be directly buried below the ice and the remaining is added to the litter pools of the vegetated part:

$$\Lambda_{\mathrm{bur}}(z) = f_{\mathrm{veg}\to\mathrm{bur}}\overline{C_{\mathrm{v}}}(z)\Delta f_{\mathrm{ice}} \tag{88}$$

$$\Lambda_{\mathrm{veg}}(z) = (1 - f_{\mathrm{veg}\to\mathrm{bur}})\overline{C_{\mathrm{v}}}(z)\Delta f_{\mathrm{ice}}. \tag{89}$$





$\overline{C_\mathrm{v}}(z)$ is the mean vegetation carbon content of the vegetated grid cell part in the different soil layers. For the purpose of litter computation the aboveground vegetation carbon is considered to be part of the top soil layer. $\Delta f_\mathrm{ice}$ is the increase of ice sheet fraction in the grid cell.

When sea level is rising and shelf areas become flooded, the flooded vegetation is assumed to die instantaneously and vegetation carbon is added directly to the shelf litter pool:

$$\Lambda_\mathrm{shelf}(z) = \overline{C_\mathrm{v}}(z)\Delta f_\mathrm{shelf}, \tag{90}$$

where $\Delta f_\mathrm{shelf}$ is the increase in shelf fraction.

Vertical carbon diffusivity in unfrozen mineral soils is assumed to be determined by bioturbation

and $D(z) = D_\mathrm{bio}$ following Braakhekke et al. (2011) (Table 7). In permafrost areas the diffusivity represents cryoturbation in the active layer. $D(z)$ is set to a constant value in the active layer and is assumed to linearly decrease below it to a value of zero at a multiple ($n_\mathrm{al}$) of the active layer thickness $z_\mathrm{al}$ (Koven et al., 2009):

$$D(z) = \begin{cases} D_\mathrm{cryo} & z \le z_\mathrm{al} \\ D_\mathrm{cryo}\left(1 - \frac{z-z_\mathrm{al}}{(n_\mathrm{al}-1)z_\mathrm{al}}\right) & z_\mathrm{al} < z \le n_\mathrm{al}z_\mathrm{al}. \end{cases} \tag{91}$$

The value of $D_\mathrm{cryo}$ is given in Table 7.

The decomposition rates for mineral soil carbon depend on temperature, liquid water content in the soil layers and inundated fraction of the grid cell:

$$k_\mathrm{x}^\mathrm{min}(z) = (1 - f_\mathrm{inun})k_\mathrm{x,10}f_\mathrm{T}(z)f_\theta(z) + f_\mathrm{inun}k_\mathrm{x,10}f_\mathrm{T}(z)f_{\theta,\mathrm{sat}}(z), \tag{92}$$

for $\mathrm{x} = \mathrm{lit, fast, slow}$. The inundated grid cell fraction is the wetland fraction with the peatland frac-

tion removed, $f_\mathrm{inun} = f_\mathrm{wet} - f_\mathrm{peat}$. $k_\mathrm{lit,10}$, $k_\mathrm{fast,10}$ and $k_\mathrm{slow,10}$ are the litter, fast and slow carbon decomposition rates at $10\,^\circ$C and field capacity and are given in Table 7. The temperature dependence follows a modified Arrhenius equation (Lloyd and Taylor, 1994):

$$f_\mathrm{T}(z) = \exp\left[308.56\left(\frac{1}{56.02} - \frac{1}{46.02 + T(z) - T_0}\right)\right]. \tag{93}$$

The soil moisture dependence is taken from Porporato et al. (2003) and gives a linear increase of the

decomposition rate up to field capacity and a hyperbolic decrease above field capacity:

$$f_\theta(z) = \begin{cases} \frac{\theta_\mathrm{w}(z)}{\theta_\mathrm{fc}} & \theta_\mathrm{w}(z) \le \theta_\mathrm{fc} \\ \frac{\theta_\mathrm{fc}}{\theta_\mathrm{w}(z)} & \theta_\mathrm{w}(z) > \theta_\mathrm{fc}. \end{cases} \tag{94}$$

The soil moisture dependence factor for inundated land, $f_{\theta,\mathrm{sat}}$, is simply the value of $f_\theta$ at saturation.

PALADYN allows for the possibility to effectively treat the carbon in frozen soils as inert. If inert permafrost carbon is switched on, the decomposition rates in Eq. 92 are additionally weighted by a

frozen soil factor, $f_\mathrm{inert}$:

$$k_\mathrm{x}^\mathrm{min}(z) = (1 - f_\mathrm{inert}(z))k_\mathrm{x}^\mathrm{min}(z) + f_\mathrm{inert}(z)k_\mathrm{inert}, \tag{95}$$



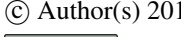

for $x = \mathrm{lit, fast, slow}$. All carbon is assumed to be inert if the fraction of frozen water in a layer exceedes $f_{\mathrm{frz,crit}}$:

$$f_{\mathrm{inert}}(z) = \min\left(1, \frac{1}{f_{\mathrm{frz,crit}}}\frac{\theta_{\mathrm{i}}}{\theta_{\mathrm{i}} + \theta_{\mathrm{w}}}\right). \tag{96}$$

Therefore in soil layers where at least a fraction $f_{\mathrm{frz,crit}}$ of water is frozen all year round, carbon is effectively decomposing at the very low rate $k_{\mathrm{inert}}$.

More details on the parameterisation of carbon dynamics below ice sheets and on the ocean shelf and of permafrost carbon will be given in a future paper dedicated to processes active on glacial-interglacial cycles time scales.

**6.3.1 Peatlands**

Peat carbon is treated slightly differently from the other carbon pools. We follow the approach of Kleinen et al. (2012) and distinguish between a surface litter layer and an acrotelm layer where carbon is decomposed partly under oxic and partly under anoxic conditions, depending on the position of the water table. Both litter and acrotelm are assumed to be contained in the top soil layer. In the

695 layers below, the catotelm, decomposition occurs without oxygen all year round. The prognostic equations for peat litter, acrotelm and catotelm carbon are:

$$\frac{\partial C_{\mathrm{lit}}^{\mathrm{peat}}}{\partial t} = \Lambda_{\mathrm{peat}} - k_{\mathrm{lit}}^{\mathrm{peat}} C_{\mathrm{lit}}^{\mathrm{peat}} \tag{97}$$

$$\frac{\partial C_{\mathrm{acro}}}{\partial t} = (1 - f_{\mathrm{lit}}^{\mathrm{resp}}) k_{\mathrm{lit}}^{\mathrm{peat}} C_{\mathrm{lit}}^{\mathrm{peat}} - k_{\mathrm{acro} \to \mathrm{cato}} C_{\mathrm{acro}} - k_{\mathrm{acro}} C_{\mathrm{acro}} \tag{98}$$

$$\frac{\partial C_{\mathrm{cato}}(z)}{\partial t} = k_{\mathrm{acro} \to \mathrm{cato}} C_{\mathrm{acro}} - k_{\mathrm{cato}}(z) C_{\mathrm{cato}}(z). \tag{99}$$

The transfer from acrotelm to catotelm carbon occurs only once a critical acrotelm carbon content $C_{\mathrm{acro,crit}} = 5\,\mathrm{kgC\,m^{-2}}$ is reached, as suggested by (Wania et al., 2009). Typical acrotelm carbon densities are around $20\,\mathrm{kgC\,m^{-3}}$, so this threshold roughly corresponds to assuming that transfer to the catotelm starts when the actrotelm reaches a thickness of $25\,\mathrm{cm}$, which is a typical value of observed acrotelm thickness. When this threshold is exceeded, acrotelm carbon is transfered to the

catotelm in the second soil layer. Peat is assumed to grow thicker by accumulating carbon on top and therefore in the model the catotelm is shifted to lower soil layers once the catotelm carbon density $C_{\mathrm{cato,crit}}$ has been exceeded in a given layer. For the same reason the vertical diffusivity of peat carbon is set to 0. Litterfall over peatlands is assumed to be the same as over mineral soil, but to be added to the top soil layer only: $\Lambda_{\mathrm{peat}} = \sum_z \Lambda_{\mathrm{veg}}(z)$. The decomposition rates for litter, acrotelm

and catotelm are given by:

$$k_{\mathrm{lit}}^{\mathrm{peat}} = k_{\mathrm{lit},10} f_{\mathrm{T}}(1)(f_{\mathrm{oxic}} + (1 - f_{\mathrm{oxic}}) f_{\theta,\mathrm{peat}}) \tag{100}$$

$$k_{\mathrm{acro}} = k_{\mathrm{acro},10} f_{\mathrm{T}}(1)(f_{\mathrm{oxic}} + (1 - f_{\mathrm{oxic}}) f_{\theta,\mathrm{peat}}) \tag{101}$$

$$k_{\mathrm{cato}}(z) = k_{\mathrm{cato},10} f_{\mathrm{T}}(z) f_{\theta,\mathrm{peat}}. \tag{102}$$





Since peatland soil temperature is not separately computed by the model, the temperature factor is
calculated using the grid cell mean temperature. $f_{\theta,\mathrm{peat}}$ is taken to be equal to 0.3 as in Wania et al.
(2009); Koven et al. (2013). The values of the reference decomposition rates are given in Table 7.
The fraction of litter and acrotelm that is respiring in oxic conditions, $f_{\mathrm{oxic}}$, is determined from the
mean grid cell water table depth $z_\nabla$ and the minimum monthly water table position $z_\nabla^{\min}$ assuming
that the seasonal water table variations in the peatland fraction follow the grid cell mean water table
and that the amplitude of water table variations in peatland is reduced compared to the grid cell mean
and limited to the acrotelm thickness:

$$f_{\mathrm{oxic}} = \frac{\min\left(z_{\mathrm{acro}}, \max\left(0, z_\nabla - z_\nabla^{\min}\right)\right)}{z_{\mathrm{acro}}}. \tag{103}$$

Peatland expansion and contraction is modelled partly following Stocker et al. (2014). The grid
cell fraction that is wetland for at least 3 months of the year is considered to be potential peatland
area $f_{\mathrm{peat}}^{\mathrm{pot}}$. The actual peatland area $f_{\mathrm{peat}}$ is simulated as:

$$f_{\mathrm{peat},n+1} = \begin{cases} \min\left((1+r)f_{\mathrm{peat},n}, f_{\mathrm{peat}}^{\mathrm{pot}}\right) & \frac{\mathrm{d}C_{\mathrm{peat}}}{\mathrm{d}t} \geq \frac{\mathrm{d}C_{\mathrm{peat}}}{\mathrm{d}t}\big|_{\mathrm{crit}} \quad \text{or} \quad C_{\mathrm{peat}} \geq C_{\mathrm{peat}}^{\mathrm{crit}} \\ \max\left((1-r)f_{\mathrm{peat},n}, f_{\mathrm{peat}}^{\min}\right) & \frac{\mathrm{d}C_{\mathrm{peat}}}{\mathrm{d}t} < \frac{\mathrm{d}C_{\mathrm{peat}}}{\mathrm{d}t}\big|_{\mathrm{crit}} \quad \text{and} \quad C_{\mathrm{peat}} < C_{\mathrm{peat}}^{\mathrm{crit}} \end{cases} \tag{104}$$

Peat is expanding if the annual mean rate of carbon uptake $(\mathrm{d}C_{\mathrm{peat}}/\mathrm{d}t)$ is greater than a critical
value $\mathrm{d}C_{\mathrm{peat}}/\mathrm{d}t\big|_{\mathrm{crit}}$ or if peat carbon exceedes a value $C_{\mathrm{peat}}^{\mathrm{crit}}$, otherwise peatland area is shrinking.
To account for inertia in lateral peatland expansion and contraction, the relative areal change rate is
limited to $1\,\%\,\mathrm{yr}^{-1}$ ($r = 0.01\,\mathrm{yr}^{-1}$). When the peat area is changing, carbon is simply redistributed
between mineral soil and peat carbon pools layer-by-layer with the following rules: $C_{\mathrm{lit}}^{\mathrm{peat}} \leftrightarrow C_{\mathrm{lit}}$,
$C_{\mathrm{acro}} \leftrightarrow C_{\mathrm{fast}}$ and $C_{\mathrm{cato}} \leftrightarrow C_{\mathrm{slow}}$.

### 6.4 Methane emissions

Methane emissions are simulated as a constant fraction of heterotrophic respiration when respiration
occurs under anaerobic conditions, as is the case in wetlands, peatlands and flooded ocean shelfs.
The fraction of carbon that is respired as methane, $f_{CH_4:C}$, is different for wetlands, peatlands and
ocean shelf (Table 7).

### 6.5 Carbon isotopes: $^{13}$C and $^{14}$C

The stable carbon isotope $^{13}$C and radiocarbon $^{14}$C are tracked in PALADYN trough all carbon pools
in vegetation and soil. Discrimination is simulated only during photosynthesis and follows the model
outlined in Lloyd and Farquhar (1994). The discrimination factor $\Delta$ for C3 and C4 photosynthesis
is given by:

$$\Delta = \begin{cases} 4.4\frac{c_a - c_i}{c_a} + 27\frac{c_i}{c_a} & \text{C3} \\ 4.4\frac{c_a - c_i}{c_a} + (-5.7 + 20 \cdot 0.35)\frac{c_i}{c_a} & \text{C4}. \end{cases} \tag{105}$$





Radiocarbon decay is ignored in the vegetation carbon pools because of their fast turnover time
relative to the $^{14}$C decay rate. In all soil carbon pools radiocarbon has a half life of 5730 years.

## 7    Model spinup

Some of the processes related to vegetation and soil carbon dynamics have very long intrinsic time
scales and therefore long simulations of at least 10000 years would be required to get the system into
an equilibrium state with prescribed boundary conditions. Even though this is in principle feasible
with PALADYN, it is in fact impractical for test and tuning purposes. Therefore the possibility to
run the vegetation and carbon cycle modules with an artificially high internal integration time step
of 1000 or more years is implemented in PALADYN. This is possible due to the fully implicit
formulation of the model components. In this equilibrium spinup mode the vegetation and carbon
cycle modules are called only at the end of each simulation year but using annually cumulated NPP
and litterfall and annual mean decomposition rates for soil carbon. Using the equilibrium spinup
mode brings the model close to equilibrium already after around 100 years of simulation. 100 years
is also the spinup time required to bring the physical state of the land model, particularly permafrost
related processes, into equilibrium with climate. The equilibrium spinup mode can however not be
applied to processes which are intrinsically out of equilibrium such as peatlands and inert permafrost
carbon. To get the present state of these pools a transient simulation over at least one glacial cycle is
required.

## 8    Evaluation

In this section the performance of PALADYN for present day is presented and discussed. The model
is designed for large scale applications and therefore the model evaluation is done at a global scale,
although in principle it would be possible to run the model in a single column mode forced with
site level observations. For the model evaluation an offline transient simulation from 1901 to 2010
is performed. In offline mode PALADYN needs several monthly climate fields as input as listed in
Table 9. In addition the annual atmospheric $CO_2$ concentration has to be provided. For the historical
simulation of the past century the WATCH climate forcing (Weedon et al., 2011, 2014) based on
ERA-40 (Uppala et al., 2005) and ERA-Interim reanalysis (Dee et al., 2011) combined with GPCC
precipitation (Schneider et al., 2014) is used. $CO_2$ is prescribed from Bereiter et al. (2015) combined
with Manua Loa data (Keeling et al., 1976). Before running the transient experiment, the model is
spun up for 100 years as described in section 7 with the mean 1901-1930 climate as forcing and the
1901 $CO_2$ concentration of 295 ppm. To get a rough estimate of peatland area and carbon, during
this equilibrium spinup phase, the peatland module is enabled to allow peatlands to establish. To
allow the fast carbon processes to equilibrate, the model is subsequently run for additional 400 years
with the dynamic vegetation and soil carbon modules called with a monthly timestep. Finally, the



model is run in transient mode for the historical period forced with annualy varying climate and $CO_2$ concentrations. Peatland area is kept constant during this phase.

Depending on the time interval covered by the different observational data products, the model climatology over the given time period is computed and used to evaluate the different model components, as described next.

## 8.1 Physical processes

The modelled net radiation absorbed at the surface is in good agreement with reanalysis data both
for the seasonal cycle and the annual mean (Fig. 4). With the downwelling shortwave and longwave radiation used as forcing, the net surface radiation is determined by the modelled surface emissivity and albedo. The surface albedo for winter and summer is well simulated in the model (Fig. 5).

A correct partitioning of the absorbed radiation between sensible, latent and ground heat flux is of fundamental importance for a land model. The modelled sensible heat flux compares well with
ERA-Interim reanalysis data except for the tropics, where it is systematically overestimated and for the subtropics, where it is underestimated in the model (Fig. 6). Evapotranspiration, and therefore the latent heat flux, tends to be overestimated by the model everywhere except for the tropics when compared to estimates from Mueller et al. (2013) (Fig. 7). However, it is in good agreement with ERA-Interim. This is to be expected because evapotranspiration strongly depends on surface air
conditions which are used to force the model, which are based on ERA-Interim. The discrepancy between model and (model–based) estimates from (Mueller et al., 2013) might therefore reflect a deficiency in the forcing rather than in the model.

Evapotranspiration is the sum of transpiration from vegetation, surface evaporation and canopy interception and re-evaporation. The partitioning of total evapotranspiration between the different
components is shown in Fig. 8. Transpiration is dominant in the tropics and generally in densely vegetated areas. A significant amount of precipitation is directy re-evaporated back to the atmosphere from plant canopies, particulary in the tropics. Surface evaporation is the only process acting in desert regions. Globally, transpiration, surface evaporation and canopy evaporation account for around 50, 30 and 20 % of total evapotranspiration, respectively. This compares favorably with
Dirmeyer et al. (2006), who estimated total global evapotranspiration to be partitioned in 48% from transpiration, 36% from soil evaporation, and 16% from canopy interception and re-evaporation using an ensemble of land surface models.

As a consequence of the overestimation of evapotranspiration, simulated runoff is underestimated, particulary over northern hemisphere mid-latitudes (Fig. 9). Compared to data from Fekete et al.
(2002) the modelled NH runoff from melting snow in spring tends to be less concentrated to May and June and more gradually distributed over the whole summer season. Global annual runoff is $35\,\mathrm{Pg\,yr^{-1}}$.





Modelled DJF and JJA soil moisture shows generally a good agreement with estimates from satellite data (Liu et al., 2012; Wagner et al., 2012) in the tropics, while the model tends to simulate a dryer top soil in high northern latitudes (Fig. 10). However, it has to be mentioned that the satellite data are representative for the soil moisture of the top few cm of soil, while the top model soil layer is 20 cm thick.

The mean annual simulated wetland area is $3.2\,\mathrm{mln\,km^2}$. Maximum monthly wetland extent is generally well captured by the model, particulary over high northern latitudes (Fig. 11). Compared to the multi-satellite product from GIEMS (Prigent et al., 2007; Papa et al., 2010) the model simulates larger wetland areas in tropical forest areas. The modelled seasonal variation in global wetland area is in very good agreement with GIEMS (Fig. 12).

The NH spring evolution of snow mass is compared to the GlobSnow dataset (Takala et al., 2011; Luojus et al., 2013) in Fig. 13. The spatial distribution of snow is well captured by the model. However, the model tends to melt snow slighlty too late in spring, as highlighted also by the seasonal evolution of the total NH snow mass (Fig. 14). The interannual variability of spring snow over the NH is also largely in agreement with the GlobSnow data, suggesting that the model has a reasonablly good sensitivity (Fig. 15).

Modelled permafrost area is $16.5\,\mathrm{mln\,km^2}$, which compares well with observations (Table 8). The permafrost extent over Siberia and Northern Canada in generally well simulated by the model (Fig. 16). Also the active layer thickness over the Yakutia region is consistent with the data from Beer et al. (2013) (Fig. 16).

## 8.2 Biogeochemistry

The modelled annual mean gross primary productivity is compared to estimates from Jung et al. (2009); Beer et al. (2010); Jung et al. (2011) in Fig. 17. Model and data are generally in good agreement, except over the Amazon where the model underestimates GPP. However, experiments using different climate forcings show a much better agreement between modelled and observed GPP over the Amazon basin (not shown). The simulated global annual GPP of $126\,\mathrm{PgC\,yr^{-1}}$ is within the range of current estimates (Table 8).

The net global land carbon flux over the time period 1959-2010 is shown in Fig. 18. The model is able to reproduce some of the interannaul variability in the net land carbon uptake, indicating that the sensitivities of net primary production and soil respiration to interannual climate variations are reasonably well represented in the model.

The modelled potential vegetation distribution for the present day is shown in Fig. 19, where it is compared to potential vegetation estimates from Ramankutty and Foley (1999). In general the model has the tendency to overestimate the areas covered by broadleaf trees in the tropics. The boreal needleleaf forest is well reproduced by the model, as are the grass and shrub coverages. Desert area is overestimated over Australia.



Global modelled vegetation carbon is 500 PgC, comparable to observations (Table 8). The geographic distribution of vegetation carbon content is in good agreement with data from Gibbs (2006) (Fig. 20).

The annual mean flux weighted discrimation is shown in Fig. 21. Mean discrimination for each plant functional type is also compared with observations from Kaplan et al. (2002) in Fig. 22. The model consistently tends to overestimate the discrimination for all PFTs.

Top meter soil carbon from the HSWD dataset (FAO/IIASA/ISRIC/ISSCAS/JRC, 2012) is well reproduced by the model, although the model underestimates carbon in peatland areas of the NH (Fig. 23). Top meter soil carbon is also compared to the NCSCD soil carbon dataset for the permafrost region (Hugelius et al., 2013b, a; Tarnocai et al., 2009) in Fig. 24.

Northern permafrost areas store large amounts of carbon at depths greater than 1 m. The NCSCD soil carbon dataset contains estimates of soil carbon down to a depth of 3 m in the permafrost regions. As expected, the model in the setup used in the presented simulations can not reproduce the large amounts of carbon stored in perenially frozen ground below 1 m because the inert permafrost carbon pool is not included (not shown). To get the carbon accumulation in permafrost a transient simulation over at least the last glacial cycle would be required. This is beyond the scope of this work, but will be discussed in a future paper. Similarly to the discussion on permafrost carbon, a proper estimate of peatland area and carbon content would also require a long transient simulation. However, an attempt has been made to estimate the peatland area and carbon using the equilibrium spinup described above. The estimated peatland area from this idealized approach is compared to NCSCD data (Hugelius et al., 2013b, a; Tarnocai et al., 2009) in Fig. 25.

Total modelled natural methane emissions for the present day are 175 $TgCH_4 \, yr^{-1}$. From these, 105 $TgCH_4 \, yr^{-1}$ are from the tropics and 70 $TgCH_4 \, yr^{-1}$ from the extratropics. This values compare well with recent estimates of natural methane emissions (Table 8). The spatial distribution of annual methane emissions is shown in Fig. 26.

## 9 Conclusions

The PALADYN model presented here represents a new tool to model the land processes which are relevant for climate and the carbon cycle on time scales from years to millions of years.

PALADYN serves as a land surface scheme, soil model, dynamic vegetation model and land carbon cycle model. It also includes a representation of peatlands and soil carbon pools in frozen ground. Compared to other land surface models it has the great advantage that all components are consistenly coupled.

PALADYN furthermore includes a representation of the processes related to changes in land-ice-shelf area, making it suitable for simulations over timescales where sea level and ice sheet areas can



not be considered as fixed boundary conditions. PALADYN is therefore designed to be included in Earth system models of intermediate complexity.

On a single CPU the model in its standard configuration (daily time step, 5x5 degree horizontal resolution and 5 soil layers) integrates one year in about 1 second (or equivalently about 100,000 model years per day), allowing to simulate e.g. one glacial cycle in one day. It is therefore indicated for paleoclimate applications or to perform large ensembles of simulations to explore uncertainties and sensitivities.

PALADYN in its offline version has been shown to perform well at reproducing a number of key characteristics of the present day land surface, soil, vegetation and land carbon cycle and is therefore ready to be included in Earth system models in a coupled setup.

## 10   Code availability

The model code is available on request from the authors.

## 895  Appendix A:  Snow age factor

The snow age factor, $f_{\mathrm{age}}$, is parameterized as a function of skin temperature and snowfall rate $P_{\mathrm{s}}$ as:

$$f_{\mathrm{age}} = 1 - \frac{\ln(1 + f_{\mathrm{age}}^{\mathrm{T}} \frac{P_{\mathrm{s,c}}}{P_{\mathrm{s}}})}{f_{\mathrm{age}}^{\mathrm{T}} \frac{P_{\mathrm{s,c}}}{P_{\mathrm{s}}}}, \tag{A1}$$

$$f_{\mathrm{age}}^{\mathrm{T}} = e^{0.05(T_\star - T_0)} + e^{(T_\star - T_0)}. \tag{A2}$$

The dependence of $f_{\mathrm{age}}$ on temperature and snowfall rate with $P_{\mathrm{s,c}} = 2 \times 10^{-5} \, \mathrm{kg \, m^{-2} \, s^{-1}}$ is shown in Fig. A1.

## Appendix B:  Soil thermal and hydraulic properties

Organic matter alters soil thermal and hydraulic properties substantially, in particular because of the much higher porosity of organic soils compared to mineral soils. The importance of accounting for

organic matter in land-surface models has been discussed in e.g. Rinke et al. (2008); Lawrence and Slater (2008); Koven et al. (2009); Ekici et al. (2014); Chadburn et al. (2015). In PALADYN, the fraction of soil that is considered to be organic for the determination of the thermal and hydraulic soil properties is computed in each soil layer from the total carbon density following Lawrence and Slater (2008):

$$f_{\mathrm{org}} = \min\left(1, \frac{C_{\mathrm{lit}} + C_{\mathrm{fast}} + C_{\mathrm{slow}}}{\rho_{\mathrm{org}}^{\max}}\right). \tag{B1}$$

$\rho_{\mathrm{org}}^{\max} = 50 \, \mathrm{kgC \, m^{-3}}$ is the maximum soil carbon density, equivalent to a typical carbon density of peat.





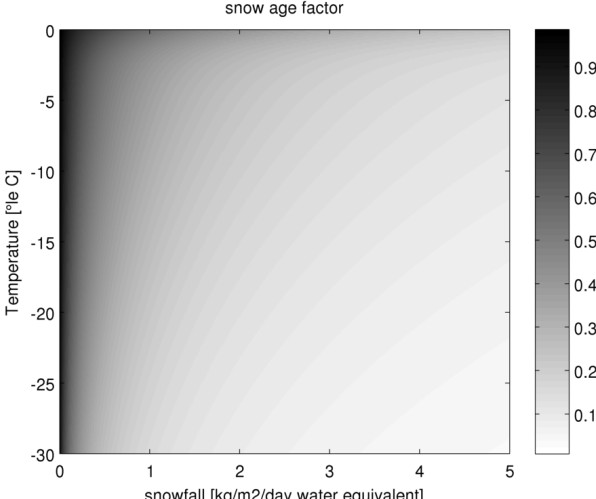

**Figure A1.** Snow age factor as a function of skin temperature and snowfall rate.

Soil thermal and hydraulic properties are simply taken to be a linear combination of mineral and organic values based on $f_{\mathrm{org}}$. This linear combination is applied to porosity ($\theta_{\mathrm{sat}}$), dry thermal conductivity ($\lambda_{\mathrm{dry}}$), solid soil thermal conductivity ($\lambda_{\mathrm{s}}$), saturated hydraulic conductivity ($k_{\mathrm{sat}}$), saturation matric potential ($\psi_{\mathrm{sat}}$) and $b$ parameter.

Mineral soil properties are computed from sand and clay fractions following Lawrence and Slater (2008), based on (Cosby et al., 1984; Farouki, 1981; Clapp and Hornberger, 1978). Sand and clay fractions are taken from either FAO/IIASA/ISRIC/ISSCAS/JRC (2012) or from Shangguan et al. (2014), with the former set as default. Sand and clay fractions are considered to be vertically uniform in each grid cell and constant in time.

Organic soil properties are also taken from Lawrence and Slater (2008), partly based on Letts et al. (2000); Farouki (1981).

**Appendix C: Photosynthesis**

The maximum daily rate of net photosynthesis $V_{\mathrm{m}}$ is given by:

$$V_{\mathrm{m}} = \frac{1}{a_{\mathrm{C3/4}}} \frac{c_1}{c_2} \left[ (2\theta_{\mathrm{r}} - 1)s - (2\theta_{\mathrm{r}}s - c_2)\sigma_{\mathrm{m}} \right] APAR. \tag{C1}$$

$$\sigma_{\mathrm{m}} = \sqrt{1 - \frac{c_2 - s}{c_2 - \theta_{\mathrm{r}} \cdot s}}, \tag{C2}$$

where:

$$s = \frac{24}{d_{\mathrm{h}}} a_{\mathrm{C3/4}} \tag{C3}$$





and $d_\mathrm{h}$ is the daylength in hours computed from orbital parameters.

All PALADYN PFTs follow the C3 photosynthetic pathway, except C4 grasses which follow the C4 pathway. For C3 plants $c_1$ and $c_2$ are given by:

$$c_1 = \alpha_{\mathrm{C3}} f_{\mathrm{temp}} C_{\mathrm{mass}} \frac{p_\mathrm{i} - \Gamma_\star}{p_\mathrm{i} + \Gamma_\star},$$ (C4)

$$c_2 = \frac{p_\mathrm{i} - \Gamma_\star}{p_\mathrm{i} + K_\mathrm{c}(1 + [\mathrm{O}_2]/K_\mathrm{o})}.$$ (C5)

$\Gamma_\star$ is the $CO_2$ compensation point:

$$\Gamma_\star = \frac{[\mathrm{O}_2]}{2\tau}.$$ (C6)

$K_\mathrm{c}$, $K_\mathrm{o}$ and $\tau$ are kinetic parameters whose temperature dependence is modeled using a Q10 relationship (Haxeltine and Prentice, 1996a). $f_{\mathrm{temp}}$ is a PFT specific temperature inhibition function 940 (Sitch et al., 2003).

For C4 plants the same equations are used but with $c_1$ and $c_2$ given by:

$$c_1 = \alpha_{\mathrm{C4}} f_{\mathrm{temp}},$$ (C7)

$$c_2 = 1.$$ (C8)

### Appendix D:  Aggregation of potential vegetation

In order to compare the modelled with the potential vegetation distribution of Ramankutty and Foley (1999), the potential vegetation needs to be aggreated to the plant functional types represented in PALADYN. We partly follow Blyth et al. (2011) and map vegetation classes of Ramankutty and Foley (1999) to the PALADYN PFTs as described in Table D1. The grass class in Table D1 is divided into C3 and C4 grasses based on the modelled grass type in each grid cell.





**Table D1.** Mapping of Ramankutty and Foley (1999) potential vegetation classes to PALADYN PFTs.

|  | BL | NL | grass | Shrubs | Bare soil |
|---|---|---|---|---|---|
| Tropical evergreen | 0.9 | 0.0 | 0.0 | 0.0 | 0.1 |
| Tropical deciduous | 0.8 | 0.0 | 0.1 | 0.0 | 0.1 |
| Temperate broadleaved evergreen | 0.9 | 0.0 | 0.0 | 0.0 | 0.1 |
| Temperate needleleaved evergreen | 0.0 | 0.9 | 0.0 | 0.0 | 0.1 |
| Temperate deciduous | 0.9 | 0.0 | 0.0 | 0.0 | 0.1 |
| Boreal evergreen | 0.0 | 0.9 | 0.0 | 0.0 | 0.1 |
| Boreal deciduous | 0.0 | 0.8 | 0.1 | 0.0 | 0.1 |
| Mixed evergreen/deciduous | 0.1 | 0.7 | 0.0 | 0.1 | 0.1 |
| Savanna | 0.3 | 0.0 | 0.6 | 0.0 | 0.1 |
| Grassland/steppe | 0.0 | 0.0 | 0.9 | 0.0 | 0.1 |
| Dense shrub | 0.0 | 0.0 | 0.1 | 0.8 | 0.1 |
| Open shrub | 0.0 | 0.0 | 0.2 | 0.5 | 0.3 |
| Tundra | 0.0 | 0.0 | 0.35 | 0.35 | 0.3 |



*Acknowledgements.*  The authors would like to thank Victor Brovkin and Daniela Dalmonech for discussions
and Catherine Pringent for providing the GIEMS dataset. M. Willeit acknowledges support by the German
Science Fundation DFG grant GA 1202/2-1.



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





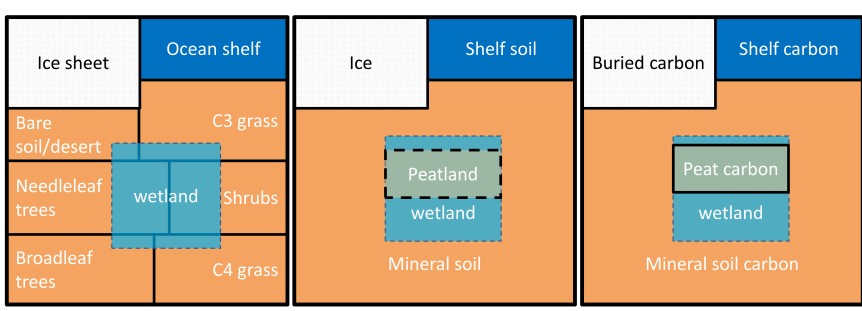

**Figure 1.** PALADYN surface types (left), 'soil' columns (middle) and soil carbon pools (right).





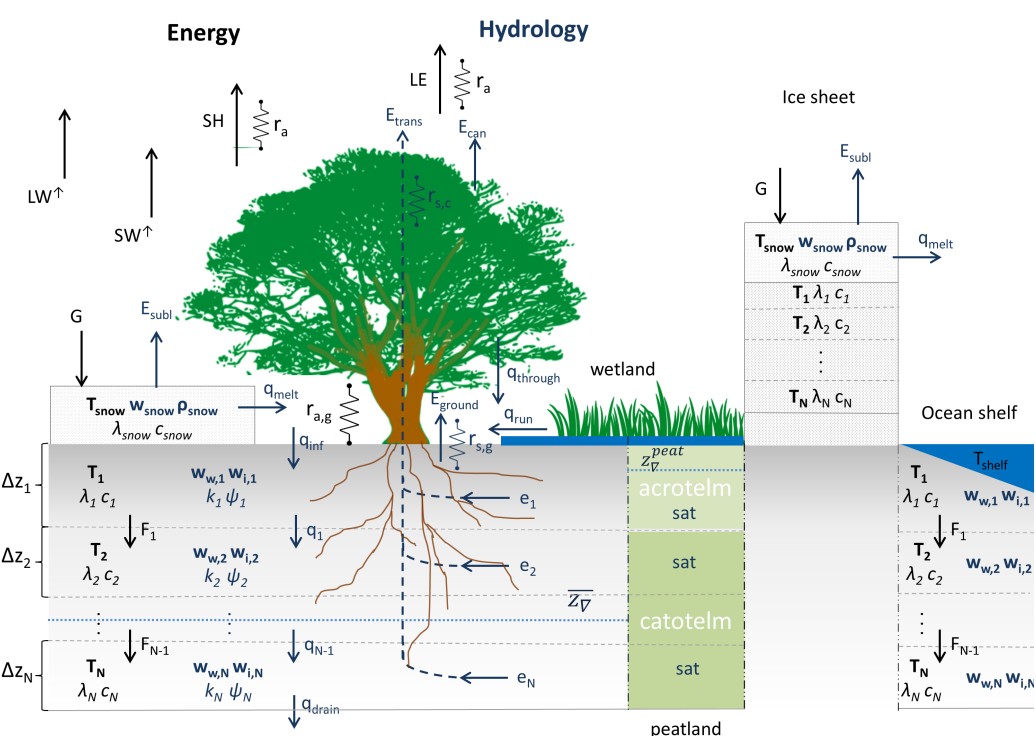

**Figure 2.** Illustration of the physical processes included in PALADYN. Energy fluxes and variables are indicated in black while water fluxes and hydrological variables are indicated in blue. Prognostic variables are in bold and fluxes are accompanied by arrows.





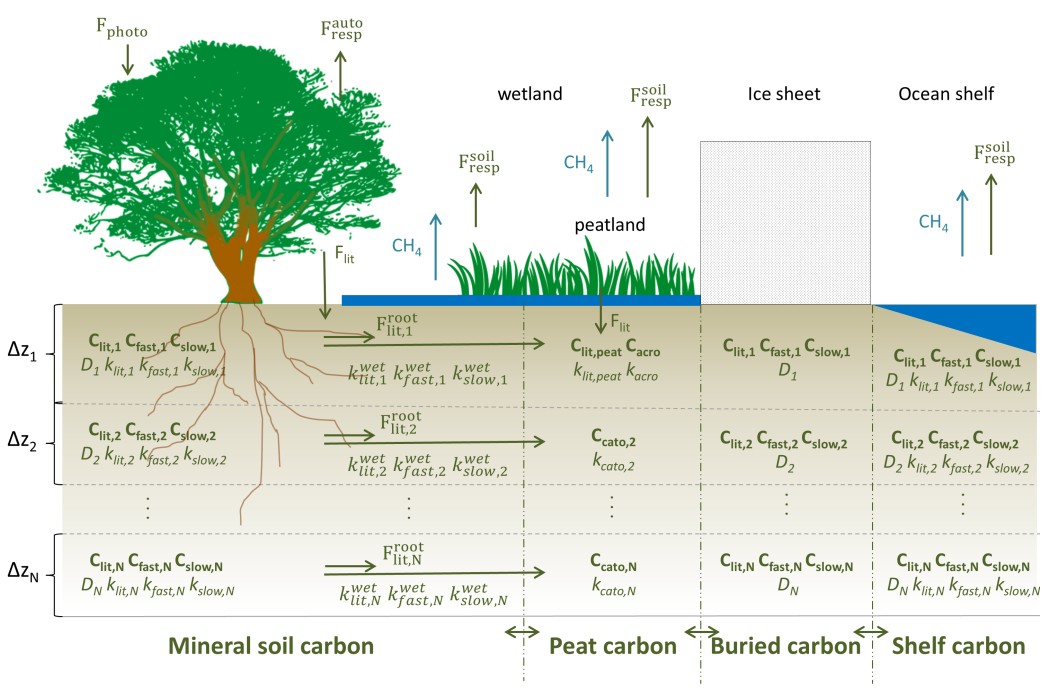

**Figure 3.** Illustration of the carbon cycle processes represented in PALADYN. Prognostic variables are in bold and fluxes are accompanied by arrows.




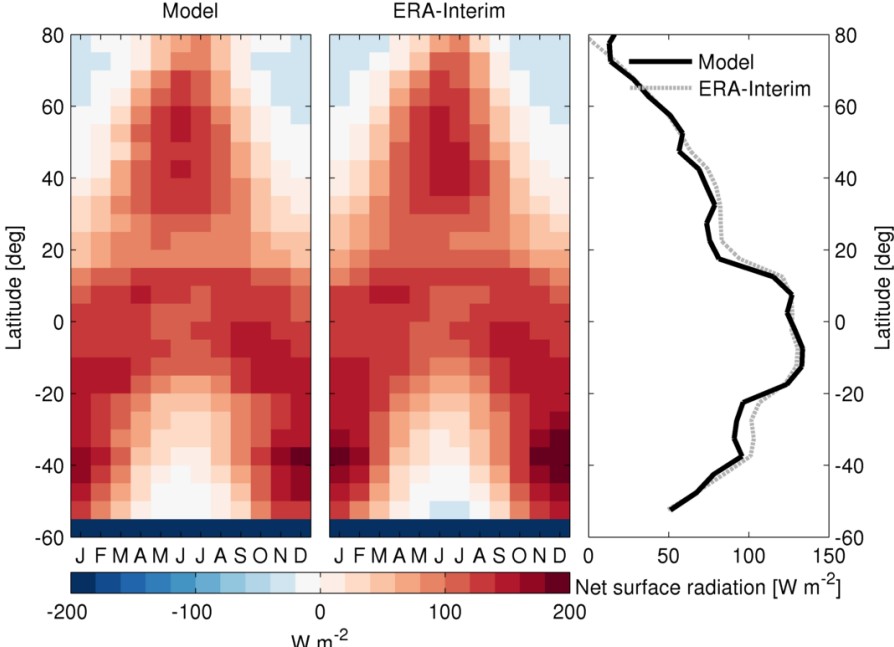

**Figure 4.** Seasonal variation of zonal mean net radiation at the surface modelled by PALADYN (a) and from ERA-Interim reanalysis (Dee et al., 2011) (b). (c) Modelled zonal annual mean net surface radiation compared to ERA-Interim reanalysis (Dee et al., 2011).

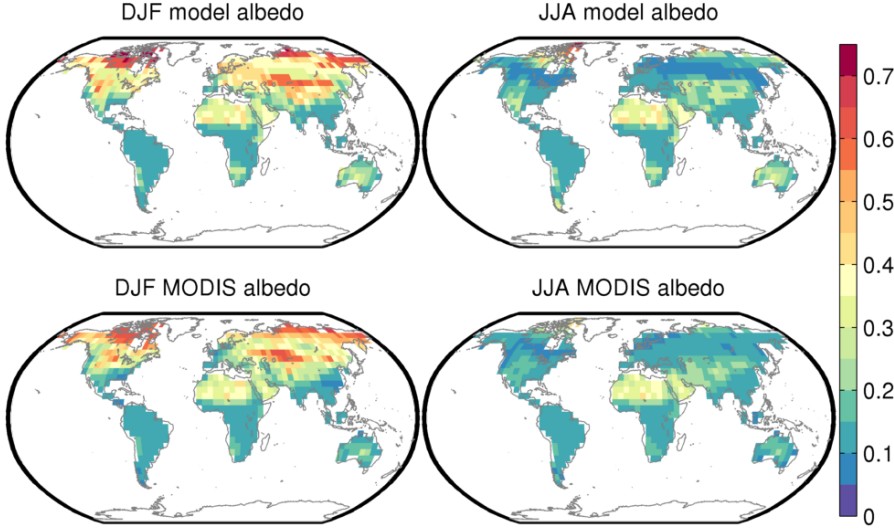

**Figure 5.** December-January-February (left) and June-July-August (right) surface albedo as modelled by PAL-ADYN (top) and derived from MODIS data (Schaaf and Wang, 2015) (bottom). The displayed surface albedo is a weighted mean of visible and near infrared broadband albedo for diffuse radiation.





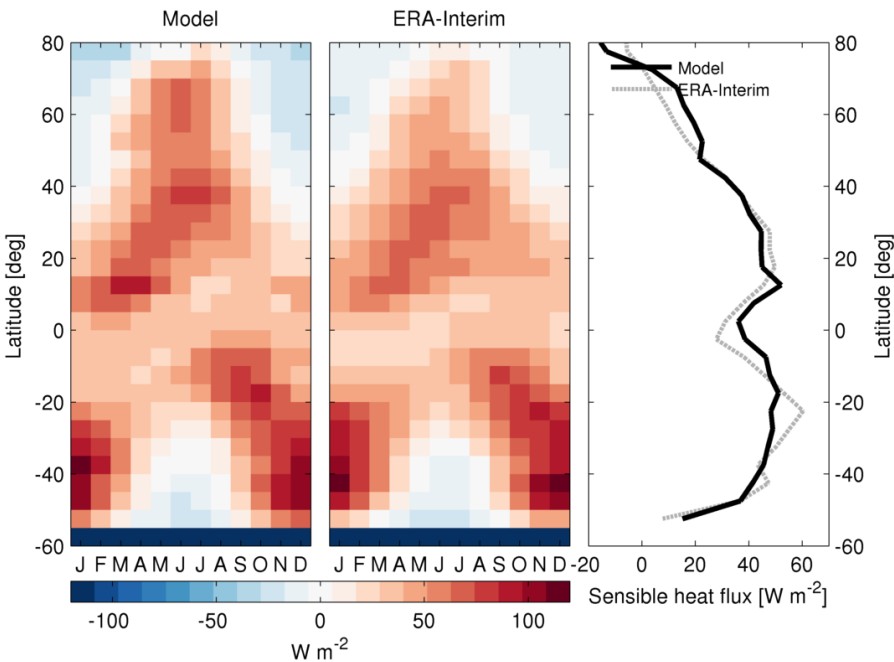

**Figure 6.** Seasonal variation of zonal mean sensible heat flux modelled by PALADYN (left) and from ERA-Interim reanalysis (Dee et al., 2011) (middle). (right) Modelled zonal annual mean sensible heat flux compared to ERA-Interim reanalysis (Dee et al., 2011).





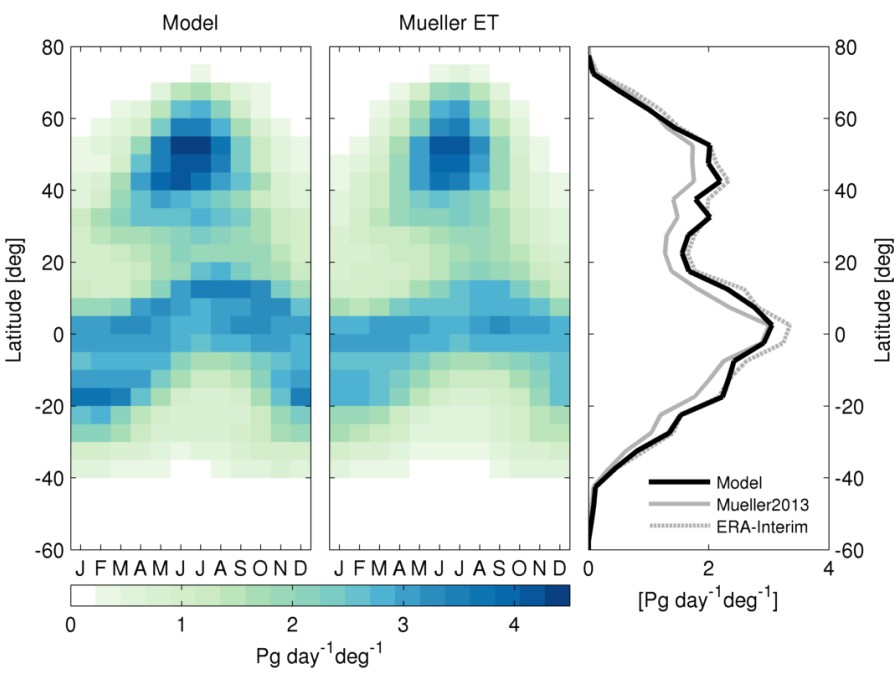

**Figure 7.** Seasonal variation of zonally integrated evapotranspiration modelled by PALADYN (left) and estimated by Mueller et al. (2013) (middle). (riht) Modelled zonal annual mean evapotranspiration compared to observation-based estimates from Mueller et al. (2013) and ERA-Interim reanalysis (Dee et al., 2011).





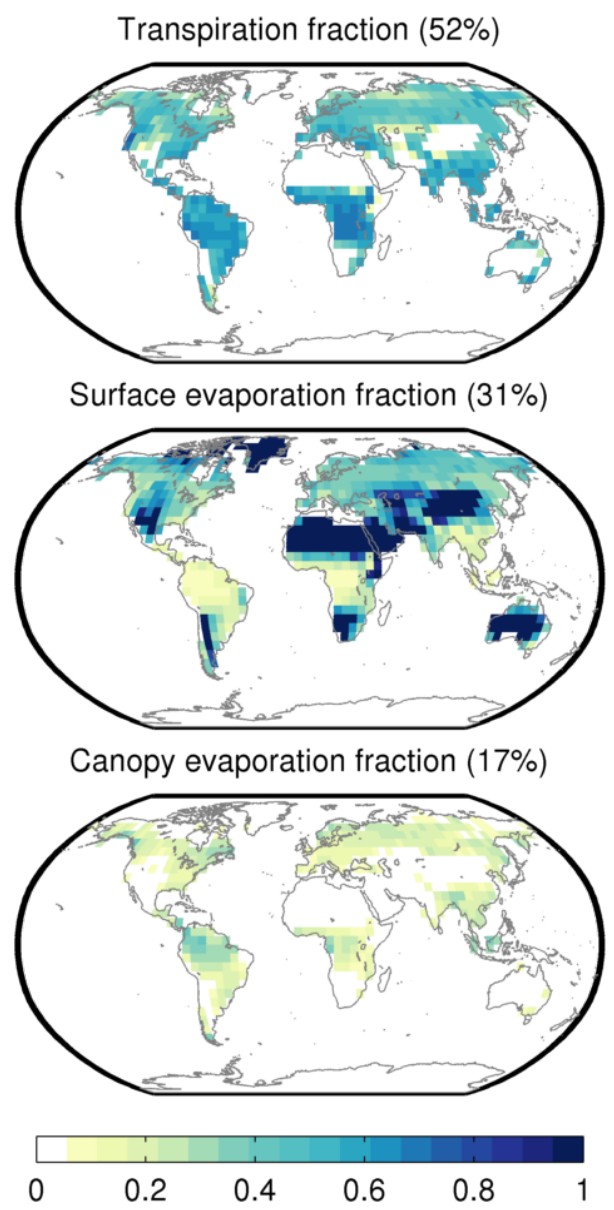

**Figure 8.** Partitioning of modelled total annual evapotranspiration between transpiration (top), surface evaporation (middle) and canopy evaporation (bottom). The global percentage of each component is shown above the corresponding plot.





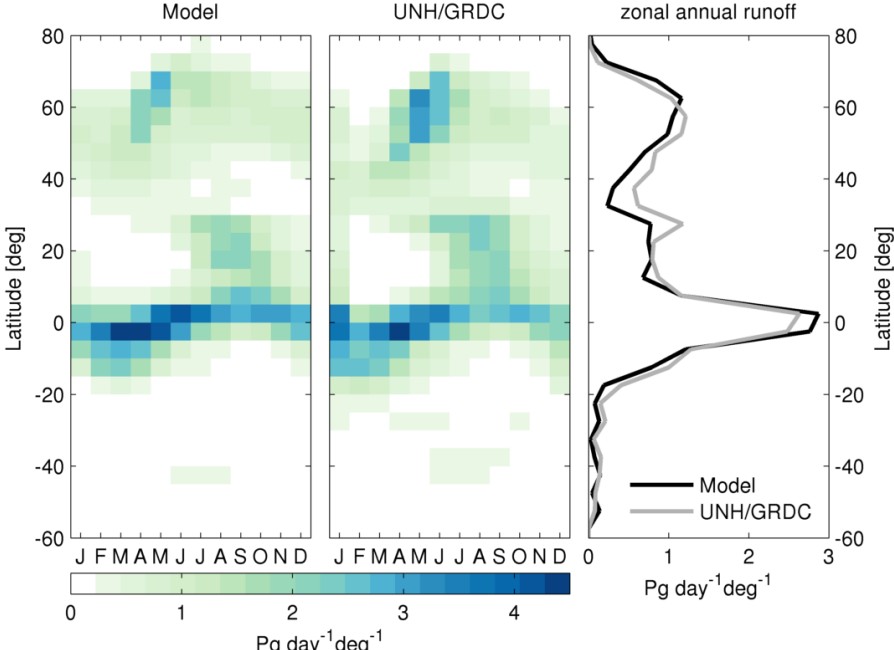

**Figure 9.** Seasonal variation of zonally integrated water runoff modelled by PALADYN (left) and observed by UNH/GRDC (Fekete et al., 2002) (middle). (right) Modelled zonal annual mean runoff compared to observation-based estimates from Fekete et al. (2002). Modelled and observed runoff is averaged over the time period 1979-2010.

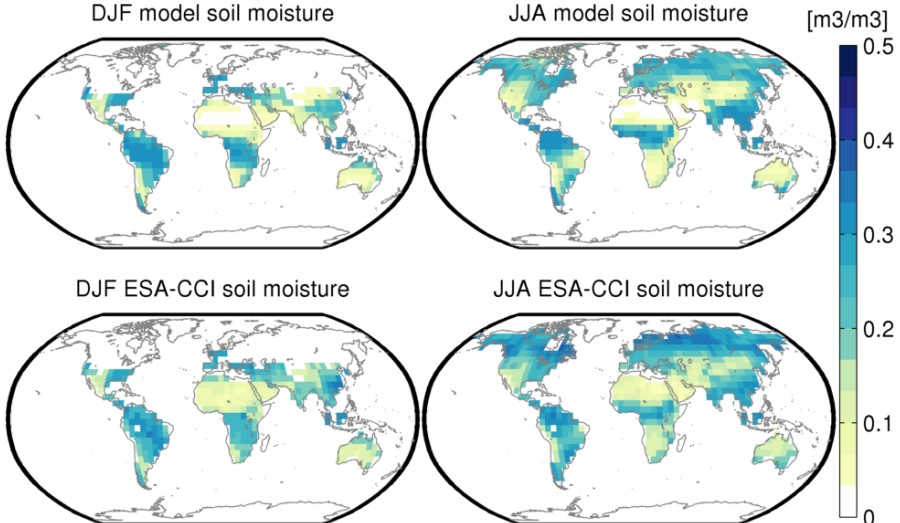

**Figure 10.** December-January-February (left) and June-July-August (right) soil moisture. The modelled soil moisture (top) is the volumetric soil moisture of the top soil layer (top 20 cm). The observed soil moisture is from ESA-CCI (Liu et al., 2012; Wagner et al., 2012) and represents the moisture content of the top few centimeters of soil. Snow covered regions are masked out.



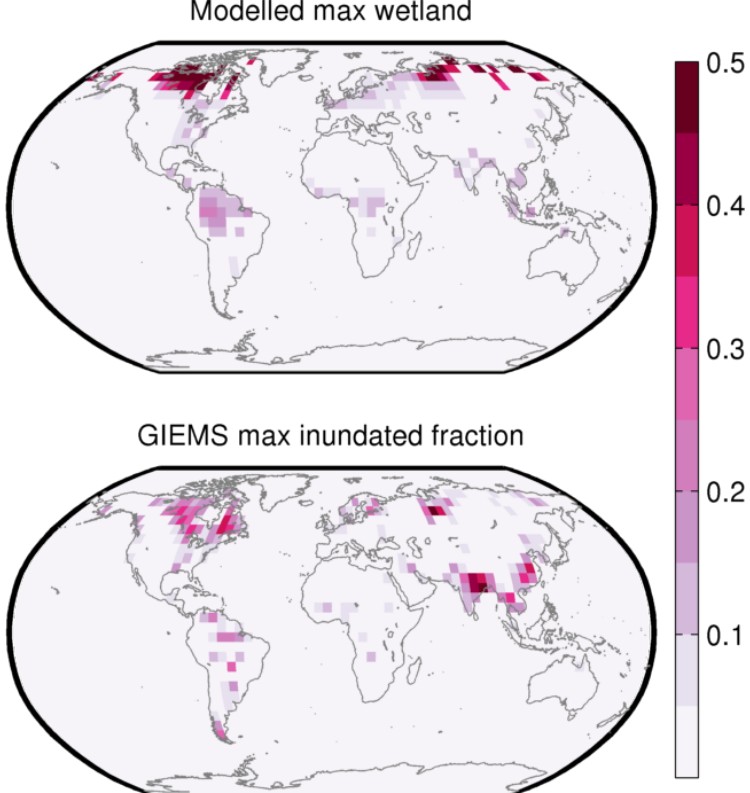

**Figure 11.** Monthly maximum wetland fraction over the time interval 1993-2007 as modelled by PALADYN (top) and inferred by GIEMS (Prigent et al., 2007; Papa et al., 2010) (bottom).

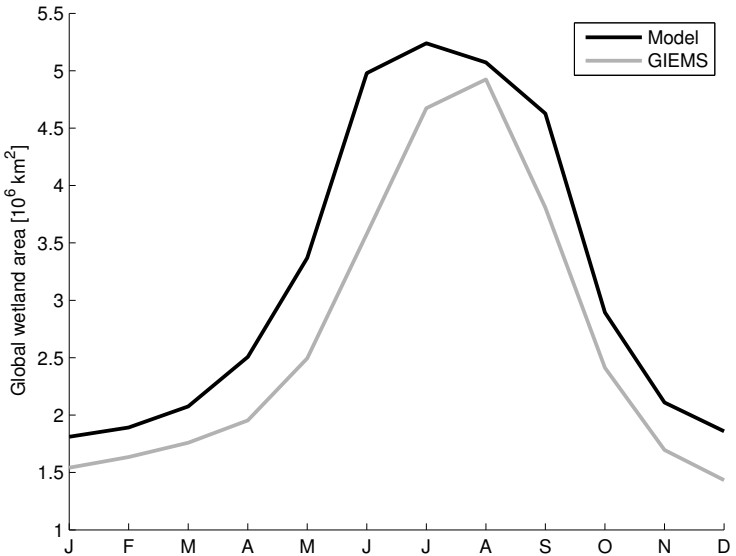

**Figure 12.** Mean seasonal global wetland extent over the time interval 1993-2007 as modelled by PALADYN and inferred by GIEMS (Prigent et al., 2007; Papa et al., 2010).



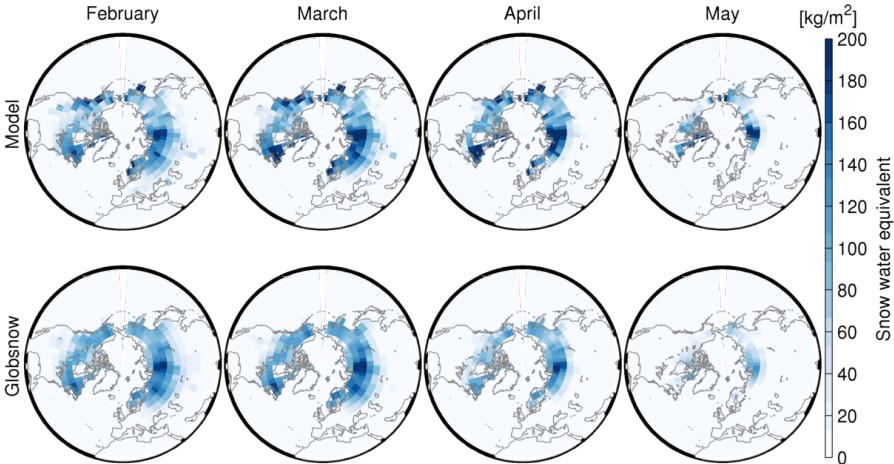

**Figure 13.** February to May snow water equivalent mean over the period 1980-2010 for PALADYN (top) compared to data from the GlobSnow project (Takala et al., 2011; Luojus et al., 2013) (bottom).

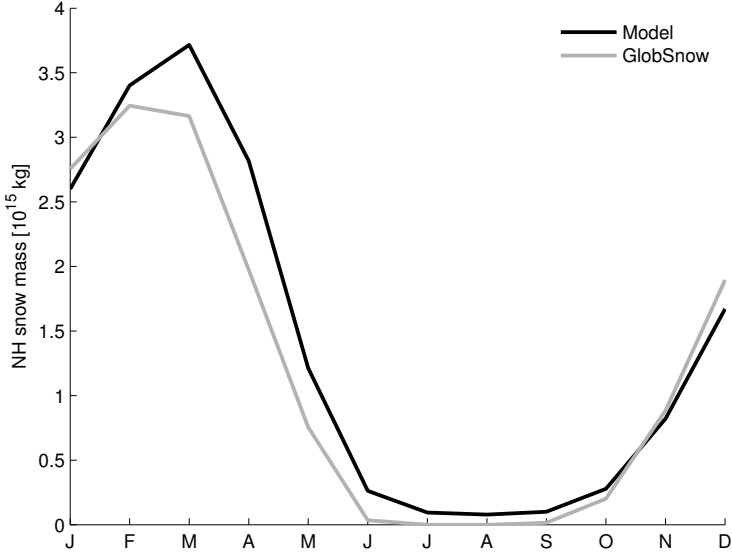

**Figure 14.** Mean 1980-2010 seasonal evolution of the total Northern Hemisphere snow mass compared to data from the GlobSnow project (Takala et al., 2011; Luojus et al., 2013).





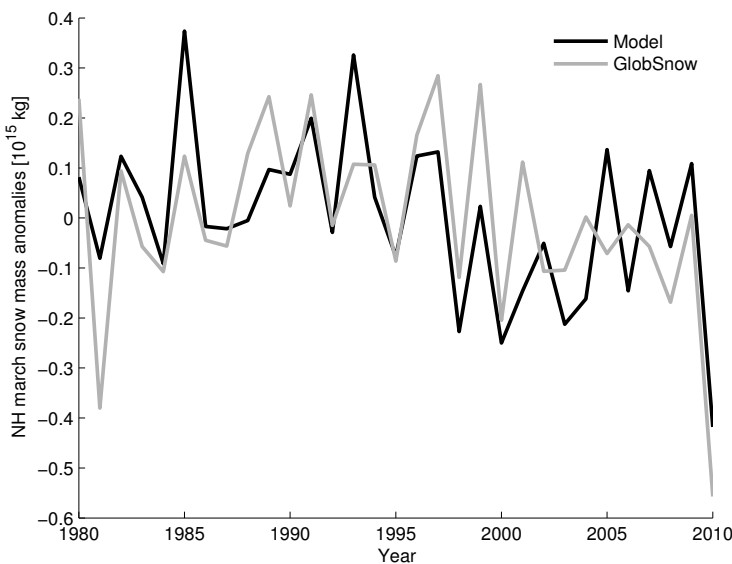

**Figure 15.** Northern Hemisphere March snow mass anomalies from 1980 to 2010 compared to data from the GlobSnow project (Takala et al., 2011; Luojus et al., 2013).

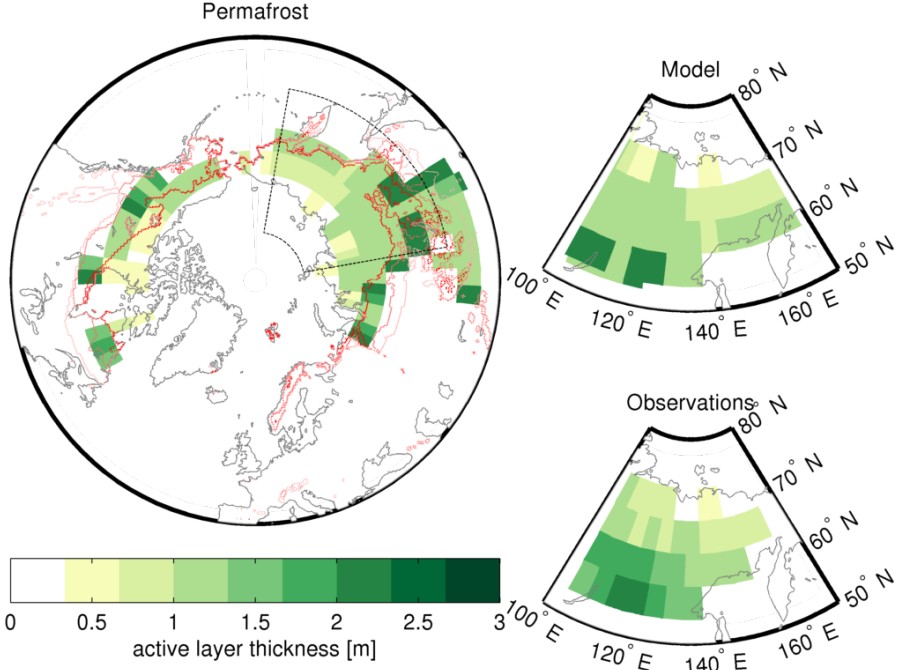

**Figure 16.** Left: modelled permafrost extent and active layer thickness compared to the oberved extent of continous, discontinuous and isolated permafrost (red lines, from dark red to light red) from (Brown et al., 2014). Right: comparison of modelled (top) and observed (bottom) active layer thickness over Yakutia. Active layer thickness data are from (Beer et al., 2013). The modelled active layer thickness is calculated as the mean over the period 1981-2010 in grid cells that are permafrost during the whole time period.





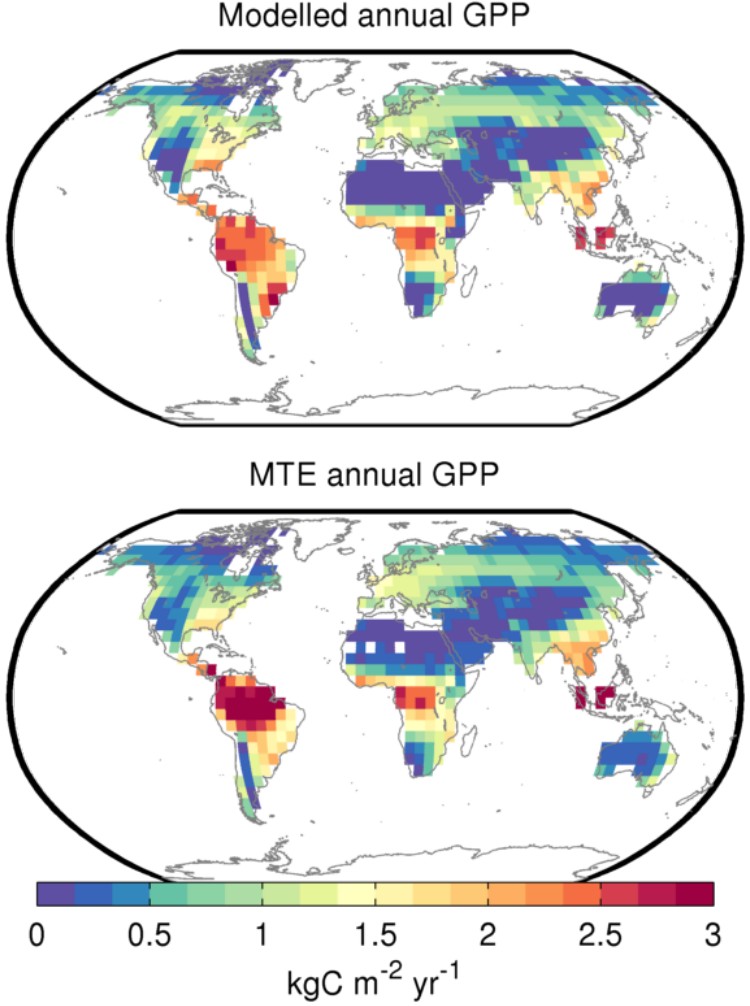

**Figure 17.** Mean annual gross primary production (GPP) over the time interval 1980-2010 as modelled by PALADYN (top) and estimated by the model tree ensemble approach (MTE) (Jung et al., 2009, 2011) (bottom).



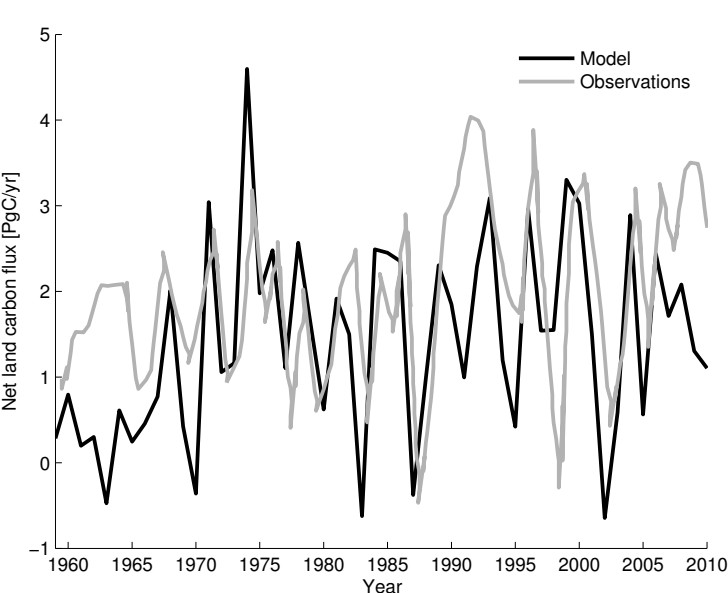

**Figure 18.** Net land carbon uptake for the historical simulation compared to observations from IPCC.





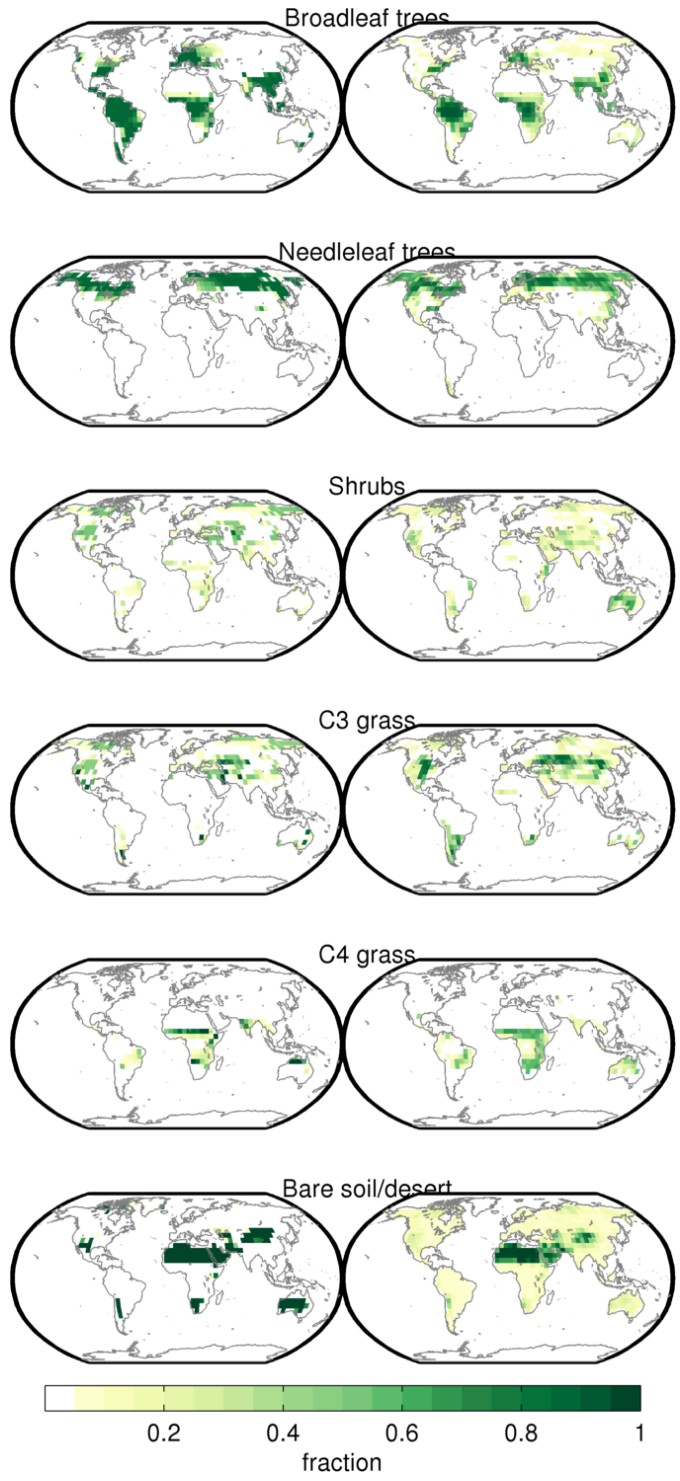

**Figure 19.** Comparison of modelled plant functional types fraction (left) with potential vegetation distribution adapted from Ramankutty and Foley (1999) as described in Appendix D (right).




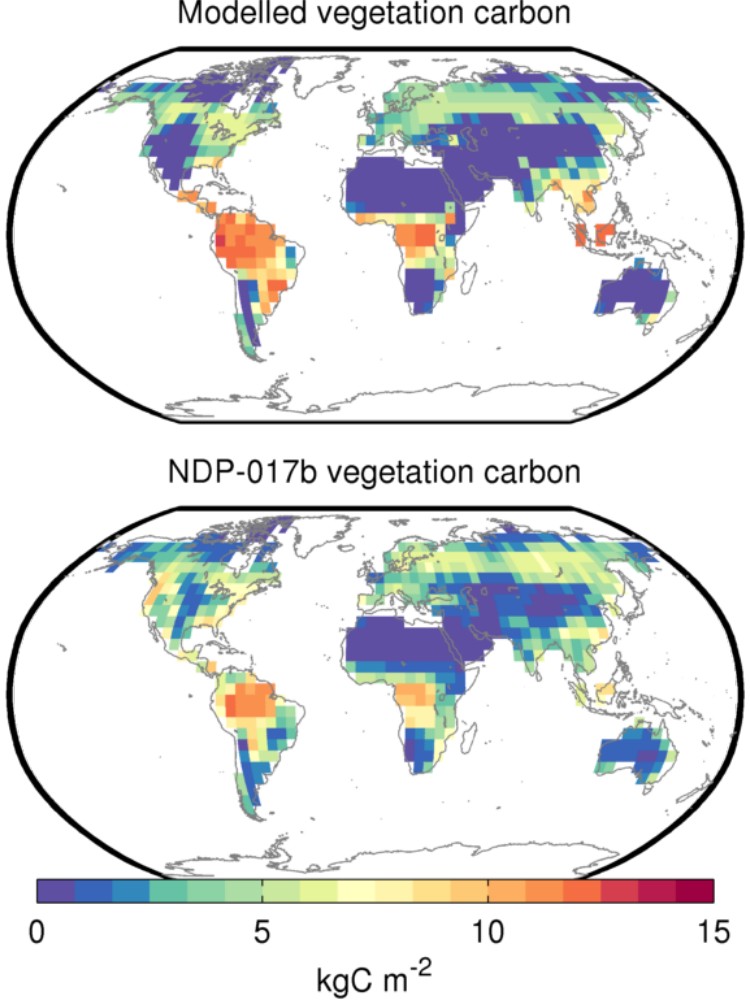

**Figure 20.** Comparison of modelled vegetation carbon content (top) with the observational estimates from the NDP-017b dataset (Gibbs, 2006) (bottom).

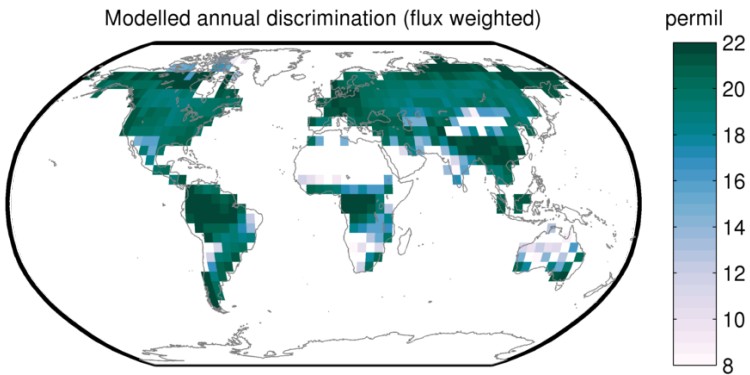

**Figure 21.** Modelled annual flux weighted discrimination during photosynthesis.





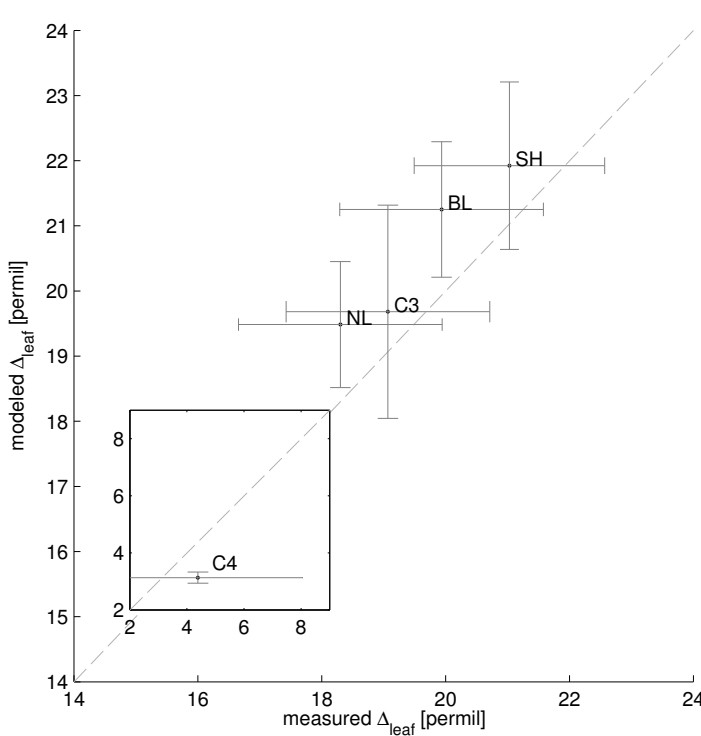

**Figure 22.** Comparison of modelled and observed discrimination during photosynthesis for different plant functional types. Observational data are from (Kaplan et al., 2002).





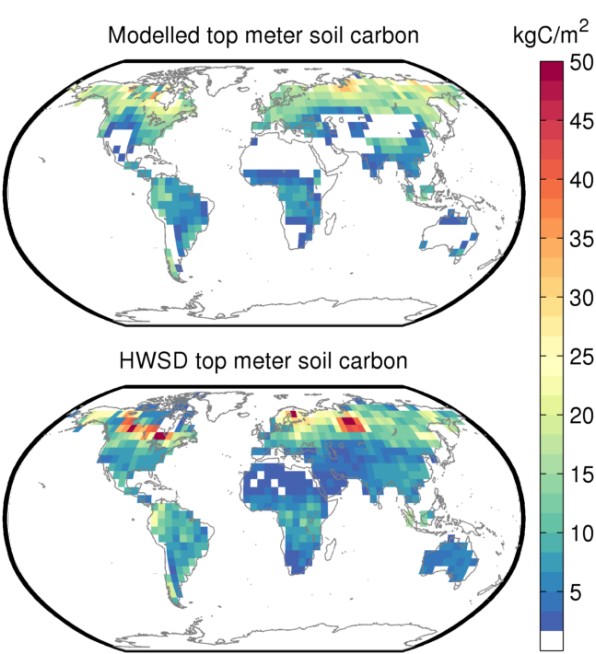

**Figure 23.** Top 1 m soil carbon as modelled by PALADYN (top) and derived from the Harmonized World Soil Database (FAO/IIASA/ISRIC/ISSCAS/JRC, 2012) (bottom).





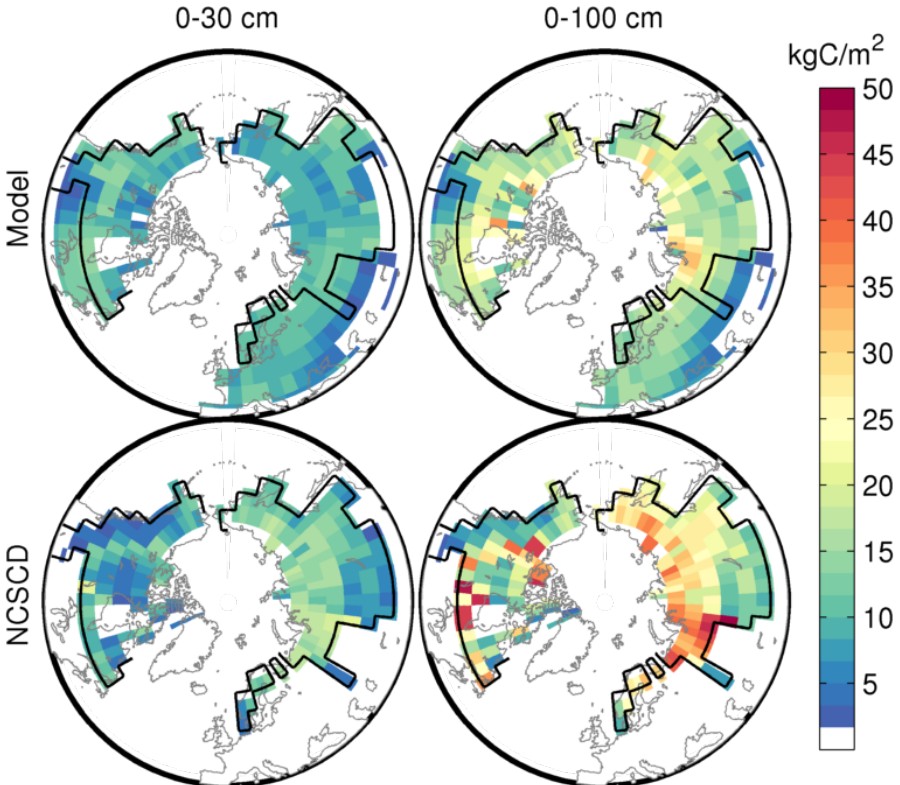

**Figure 24.** Comparison of modelled (top) soil carbon in Northern permafrost regions with estimates from the Northern Circumpolar Soil Carbon Database (NCSCD) (Hugelius et al., 2013b, a; Tarnocai et al., 2009) (bottom) for two depth ranges: (left) 0-30 cm and (right) 0-100 cm.

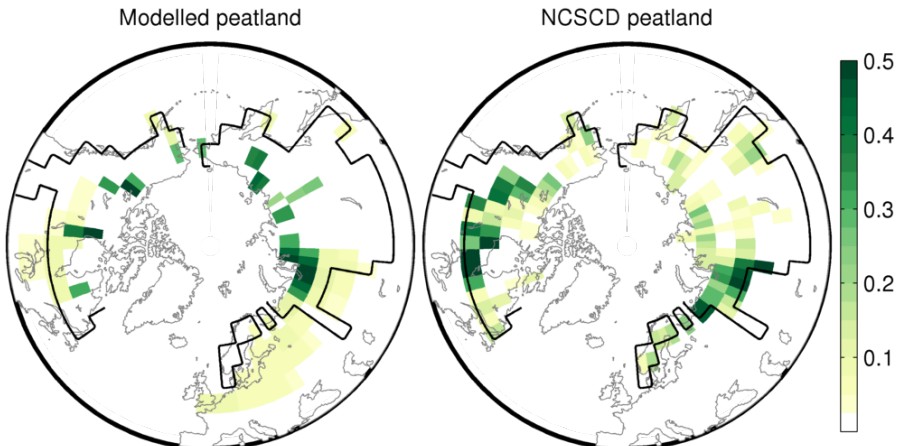

**Figure 25.** Peat fraction as modelled by PALADYN (left) compared to estimates from the Northern Circumpolar Soil Carbon Database (NCSCD) (Hugelius et al., 2013b, a; Tarnocai et al., 2009) (right). The permafrost area as defined in NCSCD is shown as black line. No data are available from the NCSCD dataset outside this area.





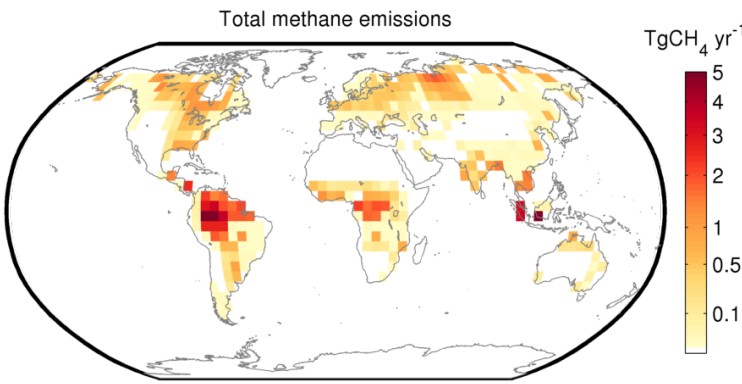

**Figure 26.** Modelled annual methane emissions.



Table 1: Symbol definitions

| Symbol | Units | Definition |
|---|---|---|
| $\Delta$ | ‰ | isotopic discrimination |
| $\Delta z_l$ | m | thickness of soil layer $l$ |
| $\Lambda$ | $\mathrm{kgC\,m^{-2}\,s^{-1}}$ | litterfall rate |
| $\Lambda_{\mathrm{bur}}$ | $\mathrm{kgC\,m^{-2}\,s^{-1}}$ | vegetation carbon burial rate under ice sheets |
| $\Lambda_{\mathrm{l}}$ | $\mathrm{kgC\,m^{-2}\,s^{-1}}$ | leaf litterfall rate |
| $\Lambda_{\mathrm{loc}}$ | $\mathrm{kgC\,m^{-2}\,s^{-1}}$ | local litterfall rate |
| $\Lambda_{\mathrm{peat}}$ | $\mathrm{kgC\,m^{-2}\,s^{-1}}$ | litterfall rate over peatland |
| $\Lambda_{\mathrm{shelf}}$ | $\mathrm{kgC\,m^{-2}\,s^{-1}}$ | litterfall rate over ocean shelf |
| $\Lambda_{\mathrm{veg}}$ | $\mathrm{kgC\,m^{-2}\,s^{-1}}$ | litterfall rate over vegetated grid cell area |
| $\alpha$ | | surface albedo |
| $\alpha_{\mathrm{a}}$ | | factor for $APAR$ |
| $\alpha^{\mathrm{dir}}$ | | albedo for direct radiation |
| $\alpha^{\mathrm{dif}}$ | | albedo for diffuse radiation |
| $\alpha^{\mathrm{vis}}$ | | visible broadband albedo |
| $\alpha^{\mathrm{nir}}$ | | near infrared broadband albedo |
| $\alpha_{\mathrm{can}}$ | | canopy albedo |
| $\alpha_{\mathrm{g}}$ | | ground albedo |
| $\alpha_{\mathrm{int}}^{s}$ | | snowfall interception factor |
| $\alpha_{\mathrm{int}}^{w}$ | | rainfall interception factor |
| $\alpha_{\mathrm{leaf}}$ | | leaf albedo |
| $\alpha_{\mathrm{sn}}$ | | snow albedo |
| $\alpha_{\mathrm{sn}}^{\mathrm{can}}$ | | albedo of snow covered canopy |
| $\alpha_{\mathrm{sn,fresh}}$ | | fresh snow albedo |
| $\alpha_{\mathrm{snfree}}$ | | snowfree surface albedo |
| $\alpha_{\mathrm{snfree}}^{\mathrm{can}}$ | | albedo of snowfree canopy |
| $\alpha_{\mathrm{soil}}$ | | snowfree soil albedo |
| $\beta_{\theta}$ | | soil moisture limitation factor for photosynthesis |
| $\beta_{\mathrm{s}}$ | | surface evaporation factor |
| $\gamma_{\mathrm{v}}$ | $\mathrm{s^{-1}}$ | PFT disturbance rate |
| $\gamma_{\mathrm{v,min}}$ | $\mathrm{s^{-1}}$ | minimum PFT disturbance rate |
| $\gamma_{\mathrm{l}}$ | $\mathrm{s^{-1}}$ | leaf turnover rate |
| $\gamma_{\mathrm{r}}$ | $\mathrm{s^{-1}}$ | root turnover rate |
| $\gamma_{\mathrm{s}}$ | $\mathrm{s^{-1}}$ | stem turnover rate |
| $\epsilon$ | | longwave emissivity |



| | | |
|---|---|---|
| $\eta$ | Pa s | snow viscosity |
| $\eta_0$ | Pa s | reference snow viscosity |
| $\theta$ | m$^3$ m$^{-3}$ | volumetric total soil moisture |
| $\theta_{\mathrm{crit}}$ | m$^3$ m$^{-3}$ | critical soil moisture for fire |
| $\theta_{\mathrm{fc}}$ | m$^3$ m$^{-3}$ | volumetric soil moisture at field capacity |
| $\theta_{\mathrm{i}}$ | m$^3$ m$^{-3}$ | volumetric frozen soil moisture |
| $\theta_{\mathrm{r}}$ | | shape parameter for photosynthesis |
| $\theta_{\mathrm{sat}}$ | m$^3$ m$^{-3}$ | soil porosity |
| $\theta_{\mathrm{w}}$ | m$^3$ m$^{-3}$ | volumetric liquid soil moisture |
| $\theta_{\mathrm{wp}}$ | m$^3$ m$^{-3}$ | volumetric soil moisture at wilting point |
| $\kappa$ | | von Karman constant |
| $\lambda$ | W m$^{-1}$ K$^{-1}$ | heat conductivity |
| $\lambda_{\mathrm{NPP}}$ | | $NPP$ partitioning factor |
| $\lambda_{\mathrm{a}}$ | W m$^{-1}$ K$^{-1}$ | heat conductivity of air |
| $\lambda_{\mathrm{c}}$ | | ratio of intercellular to atmospheric $CO_2$ |
| $\lambda_{\mathrm{dry}}$ | W m$^{-1}$ K$^{-1}$ | heat conductivity of dry soil |
| $\lambda_{\mathrm{i}}$ | W m$^{-1}$ K$^{-1}$ | heat conductivity of ice |
| $\lambda_{\mathrm{s}}$ | W m$^{-1}$ K$^{-1}$ | heat conductivity of soil |
| $\lambda_{\mathrm{s},1}$ | W m$^{-1}$ K$^{-1}$ | heat conductivity of top soil layer |
| $\lambda_{\mathrm{sat}}$ | W m$^{-1}$ K$^{-1}$ | heat conductivity of saturated soil |
| $\lambda_{\mathrm{sn}}$ | W m$^{-1}$ K$^{-1}$ | heat conductivity of snow |
| $\lambda_{\mathrm{w}}$ | W m$^{-1}$ K$^{-1}$ | heat conductivity of water |
| $\mu$ | radians | solar zenith angle |
| $\nu$ | | PFT fractional area coverage |
| $\nu_{\mathrm{seed}}$ | | PFT seed fraction |
| $\rho_{\mathrm{a}}$ | kg m$^{-3}$ | air density |
| $\rho_{\mathrm{i}}$ | kg m$^{-3}$ | density of ice |
| $\rho_{\mathrm{sn}}$ | kg m$^{-3}$ | density of snow |
| $\rho_{\mathrm{sn,fresh}}$ | kg m$^{-3}$ | density of fresh snow |
| $\rho_{\mathrm{sn,min}}$ | kg m$^{-3}$ | minimum density of snow |
| $\rho_{\mathrm{w}}$ | kg m$^{-3}$ | density of liquid water |
| $\sigma$ | W m$^{-2}$ K$^{-4}$ | Stefan-Boltzmann constant |
| $\tau_{\mathrm{fire}}$ | s | fire return time scale |
| $\tau_{\mathrm{s}}$ | s | canopy snow removal time scale |
| $\tau_{\mathrm{w}}$ | s | canopy water removal time scale |
| $\phi$ | | phenology factor |
| $\psi$ | m | soil matric potential |





| | | |
|---|---|---|
| $\psi_{\mathrm{sat}}$ | m | saturated soil matric potential |
| $APAR$ | $\mathrm{mol\,m^{-2}\,day^{-1}}$ | absorbed photosynthetically active radiation |
| $A_{\mathrm{g}}$ | $\mathrm{gC\,m^{-2}\,day^{-1}}$ | daily gross assimilation |
| $A_{\mathrm{n}}$ | $\mathrm{gC\,m^{-2}\,day^{-1}}$ | daily net assimilation |
| $A_{\mathrm{nd}}$ | $\mathrm{gC\,m^{-2}\,day^{-1}}$ | daytime net assimilation |
| $C_{\mathrm{DN}}^{\mathrm{m}}$ | | neutral drag coefficient for momentum |
| $C_{\mathrm{DN}}^{\mathrm{h}}$ | | neutral drag coefficient for heat and water |
| $C_{\mathrm{D}}^{\mathrm{m}}$ | | drag coefficient for momentum |
| $C_{\mathrm{D}}^{\mathrm{h}}$ | | drag coefficient for heat and water |
| $C_{\mathrm{acro}}$ | $\mathrm{kgC\,m^{-2}}$ | acrotelm carbon |
| $C_{\mathrm{acro,crit}}$ | $\mathrm{kgC\,m^{-2}}$ | critical acrotelm carbon for catotelm formation |
| $C_{\mathrm{bare}}$ | | bare soil drag coefficient |
| $C_{\mathrm{can}}$ | | below canopy drag coefficient |
| $C_{\mathrm{cato}}$ | $\mathrm{kgC\,m^{-3}}$ | catotelm carbon density |
| $C_{\mathrm{dense}}$ | | dense canopy drag coefficient |
| $C_{\mathrm{p}}$ | $\mathrm{J\,kg^{-1}\,K^{-1}}$ | specific heat capacity of air at constant pressure |
| $C_{\mathrm{i}}$ | $\mathrm{J\,kg^{-1}\,K^{-1}}$ | specific heat capacity of ice |
| $C_{\mathrm{lit}}$ | $\mathrm{kgC\,m^{-3}}$ | litter carbon density |
| $C_{\mathrm{lit,peat}}$ | $\mathrm{kgC\,m^{-2}}$ | peat litter carbon |
| $C_{\mathrm{fast}}$ | $\mathrm{kgC\,m^{-3}}$ | fast soil carbon density |
| $C_{\mathrm{peat}}$ | $\mathrm{kgC\,m^{-2}}$ | peat carbon |
| $C_{\mathrm{slow}}$ | $\mathrm{kgC\,m^{-3}}$ | slow soil carbon density |
| $C_{\mathrm{v}}$ | $\mathrm{kgC\,m^{-2}}$ | vegetation carbon |
| $C_{\mathrm{v,ag}}$ | $\mathrm{kgC\,m^{-2}}$ | aboveground vegetation carbon |
| $C_{\mathrm{v,high}}$ | $\mathrm{kgC\,m^{-2}}$ | aboveground vegetation carbon parameter for fire |
| $C_{\mathrm{v,l}}$ | $\mathrm{kgC\,m^{-2}}$ | leaf carbon |
| $C_{\mathrm{v,low}}$ | $\mathrm{kgC\,m^{-2}}$ | aboveground vegetation carbon parameter for fire |
| $C_{\mathrm{v,r}}$ | $\mathrm{kgC\,m^{-2}}$ | root carbon |
| $C_{\mathrm{v,s}}$ | $\mathrm{kgC\,m^{-2}}$ | stem carbon |
| $C_{\mathrm{w}}$ | $\mathrm{J\,kg^{-1}\,K^{-1}}$ | specific heat capacity of water |
| $D$ | $\mathrm{m^2\,s^{-1}}$ | vertical soil carbon diffusivity |
| $D_{\mathrm{bio}}$ | $\mathrm{m^2\,s^{-1}}$ | bioturbation carbon diffusivity |
| $D_{\mathrm{cryo}}$ | $\mathrm{m^2\,s^{-1}}$ | cryoturbation carbon diffusivity |
| $E$ | $\mathrm{kg\,m^{-2}\,s^{-1}}$ | evapotranspiration |
| $E_{\mathrm{can}}$ | $\mathrm{kg\,m^{-2}\,s^{-1}}$ | canopy evaporation and sublimation |
| $E_{\mathrm{can}}^{\mathrm{s}}$ | $\mathrm{kg\,m^{-2}\,s^{-1}}$ | canopy sublimation |
| $E_{\mathrm{can}}^{\mathrm{w}}$ | $\mathrm{kg\,m^{-2}\,s^{-1}}$ | canopy evaporation |



| | | |
|---|---|---|
| $E_\mathrm{s}$ | $\mathrm{kg\,m^{-2}\,s^{-1}}$ | snow sublimation |
| $G$ | $\mathrm{W\,m^{-2}}$ | ground heat flux |
| $H$ | $\mathrm{W\,m^{-2}}$ | sensible heat flux |
| $I_\mathrm{can}^\mathrm{s}$ | $\mathrm{kg\,m^{-2}\,s^{-1}}$ | canopy snow interception |
| $I_\mathrm{can}^\mathrm{w}$ | $\mathrm{kg\,m^{-2}\,s^{-1}}$ | canopy rain interception |
| $J_\mathrm{C}$ | $\mathrm{gC\,m^{-2}\,day^{-1}}$ | Rubisco limited photosynthesis rate |
| $J_\mathrm{E}$ | $\mathrm{gC\,m^{-2}\,day^{-1}}$ | light limited photosynthesis rate |
| $K$ | | Kersten number |
| $L$ | $\mathrm{J\,kg^{-1}}$ | latent heat of vaporisation |
| $L_\mathrm{ai}$ | $\mathrm{m^2\,m^{-2}}$ | leaf area index |
| $L_\mathrm{ai,b}$ | $\mathrm{m^2\,m^{-2}}$ | balanced leaf area index |
| $L_\mathrm{f}$ | $\mathrm{J\,kg^{-1}}$ | latent heat of fusion of water |
| $LW^\downarrow$ | $\mathrm{W\,m^{-2}}$ | downward longwave radiation at the surface |
| $LW^\uparrow$ | $\mathrm{W\,m^{-2}}$ | upward longwave radiation at the surface |
| $M_\mathrm{s}$ | $\mathrm{kg\,m^{-2}\,s^{-1}}$ | snowmelt |
| $NPP$ | $\mathrm{kgC\,m^{-2}\,s^{-1}}$ | net primary production |
| $P_\mathrm{s}$ | $\mathrm{kg\,m^{-2}\,s^{-1}}$ | snowfall rate |
| $P_\mathrm{s,g}$ | $\mathrm{kg\,m^{-2}\,s^{-1}}$ | snowfall rate reaching the ground |
| $P_\mathrm{r}$ | $\mathrm{kg\,m^{-2}\,s^{-1}}$ | rainfall rate |
| $P_\mathrm{r,g}$ | $\mathrm{kg\,m^{-2}\,s^{-1}}$ | rainfall rate reaching the ground |
| $R_\mathrm{d}$ | $\mathrm{gC\,m^{-2}\,day^{-1}}$ | leaf respiration |
| $R_\mathrm{i}$ | | bulk Richardson number |
| $R_\mathrm{w}$ | $\mathrm{kg\,m^{-2}\,s^{-1}}$ | surface water runoff |
| $SLA$ | $\mathrm{m^2\,kgC^{-1}}$ | specific leaf area |
| $S_\mathrm{ai}$ | $\mathrm{m^2\,m^{-2}}$ | stem area index |
| $SW^\downarrow$ | $\mathrm{W\,m^{-2}}$ | downward shortwave radiation at the surface |
| $T_0$ | $\mathrm{K}$ | freezing temperature of water |
| $T_\star$ | $\mathrm{K}$ | skin temperature |
| $T_\mathrm{a}$ | $\mathrm{K}$ | air temperature at height $z_\mathrm{ref}$ |
| $T_\mathrm{cmon}^\mathrm{max}$ | $\mathrm{K}$ | maximum coldest month temperature for establishment |
| $T_\mathrm{cmon}^\mathrm{min}$ | $\mathrm{K}$ | minimum coldest month temperature for establishment |
| $T_\mathrm{cmon}^\mathrm{phen}$ | $\mathrm{K}$ | coldest month temperature for phenology |
| $T_\mathrm{gdd}^\mathrm{base}$ | $\mathrm{K}$ | base temperature for phenology |
| $T_\mathrm{s,1}$ | $\mathrm{K}$ | top soil layer or snow temperature |
| $T_\mathrm{s}$ | $\mathrm{K}$ | soil/snow temperature |
| $VPD$ | $\mathrm{kPa}$ | vapor pressure deficit |
| $V_\mathrm{a}$ | $\mathrm{m\,s^{-1}}$ | wind speed at height $z_\mathrm{ref}$ |





| | | |
|---|---|---|
| $V_\mathrm{m}$ | $\mathrm{gC\,m^{-2}\,day^{-1}}$ | maximum daily rate of net photosynthesis |
| $a_\mathrm{C}$ | | factor for leaf respiration |
| $a_\mathrm{wl}$ | $\mathrm{kgC\,m^{-2}}$ | allometric coefficient |
| $a_\mathrm{ws}$ | | ratio of total to respiring stem carbon |
| $b$ | | Clapp and Hornberger parameter |
| $c$ | $\mathrm{J\,m^{-3}\,K{-}1}$ | volumetric heat capacity |
| $c_1$ | $\mathrm{g\,mol^{-1}}$ | factor for light limited assimilation |
| $c_2$ | | factor for Rubisco limited assimilation |
| $c_\mathrm{a}$ | $\mathrm{mol\,mol^{-1}}$ | atmospheric $CO_2$ mole fraction |
| $c_\mathrm{i}$ | $\mathrm{mol\,mol^{-1}}$ | intercellular $CO_2$ mole fraction |
| $c_{ij}$ | | PFT competition coefficients |
| $c_\mathrm{q}$ | $\mathrm{mol\,J^{-1}}$ | conversion factor for solar radiation |
| $c_\mathrm{s}$ | $\mathrm{J\,m^{-3}\,K{-}1}$ | volumetric heat capacity of dry soil |
| $c_\mathrm{sn}$ | $\mathrm{J\,m^{-3}\,K{-}1}$ | volumetric heat capacity of snow |
| $d$ | m | zero plane displacement |
| $d_\mathrm{h}$ | hours | daylength |
| $d_\mathrm{r,1}$ | m | root distribution paramter |
| $d_\mathrm{r,2}$ | m | root distribution parameter |
| $e$ | $\mathrm{kg\,m^{-2}\,s^{-1}}$ | soil moisture removal by evapotranspiration |
| $f_\nabla$ | | parameter for computation of water table depth |
| $f_\theta$ | | soil moisture factor for soil carbon decomposition rate |
| $f_\mathrm{\theta,peat}$ | | soil moisture factor for peat carbon decomposition rate |
| $f_\mathrm{\theta,sat}$ | | soil moisture factor for soil carbon decomposition rate at saturation |
| $f_\mathrm{T}$ | | temperature factor for soil carbon decomposition rate |
| $f_\mu$ | | solar zenith angle factor for snow albedo |
| $f_\mathrm{age}$ | | snow age factor |
| $f_\mathrm{frz,crit}$ | | critical fraction of frozen soil water for permafrost carbon |
| $f_\mathrm{ice}$ | | fraction of grid cell covered by ice sheets |
| $f_\mathrm{inert}$ | | frozen soil factor for soil carbon decomposition |
| $f_\mathrm{inun}$ | | inundated grid cell fraction |
| $f_\mathrm{lit}^\mathrm{resp}$ | | fraction of decomposed litter carbon going to atmosphere |
| $f_\mathrm{lit \to fast}$ | | fraction of decomposed litter transfered to fast carbon pool |
| $f_\mathrm{lit \to slow}$ | | fraction of decomposed litter transfered to slow carbon pool |
| $f_\mathrm{org}$ | | organic soil fraction |
| $f_\mathrm{oxic}$ | | fraction of litter and acrotelm respiring in oxic conditions |
| $f_\mathrm{peat}$ | | peatland fraction |
| $f_\mathrm{sat}$ | | saturated grid cell fraction |





| | | |
|---|---|---|
| $f_{\mathrm{sat}}^{\max}$ | | maximum saturated grid cell fraction |
| $f_{\mathrm{shelf}}$ | | fraction of grid cell below sea level |
| $f_{\mathrm{sn}}$ | | snow fraction |
| $f_{\mathrm{sn}}^{\mathrm{can}}$ | | canopy snow fraction |
| $f_{\mathrm{sv}}$ | | sky view factor |
| $f_{\mathrm{sv}}^{\mathrm{dir}}$ | | direct beam sky view factor |
| $f_{\mathrm{sv}}^{\mathrm{dif}}$ | | diffuse radiation sky view factor |
| $f_{\mathrm{veg}\to\mathrm{bur}}$ | | fraction of vegetation carbon buried below ice sheets |
| $f_{\mathrm{wet}}$ | | wetland fraction |
| $g$ | $\mathrm{m\,s^{-2}}$ | gravitational acceleration |
| $gdd$ | K | growing degree days above $T_{\mathrm{gdd}}^{\mathrm{base}}$ |
| $gdd_{\mathrm{crit}}$ | K | critical growing degree days for phenology |
| $gdd_{\mathrm{min}}$ | K | minimum growing degree days for establishment |
| $g_1$ | | parameter in optimal stomatal conductance model |
| $g_{\mathrm{can}}$ | $\mathrm{m\,s^{-1}}$ | canopy conductance |
| $g_{\mathrm{min}}$ | $\mathrm{m\,s^{-1}}$ | minimum canopy conductance |
| $h_{\mathrm{sn}}$ | m | snow thickness |
| $h_{\mathrm{soil}}$ | m | depth of the soil column |
| $h_{\mathrm{v}}$ | m | vegetation height |
| $k$ | $\mathrm{kg\,m^{-2}\,s^{-1}}$ | hydraulic soil conductivity |
| $k_{\mathrm{acro}}$ | $\mathrm{s^{-1}}$ | acrotelm carbon turnover rate |
| $k_{\mathrm{acro}\to\mathrm{cato}}$ | $\mathrm{s^{-1}}$ | catotelm formation rate |
| $k_{\mathrm{cato}}$ | $\mathrm{s^{-1}}$ | catotelm carbon turnover rate |
| $k_{\mathrm{ext}}$ | | extinction coefficient for radiation |
| $k_{\mathrm{fast}}$ | $\mathrm{s^{-1}}$ | fast carbon turnover rate |
| $k_{\mathrm{fast},10}$ | $\mathrm{s^{-1}}$ | fast soil carbon turnover rate at $10\,^{\circ}\mathrm{C}$ |
| $k_{\mathrm{inert}}$ | $\mathrm{s^{-1}}$ | inert soil carbon turnover rate |
| $k_{\mathrm{lit}}$ | $\mathrm{s^{-1}}$ | litter carbon turnover rate |
| $k_{\mathrm{lit},10}$ | $\mathrm{s^{-1}}$ | litter carbon turnover rate at $10\,^{\circ}\mathrm{C}$ |
| $k_{\mathrm{lit,peat}}$ | $\mathrm{s^{-1}}$ | peat litter carbon turnover rate |
| $k_{\mathrm{sat}}$ | $\mathrm{kg\,m^{-2}\,s^{-1}}$ | hydraulic soil conductivity at saturation |
| $k_{\mathrm{slow}}$ | $\mathrm{s^{-1}}$ | slow soil carbon turnover rate |
| $k_{\mathrm{slow},10}$ | $\mathrm{s^{-1}}$ | slow soil carbon turnover rate at $10\,^{\circ}\mathrm{C}$ |
| $k_{\rho}$ | $\mathrm{m^3\,kg^{-1}}$ | factor for density dependence of snow viscosity |
| $k_{\mathrm{T}}$ | $\mathrm{K^{-1}}$ | factor for temperature dependence of snow viscosity |
| $n_{\mathrm{al}}$ | | multiple of active layer thickness for cryoturbation |
| $p_{\mathrm{a}}$ | Pa | partial pressure of atmospheric $CO_2$ |





| | | |
|---|---|---|
| $p_i$ | Pa | partial pressure of intercellular $CO_2$ |
| $q$ | $kg\,m^{-2}\,s^{-1}$ | soil water flux |
| $q_a$ | $kg\,kg^{-1}$ | air specific humidity at height $z_{ref}$ |
| $q_{drain}$ | $kg\,m^{-2}\,s^{-1}$ | soil water drainage |
| $q_{inf}$ | $kg\,m^{-2}\,s^{-1}$ | soil water infiltration |
| $q_{inf}^{max}$ | $kg\,m^{-2}\,s^{-1}$ | maximum soil water infiltration |
| $q_{sat}$ | $kg\,kg^{-1}$ | specific humidity at saturation |
| $r$ | | cumulative root fraction |
| $r_a$ | $s\,m^{-1}$ | aerodynamic resistance |
| $r_{a,can}$ | $s\,m^{-1}$ | below-canopy aerodynamic resistance |
| $r_s$ | $s\,m^{-1}$ | surface resistance to water vapor flux |
| $u_\star$ | $m\,s^{-1}$ | friction velocity |
| $w_{can}^{w}$ | $kg\,m^{-2}$ | canopy liquid water |
| $w_{can}^{s}$ | $kg\,m^{-2}$ | canopy snow water equvalent |
| $w_i$ | $kg\,m^{-2}$ | soil frozen water content |
| $w_{sn}$ | $kg\,m^{-2}$ | snow water equivalent |
| $w_w$ | $kg\,m^{-2}$ | soil liquid water content |
| $w_w^{max}$ | $kg\,m^{-2}$ | maximum soil liquid water content |
| $z_0^{b}$ | m | bare soil roughness length |
| $z_0^{i}$ | m | ice roughness length |
| $z_0^{sn}$ | m | snow roughness length |
| $z_0^{snfree}$ | m | snow-free roughness length |
| $z_0^{v}$ | m | vegetation roughness length |
| $z_0^{w}$ | m | water roughness length |
| $z_\nabla$ | m | grid cell mean water table depth |
| $z_\nabla^{min}$ | m | minimum water table depth |
| $z_\nabla^{peat}$ | m | peatland water table depth |
| $z_{al}$ | m | active layer thickness |
| $z_{acro}$ | m | acrotelm thickness |
| $z_m$ | m | roughness length for momentum |
| $z_h$ | m | roughness length for scalars |
| $z_{ref}$ | m | reference height |





**Table 2.** Surface model parameters.

| | |
|---|---|
| $k_{\mathrm{ext}} = 0.5$ | extinction coefficient for radiation |
| $\alpha_{\mathrm{sn,fresh}}^{\mathrm{vis,dif}} = 0.95$ | diffuse visible fresh snow albedo |
| $\alpha_{\mathrm{sn,fresh}}^{\mathrm{nir,dif}} = 0.65$ | diffuse near infrared fresh snow albedo |
| $z_0^{\mathrm{b}} = 0.005\,\mathrm{m}$ | bare soil roughness length |
| $z_0^{\mathrm{i}} = 0.01\,\mathrm{m}$ | ice roughness length |
| $z_0^{\mathrm{w}} = 0.001\,\mathrm{m}$ | water roughness length |
| $z_0^{\mathrm{sn}} = 0.0024\,\mathrm{m}$ | snow roughness length |
| $C_{\mathrm{bare}} = 0.05$ | bare soil drag coefficient |
| $C_{\mathrm{dense}} = 0.005$ | dense canopy drag coefficient |
| $\alpha_{\mathrm{int}}^{\mathrm{w}} = 0.2$ | canopy water interception parameter |
| $\alpha_{\mathrm{int}}^{\mathrm{s}} = 0.5$ | canopy snow interception parameter |
| $\tau_{\mathrm{w}} = 1\,\mathrm{day}$ | canopy water removal time scale |
| $\tau_{\mathrm{s}} = 10\,\mathrm{days}$ | canopy snow removal time scale |
| $\rho_{\mathrm{sn,min}} = 50\,\mathrm{kg\,m^{-3}}$ | minimum snow density |
| $\eta_0 = 9 \times 10^6\,\mathrm{Pa\,s}$ | reference snow viscosity |
| $k_{\mathrm{T}} = 0.06\,\mathrm{K^{-1}}$ | temperature parameter for snow viscosity |
| $k_{\rho} = 0.02\,\mathrm{m^3\,kg^{-1}}$ | density parameter for snow viscosity |
| $f_{\nabla} = 1.6$ | parameter for saturated grid cell fraction |

**Table 3.** Soil model parameters.

| | |
|---|---|
| $c_{\mathrm{s}} = 2.3 \times 10^6\,\mathrm{J\,m^{-3}\,K^{-1}}$ | volumetric heat capacity of soil |
| $\lambda_{\mathrm{s}} = 5.0\,\mathrm{W\,m^{-1}\,K^{-1}}$ | soil heat conductivity at saturation |
| $\lambda_{\mathrm{dry}} = 0.2\,\mathrm{W\,m^{-1}\,K^{-1}}$ | dry soil heat conductivity |
| $\theta_{\mathrm{sat}} = 0.43\,\mathrm{m^3\,m^{-3}}$ | soil porosity |
| $\theta_{\mathrm{fc}} = 0.25\,\mathrm{m^3\,m^{-3}}$ | volumetric soil moisture at field capacity |
| $\theta_{\mathrm{wp}} = 0.14\,\mathrm{m^3\,m^{-3}}$ | volumetric soil moisture at wilting point |
| $\psi_{\mathrm{sat}} = -0.2\,\mathrm{m}$ | soil matric potential at saturation |
| $k_{\mathrm{sat}} = 520\,\mathrm{kg\,m^{-2}\,day^{-1}}$ | soil hydraulic conductivity at saturation |
| $b = 6$ | Clapp-Hornberger parameter |





**Table 4.** Photosynthesis model parameters (Sitch et al., 2003).

| | | |
|---|---|---|
| $\theta_r$ | 0.7 | co-limitation parameter |
| $\alpha_{leaf}$ | 0.17 | leaf albedo in PAR range |
| $\alpha$ | 0.5 | fraction of PAR assimilated at ecosystem level |
| $c_q$ | $4.6 \times 10^{-6}\,\mathrm{mol\,J^{-1}}$ | conversion factor for solar radiation at $550\,\mathrm{nm}$ |
| $a_{C3}$ | 0.015 | leaf respiration as a fraction of Rubisco capacity in C3 plants |
| $a_{C4}$ | 0.02 | leaf respiration as a fraction of Rubisco capacity in C4 plants |
| $\alpha_{C3}$ | 0.08 | intrinsic quantum efficiency of $CO_2$ uptake in C3 plants |
| $\alpha_{C4}$ | 0.053 | intrinsic quantum efficiency of $CO_2$ uptake in C4 plants |
| $[O_2]$ | $20.9\,\mathrm{kPa}$ | $O_2$ partial pressure |
| $C_{mass}$ | 12 | atomic mass of carbon |

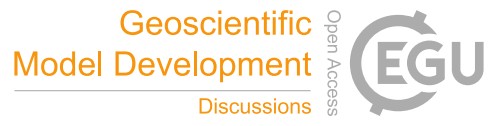

**Table 5.** Plant functional type specific model parameters.

| | | Broadleaf tree | Needleleaf tree | C3 grass | C4 grass | Shrub |
|---|---|---|---|---|---|---|
| $d_{r,1}$ | root distribution parameter (Oleson et al., 2013) | 6.5 | 7.0 | 11.0 | 11.0 | 7.0 |
| $d_{r,2}$ | root distribution parameter (Oleson et al., 2013) | 1.5 | 2.0 | 2.0 | 2.0 | 1.5 |
| $\alpha_{snfree}^{can,vis,dir}$ | snowfree visible canopy albedo for direct radiation (Houldcroft et al., 2009) | 0.011 | 0.004 | 0.038 | 0.033 | 0.035 |
| $\alpha_{snfree}^{can,vis,dif}$ | snowfree visible canopy albedo for diffuse radiation (Houldcroft et al., 2009) | 0.013 | 0.005 | 0.043 | 0.036 | 0.037 |
| $\alpha_{snfree}^{can,nir,dir}$ | snowfree near infrared canopy albedo for direct radiation (Houldcroft et al., 2009) | 0.22 | 0.141 | 0.269 | 0.244 | 0.173 |
| $\alpha_{snfree}^{can,nir,dif}$ | snowfree near infrared canopy albedo for diffuse radiation (Houldcroft et al., 2009) | 0.256 | 0.154 | 0.306 | 0.275 | 0.185 |
| $\alpha_{sn}^{can,vis,dir/dif}$ | snowfree visible canopy albedo (Moody et al., 2007) | 0.44 | 0.31 | 0.70 | 0.70 | 0.55 |
| $\alpha_{sn}^{can,nir,dir/dif}$ | snowfree near infrared canopy albedo (Moody et al., 2007) | 0.33 | 0.24 | 0.48 | 0.48 | 0.37 |
| $T_{cmon}^{min}\,[°C]$ | minimum coldest month temperature for establishment (Sitch et al., 2003) | -17.0 | - | - | 15.5 | - |
| $T_{cmon}^{max}\,[°C]$ | maximum coldest month temperature for establishment (Sitch et al., 2003) | - | -2.0 | 15.5 | - | - |
| $gdd_{min}\,[°C]$ | minimum gdd for establishment (Sitch et al., 2003) | 1200 | 350 | 0 | 0 | 0 |
| $T_{cmon}^{phen}\,[°C]$ | coldest month temperature for phenology | 5.0 | -999 | 0.0 | 0.0 | -999 |
| $T_{base}^{gdd}\,[°C]$ | base temperature for gdd (Sitch et al., 2003) | 5.0 | 2.0 | 2.0 | 5.0 | 2.0 |
| $gdd_{crit}\,[°C]$ | gdd for full phenology (Sitch et al., 2003) | 300 | - | 100 | 100 | - |
| $g_{min}\,[\mathrm{mm\,s^{-1}}]$ | minimum canopy conductance (Sitch et al., 2003) | 0.5 | 0.3 | 0.5 | 0.5 | 0.5 |
| $g_1$ | parameter in optimal stomatal conductance formulation (Lin et al., 2015) | 4.0 | 2.3 | 3 | 1.6 | 4.0 |
| $L_{ai}^{min}\,[\mathrm{m^2\,m^{-2}}]$ | minimum leaf area index modified from Clark et al. (2011) | 1.0 | 1.0 | 0.1 | 0.1 | 1.0 |
| $L_{ai}^{max}\,[\mathrm{m^2\,m^{-2}}]$ | maximum leaf area index modified from Clark et al. (2011) | 9.0 | 7.0 | 4.0 | 4.0 | 3.0 |
| $SLA\,[\mathrm{m^{-2}\,kgC^{-1}}]$ | specific leaf area (Kattge et al., 2011) | 12.5 | 6 | 20 | 20 | 12.5 |
| $\gamma_l\,[\mathrm{yr^{-1}}]$ | leaf turnover rate (Kattge et al., 2011) | 0.5 | 0.3 | 1.0 | 1.0 | 0.5 |
| $\gamma_r\,[\mathrm{yr^{-1}}]$ | root turnover rate | 0.5 | 0.3 | 0.5 | 0.5 | 0.5 |
| $\gamma_s\,[\mathrm{yr^{-1}}]$ | stem turnover rate modified from Clark et al. (2011) | 0.005 | 0.005 | 0.2 | 0.2 | 0.1 |
| $a_{wl}\,[\mathrm{kgC\,m^{-2}}]$ | allometric coefficient (Clark et al., 2011) | 0.65 | 0.75 | 0.005 | 0.005 | 0.1 |
| $a_{ws}$ | ratio of total to respiring stem carbon (Cox, 2001) | 10 | 10 | 1 | 1 | 5 |



**Table 6.** Dynamic vegetation model parameters.

| | |
|---|---|
| $\nu_{\text{seed}} = 0.001$ | vegetation seed fraction |
| $\gamma_{\text{v,min}} = 0.002\,\text{yr}^{-1}$ | minimum vegetation disturbance rate |
| $\tau_{\text{fire}} = 10\,\text{yr}$ | fire return time scale |
| $\theta_{\text{crit}} = 0.15\,\text{m}^3\,\text{m}^{-3}$ | critical soil moisture for fire disturbance |
| $C_{\text{v,low}} = 0.2\,\text{kgC}\,\text{m}^{-2}$ | minimum aboveground vegetation carbon for fire disturbance |
| $C_{\text{v,high}} = 1.0\,\text{kgC}\,\text{m}^{-2}$ | maximum aboveground vegetation carbon for fire disturbance |

**Table 7.** Soil carbon model parameters.

| | |
|---|---|
| $f_{\text{lit}}^{\text{resp}} = 0.7$ | fraction of decomposed litter carbon going to atmosphere (Sitch et al., 2003) |
| $f_{\text{lit}\rightarrow\text{fast}} = 0.985$ | fraction of decomposed litter transfered to fast carbon pool (Sitch et al., 2003) |
| $f_{\text{lit}\rightarrow\text{slow}} = 0.015$ | fraction of decomposed litter transfered to slow carbon pool (Sitch et al., 2003) |
| $D_{\text{bio}} = 1\times 10^{-4}\,\text{m}^2\,\text{year}^{-1}$ | bioturbation rate (Braakhekke et al., 2011) |
| $D_{\text{cryo}} = 5\times 10^{-4}\,\text{m}^2\,\text{year}^{-1}$ | cryoturbation rate (Koven et al., 2009, 2013) |
| $k_{\text{lit},10} = 2.86\,\text{yr}^{-1}$ | litter carbon turnover rate at $10\,^\circ\text{C}$ (Sitch et al., 2003) |
| $k_{\text{fast},10} = 33.3\,\text{yr}^{-1}$ | fast soil carbon turnover rate at $10\,^\circ\text{C}$ (Sitch et al., 2003) |
| $k_{\text{slow},10} = 1000\,\text{yr}^{-1}$ | slow soil carbon turnover rate at $10\,^\circ\text{C}$ (Sitch et al., 2003) |
| $k_{\text{acro},10} = 30\,\text{yr}^{-1}$ | acrotelm carbon turnover rate at $10\,^\circ\text{C}$ |
| $k_{\text{cato},10} = 1000\,\text{yr}^{-1}$ | catotelm carbon turnover rate at $10\,^\circ\text{C}$ (Spahni et al., 2013) |
| $k_{\text{acro}\rightarrow\text{cato}} = 15\times 10^{-3}\,\text{yr}^{-1}$ | catotlem formation rate (Wania et al., 2009; Kleinen et al., 2012) |
| $f_{\theta,\text{peat}} = 0.3$ | soil moisture factor for peat carbon decomposition rate at saturation (Wania et al., 2009) |
| $\rho_{\text{acro}} = 20\,\text{kgC}\,\text{m}^{-3}$ | acrotelm carbon density (Clymo, 1984; R. S. Clymo, 1998) |
| $\rho_{\text{cato}} = 50\,\text{kgC}\,\text{m}^{-3}$ | catotelm carbon density (Turunen et al., 2002; Malmer and Wallén, 2004) |
| $\rho_{\text{soc,max}} = \rho_{\text{cato}}$ | maximum carbon density of soil organic carbon |
| $C_{\text{acro,crit}} = 5\,\text{kgC}\,\text{m}^{-2}$ | minimum acrotelm carbon content for catotelm formation (Wania et al., 2009) |
| $C_{\text{peat}}^{\text{crit}} = 50\,\text{kgC}\,\text{m}^{-2}$ | minimum peat carbon content for peat survival (Stocker et al., 2014) |
| $\left.\frac{\text{d}C_{\text{peat}}}{\text{d}t}\right|_{\text{crit}} = 10\,\text{kgC}\,\text{m}^{-2}\,\text{yr}^{-1}$ | minimum peat carbon uptake for peat survival (Stocker et al., 2014) |
| $f_{\text{CH}_4\text{:C}}^{\text{wet}} = 0.07$ | fraction of carbon respired as methane from wetlands () |
| $f_{\text{CH}_4\text{:C}}^{\text{peat}} = 0.2$ | fraction of carbon respired as methane from peatlands (Spahni et al., 2013) |



**Table 8.** Global values of relevant model quantities over the time period 1981–2010 compared to observation based estimates.

| | Model | Observation based estimates |
|---|---|---|
| evapotranspiration [$\times 10^{15}$ kg yr$^{-1}$] | 73 | 64–73 (Mueller et al., 2013; Trenberth et al., 2007) |
| runoff [kg yr$^{-1}$] | 35 | 38–40 (Fekete et al., 2002; Baumgartner and Reichel, 1975) |
| permafrost area [mln km$^2$] | 16.5 | 13–18 (Gruber, 2012) |
| GPP [PgC yr$^{-1}$] | 126 | 115–131 (Beer et al., 2010) |
| NPP [PgC yr$^{-1}$] | 65 | 42–70 (Ito, 2011) |
| vegetation carbon [PgC] | 500 | 470–650 (Prentice et al., 2001) |
| top meter soil carbon [PgC] | 1030 | 890–1660 (Todd-Brown et al., 2013) |
| soil carbon in permafrost area [PgC] | 590 | 1100–1500 (Hugelius et al., 2014) |
| northern peat carbon [PgC] | 520 | 470–620 (Yu et al., 2010) |
| maximum monthly wetland area [mln km$^2$] | 5.7 | 5 (Prigent et al., 2007; Papa et al., 2010) |
| northern peatland area [mln km$^2$] | 2.9 | 3.6–4 (Tarnocai et al., 2009; Yu et al., 2010) |
| total CH$_4$ emissions [TgCH$_4$ yr$^{-1}$] | 175 | 115–215 (Bloom et al., 2010; Bousquet et al., 2006) |
| tropical CH$_4$ emissions [TgCH$_4$ yr$^{-1}$] | 105 | 63–119 (Bloom et al., 2010; Bousquet et al., 2006) |
| extratropical CH$_4$ emissions [TgCH$_4$ yr$^{-1}$] | 70 | 39–89 (Bloom et al., 2010; Bousquet et al., 2006) |

**Table 9.** Climate forcing fields needed to run PALADYN in offline mode.

| |
|---|
| Surface air temperature |
| Surface air specific humidity |
| Downwelling shortwave radiation at the surface |
| Downwelling longwave radiation at the surface |
| Rainfall |
| Snowfall |
| Wind speed |
| Surface pressure |