# Peer review of "PALADYN v1.0, a comprehensive land surface-vegetation-carbon cycle model of intermediate complexity"

_Geoscientific Model Development, 2016_

## Referee Comment (RC1) · Anonymous Referee #1 · 31 May 2016

The authors developed a comprehensive land model of intermediate complexity for long-term simulations and paleoclimate studies. The model descriptions provided enough details for readers to understand the model, and the overall global offline model evaluations using a variety of datasets demonstrated the adequate performance of the model. This manuscript can be accepted after addressing the following concerns.

Specific comments/suggestions:

L186: Does eq. (5) converges to eq. (4) (bare soil) when canopies (e.g., LAI+SAI approach zero) disappear? It is also unclear how the stomatal resistance rs is dependent on LAI in the model formulations Eqs. (33), (69), (70).

L432: The effects of snow metamorphism and snow melting on snow density are neglected in the model. Is this the main reason for the model's deficiency in the snow simulation (e.g., Figs. 13-15)?

L753: While the equilibrium spin-up mode is fine, the authors should at least test it against the actual spin-up for "at least 10,000 years" (say, 50,000 years). If global testing is too time-consuming, the authors could pick up a few model boxes over different climate regimes for the test.

L834: For biogeochemistry, the authors should compare the model LAI with MODIS LAI (e.g., the seasonal cycle over different latitudes with limited crop coverages). After all, MODIS LAI is one of the most reliable vegetation datasets.

L845: The value of comparing potential vegetation from one model to another in Fig. 19 seems to be limited. It may be more useful to do a comparison similar to that in Zeng et al. (2008, e.g., their Fig. 10).

L762 (Section 8: Evaluation): The authors did an excellent job in using comprehensive datasets in model evaluations. However, most of the discussions were quite qualitative. The authors should compute some simple quantities (e.g., root mean square difference, correlation coefficient, mean bias, . . .) for some of the comparisons for two purposes: to back up the qualitative statements, and for other groups to compare their models' performance against the PALADYN v1.0 model in the future. For instance, there seems to be years with opposite anomaly signs in Fig. 18 (L840), and the authors should at least compute the simple correlation coefficient to quantify the agreement.

Minor comments:

L52: It is appropriate to cite Dai et al. (2013) here

L418: Where and when does "frozen water runoff" occur?

L815: Another reason is the assumption of global uniform soil porosity in the model.

L820: The agreement of wetland areas between the model and multi-satellite data is not very good in spatial distribution (Fig. 11). Please comment.

L852: Explain "flux weighted discrimation"

Table 5: Canopy diffuse snow-free albedos (0.005 for vis and 0.154 for nir) for needle-leaf trees seem to be too small. In addition, does the model consider evergreen versus deciduous trees for LAImin = 1 and LAImax = 9 (broadleaf) or 7 (needleleaf)?

References:

Dai, Y., and 11 coauthors, 2003: The Common Land Model. Bull. Amer. Meteor. Soc., 84, 1013-1023.

Zeng, X. D., X. Zeng, and M. Barlage, 2008: Growing temperate shrubs over arid and semiarid regions in the Community Land Model–Dynamic Global Vegetation Model, Global Biogeochem. Cycles, 22, GB3003, doi:10.1029/2007GB003014

---

## Referee Comment (RC2) · Anonymous Referee #2 · 20 Jul 2016

Comments on "PALADYN 1.0, a comprehensive land surface-vegetation-carbon cycle model of intermediate complexity" submitted by Matteo Willeit and Andrey Ganopolski to Geoscientific Model Development

General comments

In this manuscript, the authors presented a new integrated terrestrial model, PALA-DYN, which includes major physical and biogeochemical processes at an intermediate complexity. The model was developed on the basis of previous models such as LPJ and TRIFFID but includes several recent findings such as new stomatal conductance model. Although many terrestrial models for similar purposes have been developed, this model has several unique and intriguing features. In particular, inclusion of peat

land and permafrost carbon dynamics will allow the model to simulate long-term (e.g., glacial time scale) simulations as proposed by the authors. This manuscript includes more than hundred equations, many tables, and schematic diagrams to fully describe the model.

To demonstrate the model performance, the authors compared major terrestrial variables with contemporary observational datasets. Overall, these results show good performance of the model, but the authors provided only very brief explanations. Although I agree that scientific insights are not necessarily included into the manuscript, I recommend clarifying the characteristics of the PALADYN model, especially in comparison with other models. Therefore, I conclude that the manuscript needs minor revision before acceptance for publication. Please look my specific comments for details.

Specific comments

Line 103: It seems that the model don't have a separate type for crops. Do you have an idea to include croplands to account for agriculture?

Line 171: In Eq.(2), the symbol sigma seems to represent Stefan-Boltzman constant, but no definition was provided.

Line 286: Vegetation height hv is later estimated by Eq. (81). It is helpful for readers to explain this at this point.

Line 397: In Eq. (42), what kind of phenomenon does the last turnover term represent? Stem flow?

Line 456: It seems that Figure 2 does not include the surface runoff Rw. Can you include Rw into Figure 2?

Line 566: Is this the single-sided (or projected) specific leaf area?

Line 606: The statement is at least partially incorrect. In East Siberia, a broad area of forest is dominated by larch, a deciduous needleleaf species.

Line 745: How did you determine the stable and radiocarbon isotope ratios of the atmospheric CO2? As you know, it has been changed by the Suess effect.

Line 772: "Manua" should be replaced by "Mauna".

Line 821: In terms of wetland extent, the model estimate seems to underestimate in Southeast Asia. I guess that the GIEMS data includes a substantial fraction of paddy fields. Is it correct?

Line 852: Can you say something about the simulated discrimination in relation to C3 and C4 plant distribution? Do you confirm that distribution and functional contribution of C3 and C4 plants were reasonably simulated?

---

## Author Response (AR1)

Dear editor,

We would like to thank the reviewers for their constructive comments on our paper. We addressed all their comments and replied to them point by point below.
A marked-up manuscript version showing the changes made during the revision process is also attached below.

We hope that the manuscript is now acceptable for publication in Geoscientific Model Development.

Kind regards,
Matteo Willeit and Andrey Ganopolski

**Response to reviewer #1**

*The authors developed a comprehensive land model of intermediate complexity for long-term simulations and paleoclimate studies. The model descriptions provided enough details for readers to understand the model, and the overall global offline model evaluations using a variety of datasets demonstrated the adequate performance of the model. This manuscript can be accepted after addressing the following concerns.*

*Specific comments/suggestions:*

*L186: Does eq. (5) converges to eq. (4) (bare soil) when canopies (e.g., LAI+SAI approach zero) disappear? It is also unclear how the stomatal resistance rs is dependent on LAI in the model formulations Eqs. (33), (69), (70).*

The dependence of $r_s$ on LAI is implicit in the dependence of $g_{can}$ on carbon assimilation, which depends on the absorbed photosynthetically active radiation, which in turn depends on LAI (Eq. (63)). What was not mentioned in the text was that $g_{min}$, the minimum canopy conductance, depends on LAI (and soil moisture). This has now been added to the text.
If LAI tends to zero then rs tends to infinity and therefore the first term on the rhs of Eq. (5) tends to zero. The canopy evaporation term (last term in Eq. (5)) also tends to zero when LAI tends to zero. The only term remaining when LAI+SAI approach zero is the term representing evaporation from the ground below the canopy (second term on the rhs). In the original formulation of Eq. (5) this term does not converge to Eq. (4) when LAI+SAI are becoming small because the undercanopy resistance $r_{a,can}$ remains finite. This has been changed by introducing a simple LAI+SAI dependence in the $r_{a,can}$ formulation (Eq. (30)):
$$r_{a,can} = \frac{L_{ai} + S_{ai}}{C_{can}u_*}.$$
There only difference that remains between Eq, (5) and Eq. (4) in the limit LAI+SAI approaching zero is the use of two different temperatures (skin and ground) in the computation of the saturated specific humidity. However, when vegetation is disappearing, the difference between the skin temperature and the top soil layer temperature becomes negligible.

*L432: The effects of snow metamorphism and snow melting on snow density are neglected in the model. Is this the main reason for the model's deficiency in the snow simulation (e.g., Figs. 13-15)?*

The model performance in simulating total snow mass is comparable to the performance of state-of-the-art CMIP5 models, which also tend to melt snow too late in spring compared to the GlobSnow dataset (see Figure 5 in Shi and Wang (2015) and http://www.earsel.org/SIG/Snow-Ice/files/oral_ws2014/Luojus_2014_EARSeL_CMIP5.pdf). This is now discussed in the text.

The main limitation of our model is probably that it includes a single snow layer, which makes it in general more inert to fast changes in atmospheric conditions.

*L753: While the equilibrium spin-up mode is fine, the authors should at least test it against the actual spin-up for "at least 10,000 years" (say, 50,000 years). If global testing is too time-consuming, the authors could pick up a few model boxes over different climate regimes for the test.*

The idea of introducing an equilibrium spin-up mode in the model was mainly to have a very efficient way to bring the model to a quasi-equilibrium state to allow fast testing and tuning of the model. For all practical applications the spin-up mode is of limited use since an additional transient model simulation is anyway required to bring the model in equilibrium with the fast seasonal processes. Also, because the model integrates one year in approximately one second, a proper transient spinup of 10,000 years can be achieved in only a few hours.

In the simulations presented in the paper we therefore slightly changed the experimental setup by removing the equilibrium spinup phase and substituting it with a 30,000 years transient spinup simulation.

***L834: For biogeochemistry, the authors should compare the model LAI with MODIS LAI (e.g., the seasonal cycle over different latitudes with limited crop coverages). After all, MODIS LAI is one of the most reliable vegetation datasets.***

We added a comparison of modelled LAI with MODIS LAI as suggested by the reviewer. We included a figure comparing annual maximum LAI and a figure comparing the zonal mean seasonal LAI variations for different latitudinal bands.

***L845: The value of comparing potential vegetation from one model to another in Fig. 19 seems to be limited. It may be more useful to do a comparison similar to that in Zeng et al. (2008, e.g., their Fig. 10).***

In addition to Figure 19 we included also a figure similar to Fig. (10) in Zeng et al. (2008) in the revised version of the paper. In the new figure modelled PFT coverage as a function of annual precipitation is compared to MODIS land cover data.

***L762 (Section 8: Evaluation): The authors did an excellent job in using comprehensive datasets in model evaluations. However, most of the discussions were quite qualitative. The authors should compute some simple quantities (e.g., root mean square difference, correlation coefficient, mean bias, : : :) for some of the comparisons for two purposes: to back up the qualitative statements, and for other groups to compare their models' performance against the PALADYN v1.0 model in the future. For instance, there seems to be years with opposite anomaly signs in Fig. 18 (L840), and the authors should at least compute the simple correlation coefficient to quantify the agreement.***

As suggested by the reviewer we computed root mean square differences and correlations for some of the model-data comparisons (in particular for global maps of different quantities) and reported the values directly on the figures to allow the reader to get a quick quantitative measure of model performance.

Additionally, the discussion of the evaluation results has been extended to include more quantitative analyses.

***Minor comments:***

***L52: It is appropriate to cite Dai et al. (2013) here***

It is not clear how a citation to Dai et al. (2003) would fit into the mentioned sentence. Dai et al. (2003) give an overview of the CLM land surface model, which uses the Ball model to link leaf photosynthesis and stomatal conductance. The Ball model is cited in the sentence, but CLM is just one of many models using the Ball model and we think that a citation to these is not appropriate here.

*L418: Where and when does "frozen water runoff" occur?*

Frozen water runoff occurs if the snow water equivalent of the snow layer exceeds 1000 kg/m2, as mentioned in the text. In practice in the model simulation for the 20$^{th}$ century it is limited to areas around Greenland, where annual snow accumulation exceeds snow melt.

*L815: Another reason is the assumption of global uniform soil porosity in the model.*

This is now mentioned in the text.

*L820: The agreement of wetland areas between the model and multi-satellite data is not very good in spatial distribution (Fig. 11). Please comment.*

The paragraph where wetland extent is evaluated has been extended including further discussion: *"The mean annual simulated wetland area is 3.2 mln km2 . Maximum monthly wetland extent is reasonably well captured by the model (Fig. 11). Compared to the multi-satellite product from GIEMS (Prigent et al., 2007; Papa et al., 2010) the model simulates larger wetland extent in tropical forest areas and northern peatland areas. However, if compared to other wetland products based on data other than from satellite, GIEMS is underestimating wetlands below dense forests (e.g. the Amazon forest (Melack and Hess, 2010)) and in peatland regions of northern Canada and Eastern Siberia (Stocker et al., 2014). In south-east Asia, the GIEMS wetland extent also includes extensive rice cultivation areas, which are not represented in the model. The modelled seasonal variation in global wetland area is in very good agreement with GIEMS (Fig. 12)."*

*L852: Explain "flux weighted discrimation"*

Has been replaced by: *"GPP-weighted isotopic discrimination during photosynthesis".*

*Table 5: Canopy diffuse snow-free albedos (0.005 for vis and 0.154 for nir) for needleleaf trees seem to be too small. In addition, does the model consider evergreen versus deciduous trees for LAImin = 1 and LAImax = 9 (broadleaf) or 7 (needleleaf)?*

The diffuse canopy albedo values are taken from Houldcroft et al. (2009) and are based on MODIS data. The values seem also to be in agreement with site level observations reported by Betts and Ball (1997).
All PFT specific parameters listed in Table 5 are the same for evergreen and deciduous trees.

*References:*
*Dai, Y., and 11 coauthors, 2003: The Common Land Model. Bull. Amer. Meteor. Soc.,*
*84, 1013-1023.*
*Zeng, X. D., X. Zeng, and M. Barlage, 2008: Growing temperate shrubs over arid and*
*semiarid regions in the Community Land Model–Dynamic Global Vegetation*

**References:**

Shi, H. X., & Wang, C. H. (2015). Projected 21st century changes in snow water equivalent over Northern Hemisphere landmasses from the CMIP5 model ensemble. *Cryosphere*, *9*(5), 1943–1953. http://doi.org/10.5194/tc-9-1943-2015

Houldcroft, C. J., Grey, W. M. F., Barnsley, M., Taylor, C. M., Los, S. O., & North, P. R. J. (2009). New Vegetation Albedo Parameters and Global Fields of Soil Background Albedo Derived from MODIS for Use in a Climate Model. *Journal of Hydrometeorology*, *10*(1), 183–198. http://doi.org/10.1175/2008JHM1021.1

Betts, A. K., & Ball, J. H. (1997). Albedo over the boreal forest. *Journal of Geophysical Research*, *102*(D24), 28901. http://doi.org/10.1029/96JD03876

**Response to reviewer #2**

*General comments*

*In this manuscript, the authors presented a new integrated terrestrial model, PALADYN, which includes major physical and biogeochemical processes at an intermediate complexity. The model was developed on the basis of previous models such as LPJ and TRIFFID but includes several recent findings such as new stomatal conductance model. Although many terrestrial models for similar purposes have been developed, this model has several unique and intriguing features. In particular, inclusion of peatland and permafrost carbon dynamics will allow the model to simulate long-term (e.g., glacial time scale) simulations as proposed by the authors. This manuscript includes more than hundred equations, many tables, and schematic diagrams to fully describe the model.*
*To demonstrate the model performance, the authors compared major terrestrial variables with contemporary observational datasets. Overall, these results show good*
*performance of the model, but the authors provided only very brief explanations. Although I agree that scientific insights are not necessarily included into the manuscript, I recommend clarifying the characteristics of the PALADYN model, especially in comparison with other models. Therefore, I conclude that the manuscript needs minor revision before acceptance for publication. Please look my specific comments for details.*

The model evaluation section has been extended to include a more in depth and quantitative analysis of the model performance, including also discussions of the model performance relative to state-of-the-art land surface models, where appropriate. To back up the qualitative analyses of model performance we additionally computed some quantitative metrics like correlation and root mean square error for a number of modelled quantities. These values are directly included in the figures to allow the reader to get a quick quantitative measure of model performance.

*Specific comments*

*Line 103: It seems that the model don't have a separate type for crops. Do you have an idea to include croplands to account for agriculture?*

We are not planning to include crops in the model. A representation of agriculture is out of the scope of the model, which is mainly designed to represent natural processes.

*Line 171: In Eq.(2), the symbol sigma seems to represent Stefan-Boltzman constant, but no definition was provided.*

Sigma is the Stefan-Boltzmann constant and is defined in Table 1. For clarity now this is stated also after Eq. (2).

*Line 286: Vegetation height hv is later estimated by Eq. (81). It is helpful for readers to explain this at this point.*

We added a reference to Eq. (81) as suggested.

*Line 397: In Eq. (42), what kind of phenomenon does the last turnover term represent? Stem flow?*

The last term in Eq. (42) crudely parameterizes all canopy water removal terms excluding evaporation. It therefore includes e.g. stem flow, dripping from the leaves and water removal by wind.

**Line 456: It seems that Figure 2 does not include the surface runoff Rw. Can you include Rw into Figure 2?**

Rw has been included into Figure 2 and several other variables in Figure 2 have been renamed to match the variable names used in the equations.

**Line 566: Is this the single-sided (or projected) specific leaf area?**

Yes, SLA is the one-sided leaf area per leaf carbon mass. This is now explicitly stated in the text.

**Line 606: The statement is at least partially incorrect. In East Siberia, a broad area of forest is dominated by larch, a deciduous needleleaf species.**

The sentence was probably not clear enough and has been reformulated. Since the model has only a single PFT to represent deciduous and evergreen needleleaf trees, both deciduous and evergreen trees share the same PFT-specific parameters (including the specific leaf area, SLA). In the model, deciduous needleleaf trees would be competitive in East Siberia if their SLA would be higher (as would be appropriate according e.g. to the TRY database) than the value of 6 m2/kgC used in the model for needleleaf trees. By assuming that needleleaf trees are evergreen independently of the climatic conditions, we allow some needleleaf forest to grow e.g. in east Siberia, where it would not grow otherwise.

**Line 745: How did you determine the stable and radiocarbon isotope ratios of the atmospheric CO2? As you know, it has been changed by the Suess effect.**

So far, regarding carbon isotopes, in the paper we have shown only the isotopic discrimination during photosynthesis, which does not depend on the isotope ratios of atmospheric CO2. The issue of stable and radiocarbon isotope ratios of the atmospheric CO2 will be discussed in a future paper where carbon isotopes will be evaluated in more detail.

**Line 772: "Manua" should be replaced by "Mauna".**

Corrected.

**Line 821: In terms of wetland extent, the model estimate seems to underestimate in Southeast Asia. I guess that the GIEMS data includes a substantial fraction of paddy fields. Is it correct?**

Yes, the wetlands in Southeast Asia in GIEMS include a substantial fraction from rice cultivations. The paragraph comparing the modelled wetland extent with GIEMS has been expanded in the paper and this issue is now discussed. (see also response to reviewer#1)

**Line 852: Can you say something about the simulated discrimination in relation to C3 and C4 plant distribution? Do you confirm that distribution and functional contribution of C3 and C4 plants were reasonably simulated?**

In the revised version of the manuscript we added some discussion of the simulated discrimination shown in Fig. (21). As shown in Fig. (22) the model reproduces the difference in discrimination between C3 and C4 plants. This, together with the modelled C4/C3 grass distribution, explains the low discrimination values in subtropical areas, particularly in Africa, in agreement with other modelling studies (e.g. Scholze et al. (2003)).

     Snow albedo is parameterized as a function of the solar zenith angle and a snow ageing factor. The diffuse albedo of freshly fallen snow is set to 0.95 in the visible band and to 0.65 in the near infrared band. The actual albedo of snow for diffuse radiation depends on a snow age factor $f_{\text{age}}$:

255    $$\alpha_{\text{sn}}^{\text{vis,dif}} = \alpha_{\text{sn,fresh}}^{\text{vis,dif}} - 0.05 f_{\text{age}}, \tag{17}$$

$$\alpha_{\text{sn}}^{\text{nir,dif}} = \alpha_{\text{sn,fresh}}^{\text{nir,dif}} - 0.25 f_{\text{
[revised manuscript text omitted]
_{\text{lit}}^{\text{peat}}}{\partial t} = \Lambda_{\text{peat}} - k_{\text{lit}}^{\text{peat}} C_{\text{lit}}^{\text{peat}} \tag{98}$$

$$\frac{\partial C_{\text{acro}}}{\partial t} = (1 - f_{\text{lit}}^{\text{resp}}) k_{\text{lit}}^{\text{peat}} C_{\text{lit}}^{\text{peat}} - k_{\text{acro}\rightarrow\text{cato}} C_{\text{acro}} - k_{\text{acro}} C_{\text{acro}} \tag{99}$$

$$\frac{\partial C_{\text{cato}}(z)}{\partial t} = k_{\text{acro}\rightarrow\text{cato}} C_{\text{acro}} - k_{\text{cato}}(z) C_{\text{cato}}(z). \tag{100}$$

The transfer from acrotelm to catotelm carbon occurs only once a critical acrotelm carbon content

725   $C_{\text{acro,crit}} = 5\,\text{kgC}\,\text{m}^{-2}$ is reached, as suggested by (Wania et al., 2009). Typical acrotelm carbon densities are around $20\,\text{kgC}\,\text{m}^{-3}$, so this threshold roughly corresponds to assuming that transfer to the catotelm starts when the actrotelm reaches a thickness of $25\,\text{cm}$, which is a typical value of observed acrotelm thickness. When this threshold is exceeded, acrotelm carbon is transfered to the catotelm in the second soil layer. Peat is assumed to grow thicker by accumulating carbon on top and

730   therefore in the model the catotelm is shifted to lower soil layers once the catotelm carbon density $C_{\text{cato,crit}}\,\rho_{\text{cato}}$ has been exceeded in a given layer. For the same reason the vertical diffusivity of peat carbon is set to 0. Litterfall over peatlands is assumed to be the same as over mineral soil, but to be added to the top soil layer only: $\Lambda_{\text{peat}} = \sum_z \Lambda_{\text{veg}}(z)$. The decomposition rates for litter, acrotelm and catotelm are given by:

735        $$k_{\text{lit}}^{\text{peat}} = k_{\text{lit},10} f_{\text{T}}(1)(f_{\text{oxic}} + (1 - f_{\text{oxic}}) f_{\theta,\text{peat}}) \tag{101}$$

$$k_{\text{acro}} = k_{\text{acro},10} f_{\text{T}}(1)(f_{\text{oxic}} + (1 - f_{\text{oxic}}) f_{\theta,\text{peat}}) \tag{102}$$

$$k_{\text{cato}}(z) = k_{\text{cato},10} f_{\text{T}}(z) f_{\theta,\text{peat}}. \tag{103}$$

Since peatland soil temperature is not separately computed by the model, the temperature factor is calculated using the grid cell mean temperature. $f_{\theta,\text{peat}}$ is taken to be equal to 0.3 as in Wania et al.

740   (2009) and Koven et al. (2013). The values of the reference decomposition rates are given in Table 7. The fraction of litter and acrotelm that is respiring in oxic conditions, $f_{\text{oxic}}$, is determined from the mean grid cell water table depth $z_\nabla$ and the minimum monthly water table position $z_\nabla^{\min}$ assuming that the seasonal water table variations in the peatland fraction follow the grid cell mean water table and that the amplitude of water table variations in peatland is reduced compared to the grid cell mean

745   and limited to the acrotelm thickness:

$$f_{\text{oxic}} = \frac{\min\left(z_{\text{acro}}, \max\left(0, z_\nabla - z_\nabla^{\min}\right)\right)}{z_{\text{acro}}}. \tag{104}$$

Peatland expansion and contraction is modelled partly following Stocker et al. (2014). The grid cell fraction that is wetland for at least 3 months of the year is considered to be potential peatland area $f_{\text{peat}}^{\text{pot}}$. The actual peatland area $f_{\text{peat}}$ is simulated as:

$$f_{\text{peat},n+1} = \begin{cases} \min\left((1+r)f_{\text{peat},n}, f_{\text{peat}}^{\text{pot}}\right) & \frac{\mathrm{d}C_{\text{peat}}}{\mathrm{d}t} \geq \frac{\mathrm{d}C_{\text{peat}}}{\mathrm{d}t}\big|_{\text{crit}} \quad \text{or} \quad C_{\text{peat}} \geq C_{\text{peat}}^{\text{crit}} \\ \max\left((1-r)f_{\text{peat},n}, f_{\text{peat}}^{\text{min}}\right) & \frac{\mathrm{d}C_{\text{peat}}}{\mathrm{d}t} < \frac{\mathrm{d}C_{\text{peat}}}{\mathrm{d}t}\big|_{\text{crit}} \quad \text{and} \quad C_{\text{peat}} < C_{\text{peat}}^{\text{crit}} \end{cases} \quad (105)$$

Peat is expanding if the annual mean rate of carbon uptake ($\mathrm{d}C_{\text{peat}}/\mathrm{d}t$) is greater than a critical value $\mathrm{d}C_{\text{peat}}/\mathrm{d}t\big|_{\text{crit}}$ or if peat carbon exceedes a value $C_{\text{peat}}^{\text{crit}}$, otherwise peatland area is shrinking. To account for inertia in lateral peatland expansion and contraction, the relative areal change rate is limited to $1\,\%\,\text{yr}^{-1}$ ($r = 0.01\,\text{yr}^{-1}$). When the peat area is changing, carbon is simply redistributed between mineral soil and peat carbon pools layer-by-layer with the following rules: $C_{\text{lit}}^{\text{peat}} \leftrightarrow C_{\text{lit}}$, $C_{\text{acro}} \leftrightarrow C_{\text{fast}}$ and $C_{\text{cato}} \leftrightarrow C_{\text{slow}}$.

**6.4 Methane emissions**

Methane emissions are simulated as a constant fraction of heterotrophic respiration when respiration occurs under anaerobic conditions, as is the case in wetlands, peatlands and flooded ocean shelfs. The fraction of carbon that is respired as methane, $f_{CH_4:C}$, is different for wetlands, peatlands and ocean shelf (Table 7).

**6.5 Carbon isotopes: $^{13}$C and $^{14}$C**

The stable carbon isotope $^{13}$C and radiocarbon $^{14}$C are tracked in PALADYN trough all carbon pools in vegetation and soil. Discrimination is simulated only during photosynthesis and follows the model outlined in Lloyd and Farquhar (1994). The discrimination factor $\Delta$ for C3 and C4 photosynthesis is given by:

$$\Delta = \begin{cases} 4.4\frac{c_{\text{a}}-c_{\text{i}}}{c_{\text{a}}} + 27\frac{c_{\text{i}}}{c_{\text{a}}} & \text{C3} \\ 4.4\frac{c_{\text{a}}-c_{\text{i}}}{c_{\text{a}}} + (-5.7 + 20 \cdot 0.35)\frac{c_{\text{i}}}{c_{\text{
[revised manuscript text omitted]
_{\text{can}}$ | $\text{kg m}^{-2}\,\text{s}^{-1}$ | canopy evaporation and sublimation |
|---|---|---|
| $E_{\text{can}}^{\text{s}}$ | $\text{kg m}^{-2}\,\text{s}^{-1}$ | canopy sublimation |
| $E_{\text{can}}^{\text{w}}$ | $\text{kg m}^{-2}\,\text{s}^{-1}$ | canopy evaporation |
| $E_{\text{s}}$ | $\text{kg m}^{-2}\,\text{s}^{-1}$ | snow sublimation |
| $G$ | $\text{W m}^{-2}$ | ground heat flux |
| $H$ | $\text{W m}^{-2}$ | sensible heat flux |
| $I_{\text{can}}^{\text{s}}$ | $\text{kg m}^{-2}\,\text{s}^{-1}$ | canopy snow interception |
| $I_{\text{can}}^{\text{w}}$ | $\text{kg m}^{-2}\,\text{s}^{-1}$ | canopy rain interception |
| $J_{\text{C}}$ | $\text{gC m}^{-2}\,\text{day}^{-1}$ | Rubisco limited photosynthesis rate |
| $J_{\text{E}}$ | $\text{gC m}^{-2}\,\text{day}^{-1}$ | light limited photosynthesis rate |
| $K$ | | Kersten number |
| $L$ | $\text{J kg}^{-1}$ | latent heat of vaporisation |
| $L_{\text{ai}}$ | $\text{m}^2\,\text{m}^{-2}$ | leaf area index |
| $L_{\text{ai,b}}$ | $\text{m}^2\,\text{m}^{-2}$ | balanced leaf area index |
| $L_{\text{f}}$ | $\text{J kg}^{-1}$ | latent heat of fusion of water |
| $LW^{\downarrow}$ | $\text{W m}^{-2}$ | downward longwave radiation at the surface |
| $LW^{\uparrow}$ | $\text{W m}^{-2}$ | upward longwave radiation at the surface |
| $M_{\text{s}}$ | $\text{kg m}^{-2}\,\text{s}^{-1}$ | snowmelt |
| $NPP$ | $\text{kgC m}^{-2}\,\text{s}^{-1}$ | net primary production |
| $P_{\text{s}}$ | $\text{kg m}^{-2}\,\text{s}^{-1}$ | snowfall rate |
| $P_{\text{s,g}}$ | $\text{kg m}^{-2}\,\text{s}^{-1}$ | snowfall rate reaching the ground |
| $P_{\text{r}}$ | $\text{kg m}^{-2}\,\text{s}^{-1}$ | rainfall rate |
| $P_{\text{r,g}}$ | $\text{kg m}^{-2}\,\text{s}^{-1}$ | rainfall rate reaching the ground |
| $R_{\text{d}}$ | $\text{gC m}^{-2}\,\text{day}^{-1}$ | leaf respiration |
| $R_{\text{i}}$ | | bulk Richardson number |
| $R_{\text{w}}$ | $\text{kg m}^{-2}\,\text{s}^{-1}$ | surface water runoff |
| $SLA$ | $\text{m}^2\,\text{kgC}^{-1}$ | specific leaf area |
| $S_{\text{ai}}$ | $\text{m}^2\,\text{m}^{-2}$ | stem area index |
| $SW^{\downarrow}$ | $\text{W m}^{-2}$ | downward shortwave radiation at the surface |
| $T_0$ | K | freezing temperature of water |
| $T_{\star}$ | K | skin temperature |
| $T_{\text{a}}$ | K | air temperature at height $z_{\text{ref}}$ |
| $T_{\text{cmon}}^{\text{max}}$ | K | maximum coldest month temperature for establishment |
| $T_{\text{cmon}}^{\text{min}}$ | K | minimum coldest month temperature for establishment |
| $T_{\text{cmon}}^{\text{phen}}$ | K | coldest month temperature for phenology |
| $T_{\text{gdd}}^{\text{base}}$ | K | base temperature for phenology |

[revised manuscript text omitted]